# QUANTIZED OPTIMISTIC DUAL AVERAGING WITH ADAPTIVE LAYER-WISE COMPRESSION

## ABSTRACT

We develop a general layer-wise and adaptive compression framework with applications to solving variational inequality problems (VI) in a large-scale and distributed setting where multiple nodes have access to local stochastic dual vectors. This framework encompasses a broad range of applications, spanning from distributed optimization to games. We establish tight error bounds and code-length bounds for *adaptive layer-wise* quantization that generalize previous bounds for global quantization. We also propose Quantized and Generalized Optimistic Dual Averaging (QODA) with adaptive learning rates, which achieves optimal rate of convergence for distributed monotone VIs. We empirically show that the adaptive layer-wise compression achieves up to a $150\%$ speedup in end-to-end training time for training Wasserstein GAN on $12+$ GPUs.

## 1 INTRODUCTION

Under classical learning theory setting, if we have sufficient training samples, computational resources, along with a powerful *first-order method* to *optimize* an empirical risk properly, then the output of the first-order method is expected to achieve a small test error. For high-dimensional and non-convex settings with deep neural networks (DNNs), minimizing the empirical risk is a challenging optimization task due to non-convexity and lack of guarantees in terms of global optimality. Beyond empirical risk minimization, formulating the problems of training generative adversarial networks (GANs) (Goodfellow et al., 2020) and equilibrium in more general and possibly non-zero-sum game-theoretic settings require more complicated mathematical frameworks. Variational inequality (VI) is a mathematical framework for modeling equilibrium problems (Facchinei & Pang, 2003; Bauschke & Combettes, 2017; Antonakopoulos et al., 2021), e.g., in applications such as robust adversarial reinforcement learning (Pinto et al., 2017), auction theory (Syrgkanis et al., 2015), and adversarially robust learning (Schmidt et al., 2018). For an operator $A : \mathbb{R}^d \to \mathbb{R}^d$, a VI finds some $\boldsymbol{x}^\star \in \mathbb{R}^d$ such that

$$\langle A(\boldsymbol{x}^\star), \boldsymbol{x} - \boldsymbol{x}^\star \rangle \geq 0 \ \text{ for all } \ \boldsymbol{x} \in \mathbb{R}^d. \tag{VI}$$

In terms of implementation in a synchronous system with $K$ nodes, first-order solvers for empirical risk minimization and VI-solvers are scaled by distributing computation among nodes, e.g., by partitioning the entire dataset in a cloud data center, followed by aggregation of local computations.[1] Nodes can be, e.g., hospitals and cellphones that train a global model or personalized models collaboratively in a federated learning setting.

In large-scale settings, communication costs for broadcasting huge stochastic gradients and dual vectors is the main performance bottleneck (Strom, 2015; Alistarh et al., 2017; Kairouz et al., 2021; Ramezani-Kebrya et al., 2023). Several methods have been proposed to accelerate large-scale training such as quantization, sparsification, and reducing the frequency of communication through local updates (Kairouz et al., 2021). In particular, *unbiased quantization* is unique due to both enjoying strong theoretical guarantees along with providing communication efficiency on the fly, i.e., it converges under the same hyperparameters tuned for uncompressed variants while providing substantial savings in communication costs (Alistarh et al., 2017; Ramezani-Kebrya et al., 2023).

Popular DNNs including convolutional architectures, transformers, and vision transformers have various *types of layers* such as feed-forward, residual, multi-head attention including self-attention

---

[1]For simplicity, in the following, we use the term *node* to refer to client, FPGA, APU, CPU, GPU, worker.

and cross-attention, bias, and normalization layers (He et al., 2016; Vaswani et al., 2017; Dosovitskiy et al., 2021). Different types of layers learn different types of features. They are also diverse in terms of number of parameters and their impact on the final accuracy (Dutta et al., 2020; Xin et al., 2023; Li et al., 2024). Similar heterogeneity has been observed for attention layers in large-scale transformers (Markov et al., 2022). The current communication-efficient literature does not rigorously take into account heterogeneity in terms of representation power, impact on the final learning outcome, and statistical heterogeneity across various layers of neural networks and across training for each layer. Recently, layer-wise and adaptive compression schemes have shown tremendous empirical success in accelerating training deep neural networks and transformers in large-scale settings (Markov et al., 2022; 2024), but they yet to have theoretical guarantees and to handle statistical heterogeneity over the course of training. Hence, these layer-wise compression schemes suffer from a dearth of generalization and statistically rigorous argument to optimize the sequence of quantization and the number of sparsification levels for each layer.

## 1.1 SUMMARY OF CONTRIBUTIONS

- We propose a theoretical framework for **layer-wise** and **adaptive** unbiased quantization schemes with novel fine-grained coding protocol analysis. We also establish tight variance and code-length bounds, which *encompass* the empirical layer-wise quantization methods in the literature (Markov et al., 2022; 2024) and *generalize* those bounds of global quantization frameworks (Alistarh et al., 2017; Faghri et al., 2020; Ramezani-Kebrya et al., 2021).

- To the best of our knowledge, this work is the first to incorporate optimism in solving distributed VI with *adaptive learning rates* and layer-wise quantization. In particular, we propose Quantized Optimistic Dual Averaging (QODA) and establish joint convergence and communication guarantees for QODA with the competitive rates $\mathcal{O}(1/\sqrt{T})$ and $\mathcal{O}(1/T)$ under absolute and relative noise models, respectively. Importantly, we obtain these bounds **without the restrictive almost sure boundedness assumption** of stochastic dual vectors that is essential related VI works (Bach & Levy, 2019; Hsieh et al., 2021; Antonakopoulos et al., 2021) including *the global quantization distributed VI-solver* Q-GenX (Ramezani-Kebrya et al., 2023).

- Empirically, we show that QODA with layer-wise compression significantly improves the convergence and training time compared to both the global quantization baseline Q-GenX (Ramezani-Kebrya et al., 2023) and the (uncompressed) full precision baseline. Indeed, QODA achieves up to a **150% speedup** in terms of end-to-end training time in an application of training Wasserstein Generative Adversarial Network (WGAN) (Arjovsky et al., 2017) on $12+$ GPUS.

## 1.2 RELATED WORKS

For empirical risk minimization, *adaptive quantization*, has been proposed to adapt quantization levels (Faghri et al., 2020; Wang et al., 2018; Makarenko et al., 2022) and the number of quantization levels (Guo et al., 2020; Agarwal et al., 2021) over the trajectory of optimization. All these quantization schemes are *global w.r.t. layers* and do not take into account heterogeneities in terms of representation power and impact on the final learning outcome across various layers of neural networks and across training for each layer. Markov et al. (2022; 2024) have proposed unbiased and *layer-wise quantization* where quantization parameters are updated across layers in a heuristic manner and have shown tremendous empirical success in training popular DNNs in large-scale settings. However, these current layer-wise schemes do not have strong theoretical guarantees and do not handle statistical heterogeneity over the course of training.

The layer-wise structure of DNNs have been leveraged in the development of different compression ideas with sketches or bandwidth awareness (Li et al., 2024; Xin et al., 2023). Moreover, several works (Mishchenko et al., 2024; Horváth et al., 2023; Wang et al., 2022) study *block quantization* that divides the operator/vector into different blocks before quantization. In Appendix A.2, we will show our layer-wise quantization is fundamentally different from block quantization.

There is a line of research that focuses on designing *distributed methods for VI and saddle points problems*. Kovalev et al. (2022) consider strongly monotone VI; Beznosikov et al. (2023b) concern with VI problems under co-coercivity assumptions. Assumptions such as strong monotonicity and co-coercivity are quite restrictive in ML applications. Beznosikov et al. (2022; 2023a) consider

VI problems with finite sum structure with an extra $\delta$-similarity assumption in (Beznosikov et al., 2023a) . Several works (Duchi et al., 2011; Yuan et al., 2012; Tsianos & Rabbat, 2012) explore *dual averaging* for distributed finite-sum minimization in networks. We include further related works on unbiased quantization and optimistic gradients in Appendix A.1. While the settings might vary, our work is the first that 1) introduces optimism in distributed VI with adaptive learning rates; 2) develops layer-wise quantization with joint convergence and communication guarantees; 3) shows improvements in end-to-end training time in multi-node setting with efficient implementation. A detailed comparison with the related methods is in Appendix A.2.

## 2 PRELIMINARIES

A summary of commonly used notations in this paper is provided in Appendix A.3. Given an operator $A : \mathbb{R}^d \to \mathbb{R}^d$, we consider these standard assumptions:

**Assumption 2.1** (Monotonicity). We have that for all $\boldsymbol{x}, \hat{\boldsymbol{x}} \in \mathbb{R}^d$, $\langle A(\boldsymbol{x}) - A(\hat{\boldsymbol{x}}), \boldsymbol{x} - \hat{\boldsymbol{x}} \rangle \geq 0$.

**Assumption 2.2** (Solution Existence). The solution set $\mathcal{X}^\star := \{\boldsymbol{x}^\star \in \mathbb{R}^d : \boldsymbol{x}^\star \text{solves (VI)}\} \neq \emptyset$.

**Assumption 2.3** (*L*-Lipschitz). Let $L \in \mathbb{R}^+$. Then an operator $A$ is $L$-Lipschitz if

$$\|A(\boldsymbol{x}) - A(\boldsymbol{x}')\|_* \leq L\|\boldsymbol{x} - \boldsymbol{x}'\| \quad \forall \boldsymbol{x}, \boldsymbol{x}' \in \mathbb{R}^d.$$

In this work, we consider methods that rely on a so-called *stochastic first-order oracle* (Nesterov, 2004). This oracle, when called at $\boldsymbol{x}$, draws an i.i.d. sample $\omega$ from a complete probability space $(\Omega, \mathcal{F}, \mathbb{P})$ and returns a *stochastic dual vector* $g(\boldsymbol{x}; \omega)$ given by:

$$g(\boldsymbol{x}; \omega) = A(\boldsymbol{x}) + U(\boldsymbol{x}; \omega), \tag{1}$$

where $U(\boldsymbol{x}; \omega)$ denotes the (possibly random) error in the measurement or noise. Next, we consider two important noise profiles: absolute noise and relative noise, formally defined as:

**Assumption 2.4** (Absolute Noise). Let $\boldsymbol{x} \in \mathbb{R}^d$ and $\omega \sim \mathbb{P}$. The oracle $g(\boldsymbol{x}; \omega)$ satisfies unbiasedness $\mathbb{E}[g(\boldsymbol{x}; \omega)] = A(\boldsymbol{x})$ and bounded absolute variance $\mathbb{E}\left[\|U(\boldsymbol{x}, \omega)\|_*^2\right] \leq \sigma^2$.

As the noise variance is independent of the value of the operator at the queried point, this type of randomness is *absolute*. This noise profile is common in the (distributed) VI literature (Woodworth et al., 2021; Ene & Le Nguyen, 2022; Tupitsa et al., 2024). It is also known as the bounded variance assumption in stochastic optimization literature (Nemirovski et al., 2009; Juditsky et al., 2011). Alternatively, a more favorable noise profile is observed when the stochastic error vanishes as we approach a solution of VI. This is formally captured by the notion of *relative noise* (Polyak, 1987):

**Assumption 2.5** (Relative Noise). Let $\boldsymbol{x} \in \mathbb{R}^d$ and $\omega \sim \mathbb{P}$. The oracle $g(\boldsymbol{x}; \omega)$ satisfies: unbiasedness $\mathbb{E}[g(\boldsymbol{x}; \omega)] = A(\boldsymbol{x})$ and bounded relative variance $\mathbb{E}\left[\|U(\boldsymbol{x}, \omega)\|_*^2\right] \leq \sigma_R \|A(\boldsymbol{x})\|_*^2$.

This relative noise model has been studied in several ML problems such as over-parameterization (Oymak & Soltanolkotabi, 2020), representation learning (Zhang et al., 2021), and multi-agent learning (Lin et al., 2020). In Appendix B.3, we provide more specific relative noise examples. This model may result in obtaining the well-known order-optimal rate of $\mathcal{O}(1/T)$ in deterministic settings.

Let $\mathcal{X} \subset \mathbb{R}^d$ denote a non-empty and compact test domain. The main measure to evaluate the quality of a candidate solution is the restricted gap function (Nesterov, 2009; Antonakopoulos et al., 2019) (more properties in Appendix B.1):

$$\text{GAP}_{\mathcal{X}}(\hat{\boldsymbol{x}}) = \sup_{\boldsymbol{x} \in \mathcal{X}} \langle A(\boldsymbol{x}), \hat{\boldsymbol{x}} - \boldsymbol{x} \rangle. \tag{GAP}$$

*Remark* 2.6. The majority of adaptive methods for VI literature (Bach & Levy, 2019; Hsieh et al., 2021; Antonakopoulos et al., 2021) including the baseline Q-GenX (Ramezani-Kebrya et al., 2023) assume almost sure boundedness of stochastic dual vectors under both absolute and relative noise profiles. In addition, previous theoretical results on global quantization are established under a similar assumption with bounded second moments of stochastic gradients (Alistarh et al., 2017; Ramezani-Kebrya et al., 2021; Faghri et al., 2020). In Section 4, we establish the joint convergence and communication guarantees of our VI-solver with layer-wise quantization without this assumption.

## 3 Quantized Optimistic Dual Averaging

Consider a distributed and synchronous setting with $K$ nodes, along the lines of the standard setting for data-parallel SGD (Dean et al., 2012; Alistarh et al., 2017). Here, the nodes partition the entire dataset among themselves such that each node retains only a local copy of the current parameter vector while having access to independent private stochastic dual vectors. In each iteration, each node receives stochastic dual vectors, aggregates them, computes an update, and broadcasts the compressed update to accelerate training. These compressed updates are decompressed before the next aggregation step at each node.

### 3.1 Adaptive Layer-wise Quantization

Adaptive layer-wise quantization has only been studied empirically in (Markov et al., 2022; 2024) with promising results in applications such as Transformer-XL on WikiText-103 and ResNet50 on CIFAR-100 training. Our goal is to provide a novel **general formulation** considering **statistical heterogeneity** across layers and establish **theoretical guarantees** for adaptive layer-wise quantization with tailored coding schemes.

We first outline the general formulation for layer-wise and unbiased quantization. We study unbiased compression, where, in expectation, the output of the decompression of a compressed vector is equal to the original uncompressed vector. Let $V_{k,t}$ and $\hat{V}_{k,t}$ denote the uncompressed and compressed stochastic dual vector in node $k$ at time $t$, respectively. Let $\boldsymbol{v} \in \mathbb{R}^d$ be a vector to be quantized. For $i = 1, \ldots, d$, let $u_i = |v_i|/\|\boldsymbol{v}\|_q$ be the normalized coordinate. At each time $t$, instead of a global sequence of quantization levels for all coordinates (Alistarh et al., 2017; Ramezani-Kebrya et al., 2023), we consider a set $\mathbb{L}^{t,M}$ of $M$ types of sequences $\{\boldsymbol{\ell}^{t,1}, \ldots, \boldsymbol{\ell}^{t,M}\}$ to be optimized with flexible and adjustable numbers of levels $\alpha_1, \ldots, \alpha_M$, respectively. We denote $\boldsymbol{\ell}^{t,m} \in \mathbb{L}^{t,M}$ the sequence of type $m$ at time $t$, given by $[\ell_0, \ell_1^{t,m}, \ldots, \ell_{\alpha_m}^{t,m}, \ell_{\alpha_m+1}]^\top$, where $0 = \ell_0 < \ell_1^{t,m} < \cdots < \ell_{\alpha_m}^{t,m} < \ell_{\alpha_m+1} = 1$. Let $\mathbb{S}^{t,m}$ be the set of all normalized coordinates that use type $m$ sequence $\boldsymbol{\ell}^{t,m}$ at time $t$. Let $\tau^{t,m}(u)$ denote the index of a level with respect to $u \in [0,1]$ such that $\ell_{\tau^{t,m}(u)}^{t,m} \leq u < \ell_{\tau^{t,m}(u)+1}^{t,m}$. Let $\xi^{t,m}(u) = (u - \ell_{\tau^{t,m}(u)}^{t,m})/(\ell_{\tau^{t,m}(u)+1}^{t,m} - \ell_{\tau^{t,m}(u)}^{t,m})$ be the relative distance of $u$ to the level $\tau^{t,m}(u) + 1$. Define the following random variable

$$q_{\boldsymbol{\ell}^{t,m}}(u) = \begin{cases} \ell_{\tau^{t,m}(u)}^{t,m} & \text{with probability } 1 - \xi^{t,m}(u); \\ \ell_{\tau^{t,m}(u)+1}^{t,m} & \text{with probability } \xi^{t,m}(u). \end{cases}$$

We then define the random quantization of vector $\boldsymbol{v}$ as $Q_{\mathbb{L}^{t,M}}(\boldsymbol{v}) = [Q_{\mathbb{L}^{t,M}}(v_1), \ldots, Q_{\mathbb{L}^{t,M}}(v_d)]^\top$ where for $m = 1, 2, \ldots, M$, and any $u_i \in \mathbb{S}^{t,m}$, we have $Q_{\mathbb{L}^{t,M}}(v_i) = \|\boldsymbol{v}\|_q \cdot \text{sign}(v_i) \cdot q_{\boldsymbol{\ell}^{t,m}}(u_i)$. Let $\boldsymbol{q}_{\mathbb{L}^{t,M}} \sim \mathbb{P}_Q$ represent $d$ variables $\{q_{\boldsymbol{\ell}^{t,m}}(u_i)\}_{i \in [d]}$ sampled independently for random quantization. As this scheme is unbiased, we can measure the quantization error by measuring the variance $\mathbb{E}_{\boldsymbol{q}_{\mathbb{L}^{t,M}}}[\|Q_{\mathbb{L}^{t,M}}(\boldsymbol{v}) - \boldsymbol{v}\|_2^2]$ given by

$$\|\boldsymbol{v}\|_q^2 \sum_{m=1}^{M} \sum_{u_i \in \mathbb{S}^{t,m}} \sigma_Q^2(u_i; \boldsymbol{\ell}^{t,m}), \tag{Var}$$

where $\sigma_Q^2(u_i; \boldsymbol{\ell}^{t,m}) = \mathbb{E}[(q_{\boldsymbol{\ell}^{t,m}}(u_i) - u_i)^2] = (\ell_{\tau^{t,m}(u_i)+1}^{t,m} - u_i)(u_i - \ell_{\tau^{t,m}(u_i)}^{t,m})$ is the variance of quantization of a single coordinate $u_i \in \mathbb{S}^{t,m}$ with type $m$ sequence $\boldsymbol{\ell}^{t,m}$. We can optimize $M$ quantization sequences by minimizing the overall quantization variance

$$\min_{\mathbb{L}^{t,M} \in \mathcal{L}^{t,M}} \mathbb{E}_\omega \mathbb{E}_{\boldsymbol{q}_{\mathbb{L}^{t,M}}} \left[ \|Q_{\mathbb{L}^{t,M}}(g(\boldsymbol{x}_t; \omega)) - A(\boldsymbol{x}_t)\|_2^2 \right],$$

where $\mathcal{L}^{t,M} = \left\{ \{\boldsymbol{\ell}^{t,1}, \ldots, \boldsymbol{\ell}^{t,M}\} : \forall m \in [M], \forall j \in [\alpha_m], \ell_j^{t,m} \leq \ell_{j+1}^{t,m}, \ell_0 = 0, \ell_{\alpha_m+1} = 1 \right\}$, denoting the collection of all feasible sets of type $m$ levels. Since random quantization and random samples are statistically independent, the above minimization is equivalent to

$$\min_{\mathbb{L}^{t,M} \in \mathcal{L}^{t,M}} \mathbb{E}_\omega \mathbb{E}_{\boldsymbol{q}_{\mathbb{L}^{t,M}}} \left[ \|Q_{\mathbb{L}^{t,M}}(g(\boldsymbol{x}_t; \omega)) - g(\boldsymbol{x}_t; \omega)\|_2^2 \right]. \tag{MQV}$$

*Remark* 3.1. We now elaborate on how *layer-wise quantization is always better than global quantization* such as (Alistarh et al., 2017; Faghri et al., 2020; Ramezani-Kebrya et al., 2021; 2023). We

optimize $M$ quantization sequences by minimizing quantization variance (MQV). Global quantization models will find an overall optimum sequence $\ell_*^t$ for all the $M$ types. Hence, the collection of $M$ sequences in this global case is simply $\mathbb{L}_{glb}^{t,M} = \{\ell_*^t, .., \ell_*^t\}$, where $\ell_*^t$ repeats $M$ times. By the minimality of (MQV), we obtain the quantization variance for layer-wise quantization is always upper bounded by that of global quantization:

$$\min_{\mathbb{L}^{t,M}} \mathbb{E}\left[\|Q_{\mathbb{L}^{t,M}}(g(\boldsymbol{x}_t;\omega)) - g(\boldsymbol{x}_t;\omega)\|_2^2\right] \leq \mathbb{E}\left[\|Q_{\mathbb{L}_{glb}^{t,M}}(g(\boldsymbol{x}_t;\omega)) - g(\boldsymbol{x}_t;\omega)\|_2^2\right].$$

### 3.2 ENCODING

Coding schemes are applied on top of our layer-wise quantization to further reduce communication costs. We now introduce two practical coding protocols for layer-wise quantization that *require a fine-grained analysis* different from those for global quantization (Alistarh et al., 2017; Faghri et al., 2020; Ramezani-Kebrya et al., 2021; 2023). For some $q \in \mathbb{Z}_+$, any vector $\boldsymbol{v} \in \mathbb{R}^d$ can be uniquely represented by a tuple $(\|\boldsymbol{v}\|_q, \boldsymbol{s}, \boldsymbol{u})$ where $\|\boldsymbol{v}\|_q$ is the $L^q$ norm of $\boldsymbol{v}$, $\boldsymbol{s} := [\text{sign}(v_1), \ldots, \text{sign}(v_d)]^\top$ comprises of signs of each coordinate $v_i$, and $\boldsymbol{u} := [u_1, \ldots, u_d]^\top$, where $u_i = |v_i|/\|\boldsymbol{v}\|_q$, are the normalized coordinates. Note that $0 \leq u_i \leq 1$ for all $i \in [d]$.

#### 3.2.1 CODING PROTOCOL 1

Let $\mathcal{A}^{t,m} = \{\ell_0^{t,m}, \ell_1^{t,m}, \ldots, \ell_{\alpha_m}^{t,m}, \ell_{\alpha_m+1}^{t,m}\}$ be the collection of all the levels of the sequence $\boldsymbol{\ell}^{t,m}$. Let $\Omega^{t,M} = \bigcup_{m=1}^M \mathcal{A}^{t,m}$ be the collection of all the levels of $M$ sequences at time $t$. The overall encoding, i.e., composition of coding and quantization, $\text{ENC}(\|\boldsymbol{v}\|_q, \boldsymbol{s}, \boldsymbol{q}_{\mathbb{L}^{t,M}}) : \mathbb{R}_+ \times \{\pm 1\}^d \times (\Omega^{t,M})^d \to \{0,1\}^*$ uses a standard floating point encoding with $C_q$ bits to represent the non-negative scalar $\|\boldsymbol{v}\|_q$, encodes the sign of each coordinate with one bit, and then utilizes an integer encoding scheme $\Psi : (\Omega^{t,M})^d \to \{0,1\}^*$ to efficiently encode every quantized coordinate with the minimum expected code-length. To solve (MQV), we sample $Z$ stochastic dual vectors $\{g(\boldsymbol{x}_t;\omega_1), \ldots, g(\boldsymbol{x}_t;\omega_Z)\}$. Let $F_z$ denote the marginal cumulative distribution function (CDF) of normalized coordinates conditioned on observing $\|g(\boldsymbol{x}_t;\omega_z)\|_q$. By law of total expectation, for $\mathbb{L}^{t,M} \in \mathcal{L}^{t,M}$, (MQV) can be approximated by:

$$\min_{\mathbb{L}^{t,M}} \sum_{z=1}^Z \|g(\boldsymbol{x}_t;\omega_z)\|_q^2 \sum_{m=1}^M \sum_{i=0}^{\alpha_m} \int_{\ell_i^{t,m}}^{\ell_{i+1}^{t,m}} \sigma_Q^2(u; \boldsymbol{\ell}^{t,m}) \, dF_z(u) \quad \text{or} \quad \min_{\mathbb{L}^{t,M}} \sum_{m=1}^M \sum_{i=0}^{\alpha_m} \int_{\ell_i^{t,m}}^{\ell_{i+1}^{t,m}} \sigma_Q^2(u; \boldsymbol{\ell}^{t,m}) \, d\tilde{F}(u), \quad (2)$$

where $\tilde{F}(u) = \sum_{z=1}^Z \lambda_z F_z(u)$ is the weighted sum of the conditional CDFs with

$$\lambda_z = \|g(\boldsymbol{x}_t;\omega_z)\|_q^2 / \sum_{z=1}^Z \|g(\boldsymbol{x}_t;\omega_z)\|_q^2. \quad (3)$$

#### 3.2.2 CODING PROTOCOL 2

With $M$ types of sequences, we call a coordinate of type $m$ at time $t$ if it is quantized with type $m$ sequence $\boldsymbol{\ell}^{t,m}$. Protocol 2 processes the coordinates of $M$ types in parallel. Each type has its own code-book where different types may share code-words to minimize the code-length, but the receiver knows the type of any code when decoding. The overall composition of coding and quantization, $\text{ENC}(\|\boldsymbol{v}\|_q, \boldsymbol{s}, \boldsymbol{q}_{\mathbb{L}^{t,M}})$ consists of $M$ parallel encoding maps $\text{ENC}(\|\boldsymbol{v}\|_q, \boldsymbol{s}, \boldsymbol{q}_{\boldsymbol{\ell}^{t,m}})$ uses a standard floating point encoding with $C_q$ bits to represent the positive scalar $\|\boldsymbol{v}\|_q$, encodes the sign of each type $m$ coordinate with one bit, and then utilizes correspondingly type $m$ *integer* encoding scheme $\Psi^m : \mathcal{A}^{t,m} \to \{0,1\}^*$ to *efficiently* encode every type $m$ quantized coordinate with the *minimum* expected code-length. To solve (MQV) for Protocol 2, we first sample $Z$ stochastic dual vectors $\{g(\boldsymbol{x}_t;\omega_1), \ldots, g(\boldsymbol{x}_t;\omega_Z)\}$. Let $F_z^m$ denote the marginal CDF of normalized coordinates of type $m$ conditioned on observing $\|g(\boldsymbol{x}_t;\omega_z)\|_q$. Note that, in this Protocol 2, we have $M$ marginal CDFs corresponding to $m$ types instead of only one marginal CDF in Protocol 1. By the law of total expectation, (MQV) can be approximated by solving $M$ problems *in parallel* for each $\boldsymbol{\ell}^{t,m}$:

$$\min_{\boldsymbol{\ell}^{t,m}} \sum_{z=1}^Z \|g(\boldsymbol{x}_t;\omega_z)\|_q^2 \sum_{i=0}^{\alpha_m} \int_{\ell_i^{t,m}}^{\ell_{i+1}^{t,m}} \sigma_Q^2(u; \boldsymbol{\ell}^{t,m}) \, dF_z^m(u) \quad \text{or} \quad \min_{\boldsymbol{\ell}^{t,m}} \sum_{i=0}^{\alpha_m} \int_{\ell_i^{t,m}}^{\ell_{i+1}^{t,m}} \sigma_Q^2(u; \boldsymbol{\ell}^{t,m}) \, d\tilde{F}^m(u), \quad (4)$$

where $\tilde{F}^m(u) = \sum_{z=1}^Z \lambda_z F_z^m(u)$ is the weighted sum of the conditional CDFs of normalized coordinates of type $m$ with weights $\lambda_z$ similar to (3). In our implementation (details in Section 6), we

utilize L-GreCo (Markov et al., 2024) which executes a dynamic programming algorithm optimizing the total compression ratio while minimizing compression error (MQV) from (2) or (4).

The decoding $\text{DEC} : \{0,1\}^* \to \mathbb{R}^d$ first reads $C_q$ bits to reconstruct $\|\boldsymbol{v}\|_q$, then applies decoding schemes $(\Psi^m)^{-1} : \{0,1\}^* \to \mathcal{A}^{t,m}$ to obtain normalized type $m$ coordinates without confusion since the number of coordinates $|\mathbb{S}^{t,m}|$, their order, and the corresponding code-book are known at the decoder. The discussion for the choice of a specific lossless prefix code and more details on coding schemes are included in Appendix D.1.

*Remark* 3.2. We note that Protocol 2 offers *higher compression ratios* through code-word sharing across different types. The improved compression ratio comes at the expense of increased encoding and decoding complexity along with possibility of increased re-transmission overhead in case of unstable networking environment. When the end-to-end delay for message passing in the underlying network is highly random such as jitters (Verma et al., 1991), Protocol 1 will be optimal since every quantization level for every type has a unique code-word. However, Protocol 2 will possibly require several transmissions in case of unstable networks. When the network is stable and delays are deterministic, we propose to adopt Protocol 2. Our coding alternatives provide a trade-off between compression ratio, re-transmission probability, and encoding/decoding complexity.

### 3.3 OPTIMISTIC DUAL AVERAGING

Our described layer-wise quantization and coding protocols are **general** with applications such as empirical risk minimization by training transformers (Markov et al., 2022; 2024). In this section, we will show one such application with our novel *Quantized Optimistic Dual Averaging (QODA)*, Algorithm 1, to efficiently solve distributed VI. Importantly, this optimistic approach **reduces one "extra" gradient step** that extra gradient methods and variants such as Q-GenX (Ramezani-Kebrya et al., 2023) take (by storing the gradient from the previous iteration, refer to line 9 and 16). Therefore, QODA **reduces the communication burden by half** decoupled from acceleration due to quantization compared to Q-GenX. At certain steps, every node calculates the sufficient statistics of a parametric distribution to estimate distribution of dual vectors in lines 3 to 5. Let $\hat{V}_{k,t} = Q(V_{k,t}) = Q(A_k(X_t) + U_k(X_t))$ denote the unbiased and quantized stochastic dual vectors for node $k \in [K]$ and iteration $t \in [T]$. The *optimistic dual averaging* updates in (5) appear in lines 10, 17 and 18. Our layer-wise quantization with $Q_{\mathbb{L}^{t,M}}$ and coding protocols are applied in lines 12 and 15. The loops are executed *in parallel* on the nodes.

---

**Algorithm 1:** Quantized Optimistic Dual Averaging

**Require:** Local training data; local copies of $X_t, Y_t$; update steps set $\mathcal{U}$; learning rates $\{\gamma_t\}, \{\eta_t\}$

1: **for** $t = 1$ **to** $T$ **do**
2:     **if** $t \in \mathcal{U}$ **then**
3:         **for** $i = 1$ **to** $K$ **do**
4:             Efficiently estimate distributions of normalized dual vectors and update $\mathbb{L}^{t,M}$ [a]
5:             Update $M$ sequences of levels *in parallel*
6:         **end for**
7:     **end if**
8:     **for** $i = 1$ **to** $K$ **do**
9:         Retrieve previously stored $\hat{V}_{k,t-1/2}$
10:       $X_{t+1/2} \leftarrow X_t - \gamma_t \sum_{k=1}^{K} \hat{V}_{k,t-1/2}/K$
11:       $V_{i,t+1/2} \leftarrow A_i(X_{t+1/2}) + U_i(X_{t+1/2})$
12:       $d_{i,t} \leftarrow \text{ENCODE}\left(Q_{\mathbb{L}^{t,M}}(V_{i,t+1/2}); \mathbb{L}^{t,M}\right)$
13:       Broadcast $d_{i,t}$
14:       Receive $d_{i,t}$ from each node $i$
15:       $\hat{V}_{i,t+1/2} \leftarrow \text{DECODE}(d_{i,t}; \mathbb{L}^{t,M})$
16:       Store $\hat{V}_{k,t+1/2}$
17:       $Y_{t+1} \leftarrow Y_t - \sum_{k=1}^{K} \hat{V}_{k,t+1/2}/K$
18:       $X_{t+1} \leftarrow \eta_{t+1} Y_{t+1} + X_1$
19:     **end for**
20: **end for**

---

[a]Additional details are provided in Remark 3.3.

$$X_{t+1/2} = X_t - \gamma_t \sum_{k=1}^{K} \hat{V}_{k,t-1/2}/K; \; Y_{t+1} = Y_t - \sum_{k=1}^{K} \hat{V}_{k,t+1/2}/K; \; X_{t+1} = X_1 + \eta_{t+1} Y_{t+1}. \quad (5)$$

In general, learning rates $\gamma_t$ and $\eta_t$ can be chosen such that they are non-increasing and $\gamma_t \geq \eta_t > 0$. We propose the following *adaptive* learning rate schedules for updates (5) and in Algorithm 1.

$$\eta_t = \gamma_t = \left(1 + \sum_{s=1}^{t-1} \sum_{k=1}^{K} \left\|\hat{V}_{k,s+1/2} - \hat{V}_{k,s-1/2}\right\|_*^2 / K^2\right)^{-1/2}. \quad (6)$$

The two learning rates here are equal, but they can be different in an alternative setting in Section 5.

*Remark* 3.3. One possible solution of efficiently estimating the distributions of dual vectors (line 4 in Algorithm 1) is to use a parametric model of density estimation such as modelling via truncated normal with efficiently computing sufficient statistics (Faghri et al., 2020). The set of update steps set $U$ in Algorithm 1 is determined by the dynamics of distribution of of normalized dual vectors over the course of training. In Section 6, we use L-GreCo (Markov et al., 2024) to update the levels.

## 4 THEORETICAL GUARANTEES

### 4.1 COMPRESSION BOUNDS

We first establish a variance bound for a general layer-wise and unbiased quantization scheme. We drop time index $t$ for notation simplicity. Let $q \in \mathbb{Z}_+$. Let $\bar{\ell}^m = \max_{0 \leq j \leq \alpha_m} \ell_{j+1}^m / \ell_j^m$, and $\bar{\ell}^M = \max_{1 \leq m \leq M} \bar{\ell}^M$. Denote the largest level 1 across $M$ types $\bar{\ell}_1^M = \max_{1 \leq m \leq M} \ell_1^M$. Let $d_{th} = (2/\bar{\ell}_1^M)^{\min\{2,q\}}$. We now present the variance bounds for our layer-wise quantization schemes:

**Theorem 4.1** (Quantization Variance Bound). *Let $\boldsymbol{v} \in \mathbb{R}^d$ be a vector to be quantized with $L^q$ normalization. With unbiased quantization of $\boldsymbol{v}$, i.e., $\mathbb{E}_{q_{\mathbb{L}^M}}[Q_{\mathbb{L}^M}(\boldsymbol{v})] = \boldsymbol{v}$, we have that*

$$\mathbb{E}_{q_{\mathbb{L}^M}} \left[ \|Q_{\mathbb{L}^M}(\boldsymbol{v}) - \boldsymbol{v}\|_2^2 \right] \leq \varepsilon_Q \|\boldsymbol{v}\|_2^2, \tag{7}$$

*where $\varepsilon_Q = \frac{(\bar{\ell}^M - 1)^2}{4\bar{\ell}^M} + \frac{(\bar{\ell}_1^M)^2 d^{2/\min\{q,2\}} \mathbb{1}\{d < d_{th}\}}{4} + (\bar{\ell}_1^M d^{2/\min\{q,2\}} - 1) d^{2/\min\{q,2\}} \mathbb{1}\{d \geq d_{th}\}$.*

The proof is provided in Appendix C.

*Remark* 4.2. For the special case of $M = 1$, our bound (7) recovers (Ramezani-Kebrya et al., 2023, Theorem 1), matching the lower bound $\Omega(d)$ in the regime of large $d$ and $L^2$ normalization. Moreover, under $M = 1$, this bound holds for general $L^q$ normalization and arbitrary sequence of quantization levels as opposed to (Alistarh et al., 2017, Theorem 3.2) and (Ramezani-Kebrya et al., 2021, Theorem 4), which only hold for $L^2$ normalization with uniform and exponentially spaced levels, respectively.

We now establish code-length bounds for both protocols, with proofs in Appendix D.2 and D.3:

**Theorem 4.3** (Code-length Bound for Protocol 1). *Let $p_j^m$ denote the probability of occurrence of $\ell_j^m$ for $m \in [M]$ and $j \in [\alpha_m]$. Under the setting specified in Theorem 4.1, the expectation $\mathbb{E}_w \mathbb{E}_{\boldsymbol{q}_{\mathbb{L}^M}} \left[ \mathrm{ENC}\left( Q_{\mathbb{L}^M}(g(\boldsymbol{x};\omega)); \mathbb{L}^M \right) \right]$ of the number of bits under Protocol 1 is bounded by*

$$\mathbb{E}_\omega \mathbb{E}_{\boldsymbol{q}_{\mathbb{L}^M}} \left[ \mathrm{ENC}\left( Q_{\mathbb{L}^M}(g(\boldsymbol{x};\omega)); \mathbb{L}^M \right) \right] = \mathcal{O}\left( \left( -\sum_{m=1}^M p_0^m - \sum_{m=1}^M \sum_{j=1}^{\alpha_m} p_j^m \log p_j^m \right) d \right). \tag{8}$$

**Theorem 4.4** (Code-length Bound for Protocol 2). *Let $\hat{p}_j^m$ denote the probability of occurrence of $\ell_j^m$ for $m \in [M]$ and $j \in [\alpha_m]$. Under the setting specified in Theorem 4.1, the expectation $\mathbb{E}_w \mathbb{E}_{\boldsymbol{q}_{\mathbb{L}^M}} \left[ \mathrm{ENC}\left( Q_{\mathbb{L}^M}(g(\boldsymbol{x};\omega)); \mathbb{L}^M \right) \right]$ of the number of bits under Protocol 2 is bounded by*

$$\mathbb{E}_w \mathbb{E}_{\boldsymbol{q}_{\mathbb{L}^M}} \left[ \mathrm{ENC}\left( Q_{\mathbb{L}^M}(g(\boldsymbol{x};\omega)); \mathbb{L}^M \right) \right] = \mathcal{O}\left( \left( -\sum_{m=1}^M \hat{p}_0^m - \sum_{m=1}^M \sum_{j=1}^{\alpha_m} \hat{p}_j^m \log \hat{p}_j^m \right) q_m d \right), \tag{9}$$

*where $q_m$ is the proportion of type $m$ coordinates across all coordinates.*

*Remark* 4.5. For the special case of $M = 1$, our bound for Protocol 1 in Theorem 4.3 recovers (Ramezani-Kebrya et al., 2023, Theorem 2). Moreover, under $M = 1$, $L^2$ normalization and $s = \sqrt{d}$ as in (Alistarh et al., 2017, Theorem 3.4), our bound (8) for Protocol 1 can be arbitrarily smaller than (Alistarh et al., 2017, Theorem 3.4) and (Ramezani-Kebrya et al., 2021, Theorem 5) depending on the probabilities $\{p_0, \ldots, p_{s+1}\}$.

### 4.2 ALGORITHM COMPLEXITY

Now, we will outline the guarantees for Algorithm 1. Here, Algorithm 1 is executed for $T$ iterations on $K$ nodes with learning rates in (6). Quantization sequence $\boldsymbol{\ell}^m$ is updated $J^m$ times where $\ell_j^m$ is used for $T_{m,j}$ iterations where $\sum_{m=1}^M \sum_{j=1}^{J^m} T_{m,j} = T$. In particular, $\ell_j^m$ has variance bound $\varepsilon_{Q,m,j}$ (7) and code-length bounds $N_{Q,m,j}$ in (8) and (9) under Protocol 1 and 2, respectively. Denote the average variance upper bound $\overline{\varepsilon_Q} = \sum_{m=1}^M \sum_{j=1}^{J^m} T_{m,j} \varepsilon_{Q,m,j}/T$ and the average expected

code-length bound $\overline{N_Q} = \sum_{m=1}^{M} \sum_{j=1}^{J^m} T_{m,j} N_{Q,m,j}/T$. Denote the average square root variance bound $\widehat{\varepsilon_Q} = \sum_{m=1}^{M} \sum_{j=1}^{J^m} T_{m,j} \sqrt{\varepsilon_{Q,m,j}}/T$. Denote $\sum_{t=1}^{T} X_{t+1/2}/T = \overline{X}_{t+1/2}$.

Algorithm 1 requires each node to send in expectation at most $\overline{N_Q}$ communication bits per iteration. Under the absolute noise model, we can bound GAP of Algorithm 1 with the proof in Appendix E.2:

**Theorem 4.6** (Algorithm 1 under Absolute Noise). *Suppose the iterates $X_t$ of Algorithm 1 are updated with learning rate schedule given in (6) for all $t = 1/2, 1, \ldots, T$. Let $\mathcal{X} \subset \mathbb{R}^d$ be a compact neighborhood of a VI solution and $D^2 := \sup_{\boldsymbol{p} \in \mathcal{X}} \|X_1 - \boldsymbol{p}\|_2^2$. Under Assumptions 2.1, 2.2, 2.3, and 2.4, we have*

$$\mathbb{E}\left[\mathrm{Gap}_{\mathcal{X}}\left(\overline{X}_{t+1/2}\right)\right] = \mathcal{O}\left(\left((LD + \|A(X_1)\|_2 + \sigma)\widehat{\varepsilon_Q} + \sigma\right) D^2 L^2 / \sqrt{TK}\right).$$

Now *only* for the relative noise profile, we introduce a mild regularity condition of co-coercivity, similar to QGen-X (Ramezani-Kebrya et al., 2023) to *obtain the fast rate $\mathcal{O}(1/T)$* [2]:

**Assumption 4.7** (Co-coercivity). For $\beta > 0$, we say operator $A$ is $\beta$-cocoercive when

$$\langle A(\boldsymbol{x}) - A(\boldsymbol{y}), \boldsymbol{x} - \boldsymbol{y}\rangle \geq \beta\|A(\boldsymbol{x}) - A(\boldsymbol{y})\|_*^2 \quad \forall \boldsymbol{x}, \boldsymbol{y} \in \mathbb{R}^d.$$

Further details about this assumption is in Appendix B.2. With this assumption, we obtain the following faster convergence guarantee for Algorithm 1 under relative noise:

**Theorem 4.8** (Algorithm 1 under Relative Noise). *Suppose the iterates $X_t$ of Algorithm 1 are updated with learning rate schedule in (6) for all $t = 1/2, 1, \ldots, T$. Let $\mathcal{X} \subset \mathbb{R}^d$ be a compact neighborhood of a VI solution. Let $D^2 := \sup_{\boldsymbol{p} \in \mathcal{X}} \|X_1 - \boldsymbol{p}\|_2^2$. Under Assumptions 2.1, 2.2, 2.3, 2.5, and 4.7, we have*

$$\mathbb{E}\left[\mathrm{Gap}_{\mathcal{X}}\left(\overline{X}_{t+1/2}\right)\right] = \mathcal{O}\left((\sigma_R \overline{\varepsilon_Q} + \overline{\varepsilon_Q} + \sigma_R)D^2/(TK)\right).$$

The proof details are included in Appendix E.3.

*Remark* 4.9. Both theorems show that increasing the number of processors $K$ lead to faster convergence for monotone VIs, matching the asymptotic rates for $T$ and $K$ in (Ramezani-Kebrya et al., 2023) which requires an extra almost sure boundedness assumption. Under the absolute noise model and by setting the number of gradients per round to one, our results match the known lower bound for convex and smooth optimization $\Omega(1/\sqrt{TK})$ (Woodworth et al., 2021, Theorem 1).[3] Previously, (Ramezani-Kebrya et al., 2023, Theorem 3) matches this lower bound but with an *extra assumption* that the operator is almost sure bounded.

## 5 ALMOST SURE BOUNDEDNESS MODEL

We proposed Algorithm 1 and proved its guarantees for the general class of monotone $L$-Lipschitz VIs. However, in practice, relevant VI works (Bach & Levy, 2019; Hsieh et al., 2021; Antonakopoulos et al., 2021) including the baseline Q-GenX (Ramezani-Kebrya et al., 2023) have an extra assumption of (almost sure) boundedness of the stochastic dual vector and previous global quantization works Alistarh et al. (2017); Ramezani-Kebrya et al. (2021); Faghri et al. (2020) has a similar assumption of a second-moment upper bound of the stochastic gradient (stochastic dual vector in our setting).

**Assumption 5.1** (Almost Sure Boundedness). There exists $J > 0$ s.t. $\|g(\boldsymbol{x}; \omega)\|_* \leq J$ almost surely.

Under this setting, the proposed learning rate (6) and its theoretical guarantees in Section 4.2 certainly still hold. We can obtain the similar rate $\mathcal{O}(1/T)$ to Theorem 4.8 for the relative noise case **without the co-coercivity Assumption** 4.7 with alternative adaptive learning rates and $\hat{q} \in (0, 1/4]$:

$$\gamma_t = \left(1 + \sum_{s=1}^{t-2} \sum_{k=1}^{K} \left\|\hat{V}_{k,s+1/2}\right\|_*^2 / K^2\right)^{\hat{q}-\frac{1}{2}}, \eta_t = \left(1 + \sum_{s=1}^{t-2} \sum_{k=1}^{K} \left\|\hat{V}_{k,s+1/2}\right\|_*^2 / K^2 + \|X_s - X_{s+1}\|_2^2\right)^{-\frac{1}{2}}. \text{(Alt)}$$

The details for this alternative (Alt) learning rates is included in Appendix F.2. Two learning rates allow a larger extrapolation step in the first line of (5), so the noise is an order of magnitude smaller than the expected variation of utilities (Hsieh et al., 2022). We now provide the convergence of Algorithm 1 under relative noise with learning rates Alt and without the co-coercivity assumption.

---

[2]Our guarantees for quantization, coding procedures and convergence under absolute noise do not require co-coercivity. This assumption is only needed to establish the fast rate $\mathcal{O}(1/T)$ under relative noise.

[3]In (Woodworth et al., 2021) their function $F$ is $L$-smooth implies that the $\nabla F$, or the operator in our case, is $L$-Lipschitz.

**Theorem 5.2** (Algorithm 1 under Relative Noise **without Co-coercivity Assumption**). *Suppose the iterates $X_t$ of Algorithm 1 are updated with learning rate schedule in (Alt) for all $t = 1/2, 1, \ldots, T$. Let $\mathcal{X} \subset \mathbb{R}^d$ be a compact neighborhood of a solution for (VI), $\overline{\varepsilon_Q}$ as in Section 4.2 and $D^2 := \sup_{\boldsymbol{p} \in \mathcal{X}} \|X_1 - \boldsymbol{p}\|_2^2$. Under Assumptions 2.1, 2.2, 2.3, 2.5, and 5.1, for Algorithm 1 with learning rates (Alt), we have*

$$\mathbb{E}\left[\mathrm{Gap}_{\mathcal{X}}\left(\overline{X}_{t+1/2}\right)\right] = \mathcal{O}\left((\sigma_R\overline{\varepsilon_Q} + \overline{\varepsilon_Q} + \sigma_R)D^4/T\right).$$

The proof is in Appendix F.5. Here, under the same assumptions as the baseline Ramezani-Kebrya et al. (2023), we can obtain the similar rate $\mathcal{O}(1/T)$ under relative noise *without the co-coercivity assumption*. To underscore the significance of eliminating the co-coercivity assumption, we note that the important class of *bilinear games*, for instance, are not co-coercive. Furthermore, we also include the guarantees for absolute noise for this model in Appendix F.15, where we also obtain the rate $\mathcal{O}(1/\sqrt{T})$ as Theorem 4.6.

## 6 NUMERICAL EXPERIMENTS

To further validate our theoretical findings, we have implemented QODA in Algorithm 1 based on the codebase of (Gidel et al., 2018) and train WGAN (Arjovsky et al., 2017) on CIFAR10 and CIFAR100 (Krizhevsky, 2009). To support efficient compression, we use the `torch_cgx` Pytorch extension (Markov et al., 2022). Moreover, we adapt compression choices layer-wise, following the L-GreCo (Markov et al., 2024) algorithm. Specifically, L-GreCo periodically collects gradients statistics, then executes a dynamic programming algorithm optimizing the total compression ratio while minimizing compression error.

In our experiments, we use 4 to 16 nodes, each with a single NVIDIA RTX 3090 GPU, in a multi-node Genesis Cloud environment with 5 Gbps inter-node bandwidth. For the communication backend, we pick the best option for quantized and full-precision regimes: OpenMPI (ope, 2023) and NCCL (ncc, 2023), respectively. The maximum bandwidth between nodes is estimated to be around 5 Gbit/second.

We follow the training recipe of Q-GenX (Ramezani-Kebrya et al., 2023), where authors set large batch size (1024) and keep all other hyperparameters as in the original codebase of (Gidel et al., 2018). For global and layer-wise compression, we use 5 bits (with bucket size 128), and run the L-GreCo adaptive compression algorithm every 10K optimization steps for both the generator and discriminator models[4]. The convergence results over three random seeds are presented in Figure 1. The figure demonstrates that the adaptive QODA approach not only *recovers the baseline accuracy* but also *improves convergence relative* to Q-GenX.

In order to illustrate the impact of QODA on the wall-clock training time, we have benchmarked the training in three different communication setups. The first is the original 5 Gbps bandwidth, whereas the second and the third reduce this to half and 1/5 of this maximum bandwidth. We measured the time per training step for uncompressed and QODA 5-bit training. Note that time per step is similar for for both data sets. Table 1 shows that layer-wise quantization achieves up to a 47% improvement in terms of end-to-end training time.

| Mode | 1 Gbps | 2.5 Gbps | 5 Gbps |
|---|---|---|---|
| Baseline | 291 | 265 | 251 |
| QODA5 | 197 | 195 | 195 |
| Speedup | 1.47× | 1.36× | 1.28× |

**Table 1:** Time per optimization step[5] (in ms) for baseline and QODA5 with different inter-node bandwidths.

---

[4] For the sake of a fair comparison to QGen-X, we did not include any additional encoding on top of quantization just as QGen-X did not.

[5] The optimization step includes forward and backward times. More precisely, the backward step consists of backpropagation, compression, communication and de-compression.

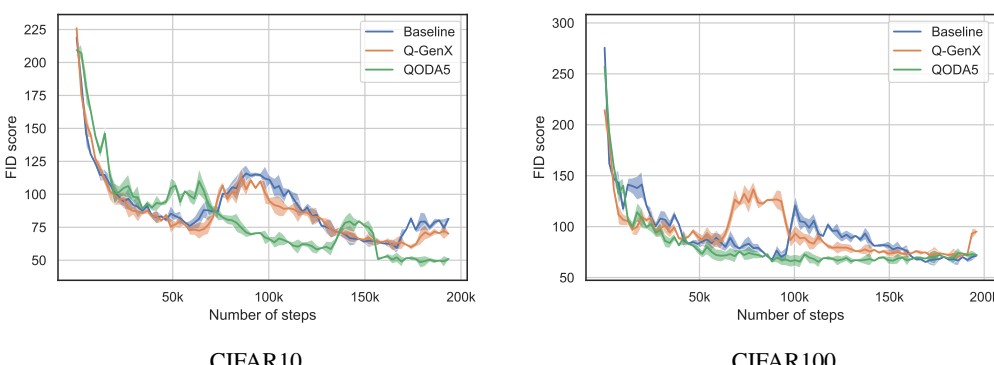

CIFAR10                                  CIFAR100

**Figure 1:** FID evolution during training. We compare basic Adam optimization against QODA-based extension of Adam with global (Q-GenX (Ramezani-Kebrya et al., 2023)) and layer-wise (L-GreCo) quantizations.

| Mode | 4 GPUs | 8 GPUs | 12 GPUs | 16 GPUs |
|---|---|---|---|---|
| baseline | 251 | 303 | 318 | 285 |
| QODA5 | 195 | 165 | 127 | 115 |
| Speedup | $1.28\times$ | $1.83\times$ | $2.50\times$ | $2.47\times$ |

**Table 2:** Time per optimization step (in ms) for baseline and QODA5 with different node counts.

Table 2 demonstrates the scalability of QODA up to 16 GPUs under weak scaling, i.e. with a constant global batch size. We observe a significant up to a $150\%$ speedup in comparison to the uncompressed baseline. Moreover, baseline step time degradation makes the scaling useless, whereas QODA allows to avoid such degradation.

## 7 LIMITATIONS AND FUTURE DIRECTIONS

While monotone VIs can cover a wide range of ML applications as stated in our introduction, there are situations that general non-monotone or (weak) minty VIs are required (Iusem et al., 2017; Kannan & Shanbhag, 2019; Beznosikov et al., 2022). Hence, for future directions, one may look into communication-efficient schemes to solve non-monotone VIs with an adaptive layer-wise compression. Furthermore, since our work is already lengthy and proposes theoretical novelties, it limits our ability to include many numerical applications without making the paper overly convoluted. Several applications of layer-wise quantization, such as training large-scale transformers, have been explored in Markov et al. (2022; 2024). Given our established theoretical results for communication-efficient QODA, in the future, it is therefore interesting to consider applications beyond GANs such as accelerating adversarial training in multi-GPU settings.

## 8 CONCLUSION

In brief, we introduce *optimism* in distributed VI with adaptive learning rates, develop layer-wise quantization with joint convergence and communication guarantees, and show improvements in end-to-end training time in a practical multi-node WGAN setting. We establish tight variance and code-length bounds for a general layer-wise and adaptive family of compression schemes that generalize previous bounds for global quantization. Furthermore, we provide convergence guarantees for QODA and achieve the fast rates $\mathcal{O}(1/\sqrt{T})$ and $\mathcal{O}(1/T)$ without the restrictive almost sure boundedness assumption on the operator under absolute and relative noise, respectively.

## 9 ETHICAL STATEMENT

Our main contributions are mainly theoretical in nature while we do offer the first truly multi-GPU communication-efficient setup for GAN training with VI solvers in Section 6. Hence, we believe this work do not pose any direct ethical concerns.

## 10 REPRODUCIBILITY STATEMENT

We discuss the details of our experiments in Section 6, and we also include all the code implementation in the supplementary material. We will release the code publicly along with the camera-ready version.

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

CONTENTS

# A    ADDITION INFORMATION

## A.1    FURTHER LITERATURE REVIEW

**Unbiased quantization** provides communication efficiency on the fly for empirical risk minimization, i.e., quantized variants of SGD converge under the same hyperparameters tuned for uncompressed variants while providing substantial savings in terms of communication costs (Alistarh et al., 2017; Wen et al., 2017; Zhang et al., 2017; Faghri et al., 2020; Ramezani-Kebrya et al., 2021; Markov et al., 2024; 2022). (Davies et al., 2021) has proposed lattice-based quantization for distributed mean estimation problem.

Beyond distributed VI settings, the Extra-gradient and their optimistic variants have a long history in the field of optimization. The Extra-gradient, first introduced by (Korpelevich, 1976), is known to achieve an optimal rate of order $\mathcal{O}(1/T)$ in monotone VIs. This method has been further extended in Nemirovski (2004); Nesterov (2007) by introducing Mirror-prox and its primal-dual counterpart Dual-extrapolation. However, all these methods require two oracle calls per iteration (one for the extrapolation and one for the update step) which makes them more expensive than the standard Forward/Backward methods. The first issue to address this issue was Popov's modified Arrow–Hurwicz algorithm Popov (1980). To that end, several extensions have been proposed such as Past Extra-gradient (PEG) of (Chiang et al., 2012; Gidel et al., 2019), Reflected Gradient (RG) of (Chambolle & Pock, 2011; Malitsky, 2015; Cui & Shanbhag, 2016), Optimistic Gradient (OG) of (Daskalakis et al., 2018; Mokhtari et al., 2019b;a; Peng et al., 2019) and Golden Ratio method of (Malitsky, 2019).

## A.2    COMPARISONS TO RELATED METHODS

**Improvement over Q-GenX Ramezani-Kebrya et al. (2023) (Optimism, Relaxed Assumptions, and Layer-wise Compression)**: Our proposed algorithm QODA (Algorithm 1) essentially consists of a distributed VI solver - Optimistic Dual Averaging (Section 3.3) - and a layer-wise compression general framework (Section 3.1). Firstly, our optimistic dual averaging distributed update step 5 reduces one extra gradient step compared to the extra-gradient approach of Q-GenX, hence reducing the communication burden by half. In addition, our algorithm QODA also requires fewer assumptions than Q-GenX Remark 2.6. We also improve the convergence relative to Q-GenX in training WGAN (Figure 1).

Furthermore, our layer-wise compression framework is much more general and is always better than the global compression framework in Q-GenX (Remark 3.2). Under the special case of $M = 1$ with only one type of layer, we recover the Q-GenX global compression. The compression framework also comes with two fine-grained coding protocols, among which our Protocol 1 is a generalization of Q-GenX coding protocol while our coding Protocol 2 is novel.

**Rigours Formulations and Tight Guarantees for Layer-wise Compression such as L-Greco Markov et al. (2024)**: We provide a novel and general theoretical formulation and establish guarantees for adaptive layer-wise quantization with tailored coding schemes, which is *not studied* in L-Greco. Layer-wise quantization schemes such as L-Greco have only been studied empirically without strong theoretical guarantees to handle the statistical heterogeneity across layers and over the course of training. Our tight variance and code-length bounds actually hold for any general layer-wise and unbiased quantization scheme. That is, we believe our framework is general enough to cover other layer-wise compression methods other than L-Greco such as Cgx (Markov et al., 2022).

All in all, a combination of SoTAs QGen-X + L-Greco does not represent our novel and general layer-wise framework with the corresponding theoretical guarantees and an associated fine-grained coding analysis while performing twice the number of gradient computations as we do.

**Comparison to Block Quantization**: Several works (Mishchenko et al., 2024; Horváth et al., 2023; Wang et al., 2022) study block quantization. We want to highlight that block (p-)quantization is *fundamentally different* from layer-wise quantization in our paper. As (Mishchenko et al., 2024, Definition B.1) suggests, the various blocks here follow the *same* scheme that is p-quantization

(Quant$_p$) which is explained in (Mishchenko et al., 2024, Definition 3.2). There are two fundamental differences compared to our layer-wise quantization:

- Each of our layer or block in this context has different adaptive sequences of levels (Section 3.1). This is why our method is named **"layer-wise."** Mishchenko et al. (2024)on the other hand applies the same p-quantization scheme Quant$_p$ to blocks with different sizes, implying that the nature and analysis of two methods are very different. Hence block quantization is not "layer-wise," and its analysis does not apply to the convergence of our methods.
- The way the quantization is calculated for each block or layer are different. Mishchenko et al. (2024) study and provide guarantees for the following type of p-quantization (for all blocks): $\widetilde{\Delta} = \|\Delta\|_p \operatorname{sign}(\Delta) \circ \xi$, where the $\xi$ are stacks of random Bernoulli variables. In our work, the sequence of levels for each layer is adaptively chosen according to the statistical heterogeneity over the course of training (refer to equation MQV).

Furthermore, the guarantee in (Mishchenko et al., 2024, Theorem 3.3) only cover p-quantization rather block p-quantization. In our Theorem 4.1, we provide the quantization variance bound for any **arbitrary** sequence of levels for each layer in contrast with only levels only based on p-quantization.

In brief, the block quantization is similar to bucketing in unbiased global quantizaiton (QSGD (Alistarh et al., 2017), NUQSGD (Ramezani-Kebrya et al., 2021)), which takes into account only the size of different blocks (sub-vectors), while for layer-wise quantization we take into account the statistical heterogeneity and impact of different layers on the final accuracy. Due to fundamental differences, our variance and code-length bounds require substantially more involved and different analyses that are not possible by simple extensions of block quantization in those works.

### A.3 NOTATIONS

We use lower-case bold letters to denote vectors. $\mathbb{E}[\cdot]$ denotes the expectation operator. $\|\cdot\|_0$ and $\|\cdot\|_*$ are number of nonzero elements of a vector and dual norm, respectively. $|\cdot|$ denotes the length of a binary string, the length of a vector, and cardinality of a set. Sets are typeset in a calligraphic font. The base-2 logarithm is denoted by $\log$, and the set of binary strings is denoted by $\{0, 1\}^*$. For any integer $n$, we use $[n]$ to denote the set $\{1, \ldots, n\}$. $\mathbb{1}$ denotes the indicator function.

## B VARIATIONAL INEQUALITY BACKGROUND

### B.1 GAP

Several properties of (GAP) have been explored in the literature (Nesterov, 2009; Antonakopoulos et al., 2019). In particular, the following classical result characterizes the solutions of (VI) via zeros of (GAP).

**Proposition B.1.** *(Nesterov, 2009) Let $\mathcal{X} \subseteq \mathbb{R}^d$ be a non-empty and convex set. Then, we have*

- *$GAP_{\mathcal{X}}(\hat{\boldsymbol{x}}) \geq 0$ for all $\hat{\boldsymbol{x}} \in \mathcal{X}$;*

- *If $GAP_{\mathcal{X}}(\hat{\boldsymbol{x}}) = 0$ and $\mathcal{X}$ contains a neighbourhood of $\hat{\boldsymbol{x}}$, then $\hat{\boldsymbol{x}}$ is a solution of (VI).*

### B.2 CO-COERCIVITY ASSUMPTION

We recall the co-coercivity assumption is as follows

**Assumption B.2** (Co-coercivity (Bauschke & Combettes, 2017))**.** For $\beta > 0$, we say operator $A$ is $\beta$-cocoercive when

$$\langle A(\boldsymbol{x}) - A(\boldsymbol{y}), \boldsymbol{x} - \boldsymbol{y} \rangle \geq \beta \|A(\boldsymbol{x}) - A(\boldsymbol{y})\|_*^2 \quad \forall \boldsymbol{x}, \boldsymbol{y} \in \mathbb{R}^d.$$

Note that by Cauchy-Schwarz, we further deduce for a co-coercive operator

$$\|A(\boldsymbol{x}) - A(\boldsymbol{y})\|_2 \|\boldsymbol{x} - \boldsymbol{y}\|_2 \geq \beta \|A(\boldsymbol{x}) - A(\boldsymbol{y})\|_2^2,$$

implying

$$\|\boldsymbol{x} - \boldsymbol{y}\|_2^2 \geq \beta^2 \|A(\boldsymbol{x}) - A(\boldsymbol{y})\|_2^2.$$

We refer the readers to (Bauschke & Combettes, 2017, Section 4.2) for further properties of co-coercive operators.

## B.3 RELATIVE NOISE EXAMPLES

Here we provide two examples in practice where the noise profile can be characterized as relative noise:

- Random coordinate descent (RCD): At iteration $t$, the RCD algorithm for a smooth convex function $f$ over $\mathbb{R}^d$ draws one coordinate $i_t \in [d]$ uniformly random and computes the partial derivative $v_{i,t} = \partial f / \partial x_{i_t}$. The $i$-th derivative is updated as $X_{i,t+1} = X_{i,t} - d \cdot \alpha \cdot v_{i,t}$ for step-size $\alpha > 0$. This update rule can also be written as $\mathbf{x}^+ = \mathbf{x} - \alpha g(\mathbf{x}; \mu)$ where $g_i(\mathbf{x}; \mu) = d \cdot \partial f / \partial x_i \cdot \mu$ and $\mu$ is drawn uniformly at random from the set of $\mathbb{R}^d$ basis vectors $\{\mathbf{e}_1, \ldots, \mathbf{e}_d\}$. Since $\partial f / \partial x_i = 0$ at the minima of $f$, we also have $g(\mathbf{x}^*; \mu) = 0$ if $\mathbf{x}^*$ is a minimizer of $f$, i.e., the variance of the random vector $g(\mathbf{x}; \mu)$ vanishes at the minima of $f$.
- Random player updating: Given an $N$-player convex game with loss functions $f_i, i \in [N]$. Suppose, at each stage, player $i$ is selected with probability $p_i$ to play an action following its individual gradient descent rule $X_{i,t+1} = X_{i,t} + \gamma_t / p_i V_{i,t}$ where $V_{i,t} = \nabla_i f_i(X_t)$ denotes player $i$ 's individual gradient at the state $X_t = (X_{1,t}, \ldots, X_{N,t})$ and $p_i$ is included for scaling reasons. One can show that all individual components of $A$ vanish at the game's Nash equilibria.

## C PROOF OF QUANTIZATION VARIANCE BOUND

**Theorem 4.1** (Quantization Variance Bound). *Let $\boldsymbol{v} \in \mathbb{R}^d$ be a vector to be quantized with $L^q$ normalization. With unbiased quantization of $\boldsymbol{v}$, i.e., $\mathbb{E}_{q_{\mathbb{L}^M}}[Q_{\mathbb{L}^M}(\boldsymbol{v})] = \boldsymbol{v}$, we have that*

$$\mathbb{E}_{q_{\mathbb{L}^M}}\left[\|Q_{\mathbb{L}^M}(\boldsymbol{v}) - \boldsymbol{v}\|_2^2\right] \leq \varepsilon_Q \|\boldsymbol{v}\|_2^2, \tag{7}$$

*where $\varepsilon_Q = \frac{(\bar{\ell}^M - 1)^2}{4\bar{\ell}^M} + \frac{(\bar{\ell}_1^M)^2 d^{2/\min\{q,2\}} \mathbb{1}\{d < d_{th}\}}{4} + (\bar{\ell}_1^M d^{2/\min\{2,q\}} - 1) d^{2/\min\{q,2\}} \mathbb{1}\{d \geq d_{th}\}.$*

*Proof.* First let us remind ourselves of the notations in the main paper. Fix a time $t$. Let the normalized coordinates be $\boldsymbol{u}$. Let $\bar{\ell}^m = \max_{0 \leq j \leq \alpha_m} \ell_{j+1}^m / \ell_j^m$, and $\bar{\ell}^M = \max_{1 \leq m \leq M} \bar{\ell}^M$. Denote the largest level 1 among the M sequences $\bar{\ell}_1^M = \max_{1 \leq m \leq M} \ell_1^M$. Also let $d_{th} = (2/\bar{\ell}_1^M)^{\min\{2,q\}}$. Let $\mathcal{B}_j^m := [\ell_j^m, \ell_{j+1}^m]$ for $m \in [M], j \in [\alpha_m]$.

Now, we can rewrite the equation (Var) for a fixed time $t$ as follows

$$\mathbb{E}_{q_{\mathbb{L}^M}}\left[\|Q_{\mathbb{L}^M}(\boldsymbol{v}) - \boldsymbol{v}\|_2^2\right] = \|\boldsymbol{v}\|_q^2 \sum_{m=1}^{M} \sum_{u_i \in \mathbb{S}^m} \sigma_Q^2(u_i; \boldsymbol{\ell}^m)$$

$$= \|\boldsymbol{v}\|_q^2 \sum_{m=1}^{M} \sum_{u_i \in \mathbb{S}^m} (\ell_{\tau^m(u_i)+1}^m - u_i)(u_i - \ell_{\tau^m(u_i)}^m)$$

$$= \|\boldsymbol{v}\|_q^2 \sum_{m=1}^{M} \left( \sum_{u_i \in \mathcal{B}_0^m} (\ell_1^m - u_i)u_i + \sum_{j=1}^{\alpha_m} \sum_{u_i \in \mathcal{B}_j^m} (\ell_{j+1}^m - u_i)(u_i - \ell_j^m) \right).$$

We now find the minimum $k_j^m$, satisfying $(\ell_{j+1}^m - u_i)(u_i - \ell_j^m) \le k_j^m u_i^2$ for $u_i \in \mathcal{B}_j^m$ for $m \in [M]$, $j \in [\alpha_m]$. Let $u_i = \ell_j^m \theta$ for $1 \le \theta \le \ell_{j+1}^m / \ell_j^m$. Then, we have

$$
\begin{aligned}
k_j^m &= \max_{1 \le \theta \le \ell_{j+1}^m/\ell_j^m} \frac{(\ell_{j+1}^m - u_i)(u_i - \ell_j^m)}{(\ell_j^m \theta)^2} \\
&= \max_{1 \le \theta \le \ell_{j+1}^m/\ell_j^m} \frac{(\ell_{j+1}^m/\ell_j^m - \theta)(\theta - 1)}{\theta^2} \\
&= \frac{(\ell_{j+1}^m/\ell_j^m - 1)^2}{4(\ell_{j+1}^m/\ell_j^m)},
\end{aligned}
$$

where the last equality follows from a simple differentiation with respect to $\theta$. Since the function $(x-1)^2/(4x)$ is monotonically increasing function for $x > 1$, we obtain

$$
\frac{(\ell_{j+1}^m/\ell_j^m - 1)^2}{4(\ell_{j+1}^m/\ell_j^m)} \le \frac{(\bar{\ell}^M - 1)^2}{4\bar{\ell}^M},
$$

which leads to

$$
\begin{aligned}
\sum_{j=1}^{\alpha_m} \sum_{u_i \in \mathcal{B}_j^m} (\ell_{j+1}^m - u_i)(u_i - \ell_j^m) &\le \sum_{j=1}^{\alpha_m} \sum_{u_i \in \mathcal{B}_j^m} k_j^m u_i^2 \\
&= \sum_{j=1}^{\alpha_m} \sum_{u_i \in \mathcal{B}^m} \frac{(\ell_{j+1}^m/\ell_j^m - 1)^2}{4(\ell_{j+1}^m/\ell_j^m)} u_i^2 \\
&\le \sum_{j=1}^{\alpha_m} \sum_{u_i \in \mathcal{B}^m} \frac{(\bar{\ell}^M - 1)^2}{4\bar{\ell}^M} u_i^2 \\
&= \frac{(\bar{\ell}^M - 1)^2}{4\bar{\ell}^M} \sum_{u_i \in \mathbb{S}^m/\mathcal{B}_0^m} u_i^2,
\end{aligned}
$$

yielding

$$
\begin{aligned}
\|\boldsymbol{v}\|_q^2 \sum_{m=1}^M \sum_{j=1}^{\alpha_m} \sum_{u_i \in \mathcal{B}_j^m} (\ell_{j+1}^m - u_i)(u_i - \ell_j^m) &\le \|\boldsymbol{v}\|_q^2 \sum_{m=1}^M \frac{(\bar{\ell}^M - 1)^2}{4\bar{\ell}^M} \sum_{u_i \in \mathbb{S}^m/\mathcal{B}_0^m} u_i^2 \\
&= \|\boldsymbol{v}\|_q^2 \frac{(\bar{\ell}^M - 1)^2}{4\bar{\ell}^M} \sum_{m=1}^M \sum_{u_i \in \mathbb{S}^m/\mathcal{B}_0^m} u_i^2 \\
&\le \|\boldsymbol{v}\|_q^2 \frac{(\bar{\ell}^M - 1)^2}{4\bar{\ell}^M} \frac{\|\boldsymbol{v}\|_2^2}{\|\boldsymbol{v}\|_q^2} \\
&= \frac{(\bar{\ell}^M - 1)^2}{4\bar{\ell}^M} \|\boldsymbol{v}\|_2^2.
\end{aligned}
$$

Next, we attempt to bound $\sum_{m=1}^M \sum_{u_i \in \mathcal{B}_0^m} (\ell_1^m - u_i) u_i$ with these two known lemmas

**Lemma C.1.** *Let $\boldsymbol{v} \in \mathbb{R}^d$. Then, for all $0 < p < q$, we have $\|\boldsymbol{v}\|_q \le \|\boldsymbol{v}\|_p \le d^{1/p - 1/q} \|\boldsymbol{v}\|_q$. This holds even when $q < 1$ and $\|\cdot\|$ is merely a seminorm.*

**Lemma C.2.** *(Ramezani-Kebrya et al., 2021, Lemma 15) Let $p \in (0, 1)$ and $u \in \mathcal{B}_0$. Then we have $u(\ell_1 - u) \le K_p \ell_1^{2-p} u^p$, where*

$$
K_p = \frac{1/p}{2/p - 1} \left( \frac{1/p - 1}{2/p - 1} \right)^{1-p}.
$$

Now, from these two lemma, for any $0 < p < 1$ and $q \leq 2$, we obtain that

$$\|\boldsymbol{v}\|_q^2 \sum_{m=1}^{M} \sum_{u_i \in \mathcal{B}_0^m} (\ell_1^m - u_i)u_i \leq \|\boldsymbol{v}\|_q^2 \sum_{m=1}^{M} \sum_{u_i \in \mathcal{B}_0^m} K_p(\ell_1^m)^{2-p} u_i^p$$

$$\leq \|\boldsymbol{v}\|_q^2 K_p(\bar{\ell}_1^M)^{2-p} \sum_{m=1}^{M} \sum_{u_i \in \mathcal{B}_0^m} u_i^p$$

$$= \|\boldsymbol{v}\|_q^2 K_p(\bar{\ell}_1^M)^{2-p} \sum_{m=1}^{M} \sum_{u_i \in \mathcal{B}_0^m} \frac{|v_i|^p}{\|\boldsymbol{v}\|_q^p}$$

$$\leq K_p(\bar{\ell}_1^M)^{2-p} \|\boldsymbol{v}\|_p^p \|\boldsymbol{v}\|_q^{2-p}$$

$$\leq K_p(\bar{\ell}_1^M)^{2-p} \|\boldsymbol{v}\|_2^p d^{1-p/2} \|\boldsymbol{v}\|_2^{2-p}$$

$$= K_p(\bar{\ell}_1^M)^{2-p} d^{1-p/2} \|\boldsymbol{v}\|_2^2,$$

where the penultimate inequality holds due to the first given lemma and $\|\boldsymbol{v}\|_q \leq \|\boldsymbol{v}\|_2$ for $q \geq 2$. Now combining the bounds, we obtain

$$\mathbb{E}_{q_{\mathbb{L}^M}}[\|Q_{\mathbb{L}^M}(\boldsymbol{v}) - \boldsymbol{v}\|_2^2] \leq \left( \frac{(\bar{\ell}^M - 1)^2}{4\bar{\ell}^M} + K_p(\bar{\ell}_1^M)^{2-p} d^{1-p/2} \right) \|\boldsymbol{v}\|_2^2.$$

Moreover, if $q \geq 1$, note that $\|\boldsymbol{v}\|_q^{2-p} \leq \|\boldsymbol{v}\|_2^{2-p} d^{\frac{2-p}{\min\{2,q\}} - \frac{2-p}{2}}$, yielding

$$\mathbb{E}_{q_{\mathbb{L}^M}}[\|Q_{\mathbb{L}^M}(\boldsymbol{v}) - \boldsymbol{v}\|_2^2] \leq \left( \frac{(\bar{\ell}^M - 1)^2}{4\bar{\ell}^M} + K_p(\bar{\ell}_1^M)^{2-p} d^{\frac{2-p}{\min\{2,q\}}} \right) \|\boldsymbol{v}\|_2^2.$$

Now we can minimize $\varepsilon_Q$ with finding the optimal $p^*$ by minimizing

$$\lambda(p) = \frac{1/p}{2/p - 1} \left( \frac{1/p - 1}{2/p - 1} \right)^{1-p} v^{1-p} = \frac{1}{2-p} \left( \frac{1-p}{2-p} \right)^{1-p} v^{1-p} = (2-p)^{p-2}(1-p)^{1-p} v^{1-p},$$

where $v = \bar{\ell}_1^M d^{\frac{1}{\min\{2,q\}}}$. This is equivalent to minimizing the log

$$\log \lambda(p) = (p-2)\log(2-p) + (1-p)\log(1-p) + (1-p)\log(v)$$

Setting the derivative of $\log \lambda(p)$ to zero, we have

$$-1 + \log(2 - p^*) + 1 - \log(1 - p^*) + \log(v) = 0,$$

yielding the optimal $p^*$ to be

$$p^* = \begin{cases} \dfrac{v-2}{v-1}, & v \geq 2 \quad \text{or} \quad d \geq d_{th} \\ 0, & v < 2 \quad \text{or} \quad d < d_{th}. \end{cases}$$

In brief, we have

$$\varepsilon_Q = \frac{(\bar{\ell}^M - 1)^2}{4\bar{\ell}^M} + (\bar{\ell}_1^M d^{\frac{2}{\min\{q,2\}}} - 1)d^{\frac{2}{\min\{q,2\}}} \mathbb{1}\{d \geq d_{th}\} + \frac{1}{4}(\bar{\ell}_1^M)^2 d^{\frac{2}{\min\{q,2\}}} \mathbb{1}\{d < d_{th}\}.$$

∎

# D  CODING FRAMEWORK

## D.1  FURTHER DETAILS ON CODING FRAMEWORK

The choice of a specific lossless prefix code for encoding $\boldsymbol{q}_{\mathbb{L}^t,M}$ relies on the extent to which the distribution of the discrete alphabet of levels is known. If we can estimate or know the distribution of the frequency of the discrete alphabet $\Omega^{t,M}$, we can apply the classical Huffman coding with an

efficient encoding/decoding scheme and achieve the minimum expected code-length among methods encoding symbols separately (Cover & Thomas, 2006; Huffman, 1952). On the other hand, if we only know smaller values are more frequent than larger values without knowing the distribution of the discrete alphabet, we can consider Elias recursive coding (ERC) (Elias, 1975).

The decoding DEC : $\{0,1\}^* \to \mathbb{R}^d$ first reads $C_q$ bits to reconstruct $\|\boldsymbol{v}\|_q$, then applies decoding scheme $\Psi^{-1} : \{0,1\}^* \to (\Omega^{t,M})^d$ to obtain normalized coordinates.

Given quantization levels $\boldsymbol{\ell}^{t,m}$ and the marginal PDF of normalized coordinates, $K$ nodes can construct the Huffman tree in parallel. A Huffman tree of a source with $s + 2$ symbols can be constructed in time $\mathcal{O}(s)$ through sorting the symbols by the associated probabilities. It is well-known that Huffman codes minimize the expected code-length:

**Theorem D.1.** *(Cover & Thomas, 2006, Theorems 5.4.1 and 5.8.1) Let $Z$ denote a random source with a discrete alphabet $\mathcal{Z}$. The expected code-length of an optimal prefix code to compress $Z$ is bounded by $H(Z) \leq \mathbb{E}[L] \leq H(Z) + 1$ where $H(Z) \leq \log_2(|\mathcal{Z}|)$ is the entropy of $Z$ in bits.*

### D.2 PROOF OF CODE LENGTH BOUND FOR PROTOCOL 1

**Theorem 4.3** (Code-length Bound for Protocol 1). *Let $p_j^m$ denote the probability of occurrence of $\ell_j^m$ for $m \in [M]$ and $j \in [\alpha_m]$. Under the setting specified in Theorem 4.1, the expectation $\mathbb{E}_w \mathbb{E}_{\boldsymbol{q}_{\mathbb{L}^M}} \left[ \mathrm{ENC}\left(Q_{\mathbb{L}^M}(g(\boldsymbol{x};\omega)); \mathbb{L}^M\right) \right]$ of the number of bits under Protocol 1 is bounded by*

$$\mathbb{E}_\omega \mathbb{E}_{\boldsymbol{q}_{\mathbb{L}^M}} \left[ \mathrm{ENC}\left(Q_{\mathbb{L}^M}(g(\boldsymbol{x};\omega)); \mathbb{L}^M\right) \right] = \mathcal{O}\left( \left( -\sum_{m=1}^{M} p_0^m - \sum_{m=1}^{M}\sum_{j=1}^{\alpha_m} p_j^m \log p_j^m \right) d \right). \tag{8}$$

*Proof.* Following the Protocol 1, we first use a constant $C_q$ bits to represent the positive scalar $\|\boldsymbol{v}\|_q$ with a standard 32-bit floating point encoding. Then we use 1 bit to encode the sign of each nonzero entry of $\boldsymbol{u}$. Next, the probabilities associated with the symbols to be encoded, i.e., the levels in $\Omega^M$, can be computed using the weighted sum of the conditional CDFs of normalized coordinates as follows.

**Proposition D.2.** *Let $j \in [\alpha_m]$, we have the probability $p_j^m$ of occurrence of $\ell_j^m$ is*

$$p_j^m = Pr(\ell_j^m) = \int_{\ell_{j-1}^m}^{\ell_j^m} \frac{u - \ell_{j-1}^m}{\ell_j^m - \ell_{j-1}^m} \, \mathrm{d}\tilde{F}(u) + \int_{\ell_j^m}^{\ell_{j+1}^m} \frac{\ell_{j+1}^m - u}{\ell_{j+1}^m - \ell_j^m} \, \mathrm{d}\tilde{F}(u),$$

*where $\tilde{F}(u)$ is the weighted sum of the conditional CDFs as defined in (2). Consequently we deduce*

$$p_0^m = Pr(\ell_0^m) = \int_{\ell_0^m}^{\ell_1^m} \frac{\ell_1^m - u}{\ell_1^m - \ell_0^m} \, \mathrm{d}\tilde{F}(u) = \int_0^{\ell_1^m} \frac{\ell_1^m - u}{\ell_1^m} \, \mathrm{d}\tilde{F}(u),$$

$$p_{\alpha_m+1}^m = Pr(\ell_{\alpha_m+1}^m) = \int_{\ell_{\alpha_m}^m}^{\ell_{\alpha_m+1}^m} \frac{u - \ell_{\alpha_m}^m}{\ell_{\alpha_m+1}^m - \ell_{\alpha_m}^m} \, \mathrm{d}\tilde{F}(u) = \int_{\ell_{\alpha_m}^m}^1 \frac{u - \ell_{\alpha_m}^m}{1 - \ell_{\alpha_m}^m} \, \mathrm{d}\tilde{F}(u).$$

Then, we can get the expected number of non-zeros after quantization.

**Lemma D.3.** *For arbitrary $\boldsymbol{v} \in \mathbb{R}^d$, the expected number of non-zeros in $Q_{\mathbb{L}}^M(\boldsymbol{v})$ is*

$$\mathbb{E}\left[ \|Q_{\mathbb{L}}^M(\boldsymbol{v})\|_0 \right] = \left( 1 - \sum_{m=1}^{M} p_0^m \right) d.$$

The optimal expected code-length for transmitting one random symbol is within one bit of the entropy of the source (Cover & Thomas, 2006). Hence, we can transmit entries of normalized $\boldsymbol{u}$ in at most $\left( \sum_{m=1}^{M} H(\boldsymbol{\ell}^m) + 1 \right) d$, where $H(\boldsymbol{\ell}^m) = -\sum_{j=1}^{\alpha_m} p_j^m \log(p_j^m)$ is the entropy in bits.

In brief, we obtain

$$\mathbb{E}_w \mathbb{E}_{\boldsymbol{q}_{\mathbb{L}^M}} \left[ \text{ENC} \left( Q_{\mathbb{L}^M}(g(\boldsymbol{x}; \omega)); \mathbb{L}^M \right) \right] = C_q + \left( 1 - \sum_{m=1}^{M} p_0^m \right) d + \left( \sum_{m=1}^{M} H(\boldsymbol{\ell}^m) + 1 \right) d.$$

∎

### D.3 PROOF OF CODE LENGTH BOUND FOR PROTOCOL 2

**Theorem 4.4** (Code-length Bound for Protocol 2). *Let $\hat{p}_j^m$ denote the probability of occurrence of $\ell_j^m$ for $m \in [M]$ and $j \in [\alpha_m]$. Under the setting specified in Theorem 4.1, the expectation $\mathbb{E}_w \mathbb{E}_{\boldsymbol{q}_{\mathbb{L}^M}} \left[ \text{ENC} \left( Q_{\mathbb{L}^M}(g(\boldsymbol{x}; \omega)); \mathbb{L}^M \right) \right]$ of the number of bits under Protocol 2 is bounded by*

$$\mathbb{E}_w \mathbb{E}_{\boldsymbol{q}_{\mathbb{L}^M}} \left[ \text{ENC} \left( Q_{\mathbb{L}^M}(g(\boldsymbol{x}; \omega)); \mathbb{L}^M \right) \right] = \mathcal{O} \left( \left( - \sum_{m=1}^{M} \hat{p}_0^m - \sum_{m=1}^{M} \sum_{j=1}^{\alpha_m} \hat{p}_j^m \log \hat{p}_j^m \right) q_m d \right), \quad (9)$$

*where $q_m$ is the proportion of type $m$ coordinates across all coordinates.*

*Proof.* Following the Protocol 2, we first use a constant $C_q$ bits to represent the positive scalar $\|\boldsymbol{v}\|_q$ with a standard 32-bit floating point encoding. We now carry out the encoding and decoding procedure in parallel for each of the M types of coordinates. We use 1 bit to encode the sign of each nonzero type-$m$ entry. Next, the probabilities associated with the symbols to be encoded, i.e., the type-$m$ levels, can be computed using the weighted sum of the conditional CDFs of normalized type-$m$ coordinates as follows.

**Proposition D.4.** *Let $j \in [\alpha_m]$, we have the probability $\hat{p}_j^m$ of occurrence of $\ell_j^m$ is*

$$\hat{p}_j^m = Pr(\ell_j^m) = \int_{\ell_{j-1}^m}^{\ell_j^m} \frac{u - \ell_{j-1}^m}{\ell_j^m - \ell_{j-1}^m} \, \mathrm{d}\tilde{F}^m(u) + \int_{\ell_j^m}^{\ell_{j+1}^m} \frac{\ell_{j+1}^m - u}{\ell_{j+1}^m - \ell_j^m} \, \mathrm{d}\tilde{F}^m(u),$$

*where $\tilde{F}^m(u)$ is the weighted sum of the type-$m$ conditional CDFs in (4). Hence we get*

$$\hat{p}_0^m = Pr(\ell_0^m) = \int_{\ell_0^m}^{\ell_1^m} \frac{\ell_1^m - u}{\ell_1^m - \ell_0^m} \, \mathrm{d}\tilde{F}^m(u) = \int_0^{\ell_1^m} \frac{\ell_1^m - u}{\ell_1^m} \, \mathrm{d}\tilde{F}^m(u),$$

$$\hat{p}_{\alpha_m+1}^m = Pr(\ell_{\alpha_m+1}^m) = \int_{\ell_{\alpha_m}^m}^{\ell_{\alpha_m+1}^m} \frac{u - \ell_{\alpha_m}^m}{\ell_{\alpha_m+1}^m - \ell_{\alpha_m}^m} \, \mathrm{d}\tilde{F}^m(u) = \int_{\ell_{\alpha_m}^m}^{1} \frac{u - \ell_{\alpha_m}^m}{1 - \ell_{\alpha_m}^m} \, \mathrm{d}\tilde{F}^m(u).$$

Then, we can get the expected number of non-zeros after quantization.

**Lemma D.5.** *For arbitrary $\boldsymbol{v} \in \mathbb{R}^d$, the expected number of non-zeros in $Q_{\mathbb{L}}^M(\boldsymbol{v})$ is*

$$\mathbb{E} \left[ \|Q_{\mathbb{L}}^M(\boldsymbol{v})\|_0 \right] = \sum_{m=1}^{M} \left( 1 - \hat{p}_0^m \right) q_m d.$$

The optimal expected code-length for transmitting one random symbol is within one bit of the entropy of the source (Cover & Thomas, 2006). Hence, we can transmit entries of normalized $\boldsymbol{u}$ in at most $\sum_{m=1}^{M} \left( H(\boldsymbol{\ell}^m) + 1 \right) q_m d$, where $q_m$ is the proportion of type-$m$ coordinates w.r.t all coordinates and $H(\boldsymbol{\ell}^m) = - \sum_{j=1}^{\alpha_m} \hat{p}_j^m \log(\hat{p}_j^m)$ is the entropy in bits.

In brief, we obtain

$$\mathbb{E}_w \mathbb{E}_{\boldsymbol{q}_{\mathbb{L}^M}} \left[ \mathrm{ENC} \left( Q_{\mathbb{L}^M}(g(\boldsymbol{x};\omega)); \mathbb{L}^M \right) \right]$$

$$= C_q + \sum_{m=1}^{M} (1 - \hat{p}_0^m) \, q_m d + \sum_{m=1}^{M} \left( - \sum_{j=1}^{\alpha_m} \left( \hat{p}_j^m \log(\hat{p}_j^m) \right) + 1 \right) q_m d$$

$$= \mathcal{O} \left( \left( - \sum_{m=1}^{M} \hat{p}_0^m - \sum_{m=1}^{M} \sum_{j=1}^{\alpha_m} \hat{p}_j^m \log \hat{p}_j^m \right) q_m d \right),$$

as desired. ∎

### D.4 Unbiased Compression under Both Noises Profiles

The following two lemmas show how additional noise due to compression affects the upper bounds under absolute noise Assumption 2.4 and relative noise models Assumption 2.5, respectively. Let's keep in mind that $\boldsymbol{q}_{\mathbb{L}^M} \sim \mathbb{P}_Q$ represent $d$ variables sampled independently for random quantization, and $\boldsymbol{q}_{\mathbb{L}^M}$ is independent of random sample $w \sim \mathbb{P}$.

**Lemma D.6** (Unbiased Compression under Absolute Noise). *Let $\boldsymbol{x} \in \mathcal{X}$ and $w \sim \mathbb{P}$. Suppose the oracle $g(\boldsymbol{x};\omega)$ satisfies Assumption 2.4. Suppose $Q_{\mathbb{L}^M}$ satisfies Theorem 4.1 and Theorem 4.3, then the compressed $Q_{\mathbb{L}^M}(g(\boldsymbol{x};\omega))$ satisfies Assumption 2.4 with*

$$\mathbb{E} \left[ \|Q_{\mathbb{L}^M}(g(\boldsymbol{x};\omega)) - A(\boldsymbol{x})\|_2^2 \right] \leq \varepsilon_Q (2L^2 D^2 + 2\|A(X_1)\|_2^2 + \sigma^2) + \sigma^2.$$

*Proof.* The unbiasedness property immediately follows from the construction of the unbiased quantization $Q_{\mathbb{L}^M}$. Next, we note that that the maximum norm increase when compressing $Q_{\mathbb{L}^M}(g(\boldsymbol{x};\omega))$ occurs when each normalized coordinate of $g(\boldsymbol{x};\omega)$, $\{u_i\}_{i \in [d]}$, is mapped to the upper level $\ell_{\tau^m(u_i)+1}^m$ for some $m \in [M]$. We can show bounded absolute variance as follows

$$\mathbb{E}_w \mathbb{E}_{\boldsymbol{q}_{\mathbb{L}^M}} \left[ \|Q_{\mathbb{L}^M}(g(\boldsymbol{x};\omega)) - A(x)\|_2^2 \right] = \mathbb{E}_w \mathbb{E}_{\boldsymbol{q}_{\mathbb{L}^M}} \left[ \|Q_{\mathbb{L}^M}(g(\boldsymbol{x};\omega)) - g(\boldsymbol{x};\omega) \right.$$
$$\left. + g(\boldsymbol{x};\omega) - A(x)\|_2^2 \right]$$
$$= \mathbb{E}_w \mathbb{E}_{\boldsymbol{q}_{\mathbb{L}^M}} \left[ \|Q_{\mathbb{L}^M}(g(\boldsymbol{x};\omega)) - g(\boldsymbol{x};\omega)\|_2^2 \right]$$
$$+ \mathbb{E}_w \left[ \|U(\boldsymbol{x};\omega)\|_2^2 \right]$$
$$\leq \varepsilon_Q \mathbb{E}_w \left[ \|g(\boldsymbol{x};\omega)\|_2^2 \right] + \sigma^2$$
$$= \varepsilon_Q \mathbb{E}_w \left[ \|A(\boldsymbol{x}) + U(\boldsymbol{x};\omega)\|_2^2 \right] + \sigma^2$$
$$= \varepsilon_Q \|A(\boldsymbol{x})\|_2^2 + \varepsilon_Q \mathbb{E}_w \left[ \|U(\boldsymbol{x};\omega)\|_2^2 \right] + \sigma^2$$
$$\leq \varepsilon_Q \|A(\boldsymbol{x})\|_2^2 + \varepsilon_Q \sigma^2 + \sigma^2,$$

where the second equality occurs due to unbiasedness of $\boldsymbol{q}_{\mathbb{L}^M}$, the third steps follos from Theorem 4.1, and the last inequality holds according to Assumption 2.4 for $g(\boldsymbol{x};\omega)$.

Now we note that in Theorem 4.6, $D^2 := \sup_{\boldsymbol{x} \in \mathcal{X}} \|X_1 - \boldsymbol{x}\|_2^2$, where $\mathcal{X} \subset \mathbb{R}^d$ is a compact neighborhood of a VI solution. Since $A$ is $L$-Lipschitz (Assumption 2.3), we note that

$$\|A(X_1) - A(\boldsymbol{x})\|_2^2 \leq L^2 \|X_1 - \boldsymbol{x}\|_2^2 \leq L^2 D^2 \quad \forall \, \boldsymbol{x} \in \mathcal{X}.$$

Since $X_1$ is our initialization, $A(X_1)$ has a finite value, so $A(\boldsymbol{x})$ is bounded for all $\boldsymbol{x} \in \mathcal{X}$. Hence for the quantization in Algorithm 1, we can obtain

$$\|A(\boldsymbol{x})\|_2^2 \leq 2\|A(X_1) - A(\boldsymbol{x})\|_2^2 + 2\|A(X_1)\|_2^2 \leq 2L^2 D^2 + 2\|A(X_1)\|_2^2,$$

which implies the desired conclusion. ∎

**Lemma D.7** (Unbiased Compression under Relative Noise). *Let $\boldsymbol{x} \in \mathcal{X}$ and $w \sim \mathbb{P}$. Suppose the oracle $g(\boldsymbol{x};\omega)$ satisfies Assumption 2.5. Suppose $Q_{\mathbb{L}^M}$ satisfies Theorem 4.1 and Theorem 4.4, then the compressed $Q_{\mathbb{L}^M}(g(\boldsymbol{x};\omega))$ satisfies Assumption 2.5 with*

$$\mathbb{E} \left[ \|Q_{\mathbb{L}^M}(g(\boldsymbol{x};\omega)) - A(\boldsymbol{x})\|_2^2 \right] \leq (\varepsilon_Q \sigma_R + \varepsilon_Q + \sigma_R) \|A(\boldsymbol{x})\|_2^2. \tag{10}$$

*Proof.* The unbiasedness assumption holds similar to D.6. We can show bounded absolute variance as follows

$$
\begin{aligned}
\mathbb{E}_w \mathbb{E}_{\boldsymbol{q}_{\mathbb{L}M}} \left[ \|Q_{\mathbb{L}M}(g(\boldsymbol{x};\omega)) - A(x)\|_2^2 \right] &= \mathbb{E}_w \mathbb{E}_{\boldsymbol{q}_{\mathbb{L}M}} \left[ \|Q_{\mathbb{L}M}(g(\boldsymbol{x};\omega)) - g(\boldsymbol{x};\omega) \right. \\
&\quad \left. + g(\boldsymbol{x};\omega) - A(x)\|_2^2 \right] \\
&= \mathbb{E}_w \mathbb{E}_{\boldsymbol{q}_{\mathbb{L}M}} \left[ \|Q_{\mathbb{L}M}(g(\boldsymbol{x};\omega)) - g(\boldsymbol{x};\omega)\|_2^2 \right] \\
&\quad + \mathbb{E}_w \left[ \|U(\boldsymbol{x};\omega)\|_2^2 \right] \\
&\leq \varepsilon_Q \mathbb{E}_w \left[ \|g(\boldsymbol{x};\omega)\|_2^2 \right] + \sigma_R \|A(\boldsymbol{x})\|_2^2 \\
&= \varepsilon_Q \mathbb{E}_w \left[ \|A(\boldsymbol{x}) + U(\boldsymbol{x};\omega)\|_2^2 \right] + \sigma_R \|A(\boldsymbol{x})\|_2^2 \\
&= \varepsilon_Q \|A(\boldsymbol{x})\|_2^2 + \varepsilon_Q \mathbb{E}_w \left[ \|U(\boldsymbol{x};\omega)\|_2^2 \right] + \sigma_R \|A(\boldsymbol{x})\|_2^2 \\
&\leq (\varepsilon_Q \sigma_R + \varepsilon_Q + \sigma_R) \|A(\boldsymbol{x})\|_2^2,
\end{aligned}
$$

where the second equality occurs due to the unbiasedness of $\boldsymbol{q}_{\mathbb{L}M}$, the fifth equality holds because of the unbiasedness of the noise model and the last inequality holds according to Assumption 2.5 for $g(\boldsymbol{x};\omega)$. ∎

# E ANALYSIS IN THE GENERAL SETTING

## E.1 TEMPLATE INEQUALITY

**Proposition E.1** (Template Inequality). *Suppose the iterates $X_t$ of (5) are updated with non-increasing step-size schedule $\gamma_t$ and $\eta_t$ as in (6) for all $t = 1/2, 1, \ldots$. Then for any $X \in \mathbb{R}^d$, we have*

$$
\begin{aligned}
&\sum_{t=1}^{T} \left\langle \frac{1}{K} \sum_{k=1}^{K} \hat{V}_{k,t+1/2}, X_{t+1/2} - X \right\rangle \\
&\leq \frac{\|X\|_*^2}{2\eta_{T+1}} + \sum_{t=1}^{T} \frac{\eta_t}{2K^2} \sum_{k=1}^{K} \left\| \hat{V}_{k,t+1/2} - \hat{V}_{k,t-1/2} \right\|_*^2 - \sum_{t=1}^{T} \frac{\|X_t - X_{t+1/2}\|_*^2}{2\eta_t}.
\end{aligned}
$$

*Proof.* First, decompose the LHS individual term $\frac{1}{K} \left\langle \sum_{k=1}^{K} \hat{V}_{k,t+1/2}, X_{t+1/2} - X \right\rangle$ into two terms as follows

$$
\frac{1}{K} \left\langle \sum_{k=1}^{K} \hat{V}_{k,t+1/2}, X_{t+1/2} - X \right\rangle = A + B,
$$

where

$$
A = \frac{1}{K} \left\langle \sum_{k=1}^{K} \hat{V}_{k,t+1/2}, X_{t+1/2} - X_{t+1} \right\rangle, \quad B = \frac{1}{K} \left\langle \sum_{k=1}^{K} \hat{V}_{k,t+1/2}, X_{t+1} - X \right\rangle.
$$

From the update rule of 5 (with $\eta_t$), note that

$$
\begin{aligned}
B &= \langle Y_t - Y_{t+1}, X_{t+1} - X \rangle \\
&= \left\langle Y_t - \frac{\eta_{t+1}}{\eta_t} Y_{t+1}, X_{t+1} - X \right\rangle + \left\langle \frac{\eta_{t+1}}{\eta_t} Y_{t+1} - Y_{t+1}, X_{t+1} - X \right\rangle \\
&= \frac{1}{\eta_t} \langle \eta_t Y_t - \eta_{t+1} Y_{t+1}, X_{t+1} - X \rangle + \left( \frac{1}{\eta_{t+1}} - \frac{1}{\eta_t} \right) \langle -\eta_{t+1} Y_{t+1}, X_{t+1} - X \rangle \\
&= \frac{1}{\eta_t} \langle X_t - X_{t+1}, X_{t+1} - X \rangle + \left( \frac{1}{\eta_{t+1}} - \frac{1}{\eta_t} \right) \langle X_1 - X_{t+1}, X_{t+1} - X \rangle \\
&= \frac{1}{2\eta_t} \left( \|X_t - X\|_*^2 - \|X_t - X_{t+1}\|_*^2 - \|X_{t+1} - X\|_*^2 \right) \\
&\quad + \left( \frac{1}{2\eta_{t+1}} - \frac{1}{2\eta_t} \right) \left( \|X_1 - X\|_*^2 - \|X_1 - X_{t+1}\|_*^2 - \|X_{t+1} - X\|_*^2 \right) \\
&\leq \frac{1}{2\eta_t} \|X_t - X\|_*^2 - \frac{1}{2\eta_t} \|X_t - X_{t+1}\|_*^2 \\
&\quad - \frac{1}{2\eta_{t+1}} \|X_{t+1} - X\|_*^2 + \left( \frac{1}{2\eta_{t+1}} - \frac{1}{2\eta_t} \right) \|X_1 - X\|_*^2,
\end{aligned}
$$

the last inequality holds as the non-positive term $-\left( \frac{1}{2\eta_{t+1}} - \frac{1}{2\eta_t} \right) \|X_1 - X_{t+1}\|_*^2$ is dropped. We can rearrange the above inequality as

$$
\begin{aligned}
\frac{1}{2\eta_{t+1}} \|X_{t+1} - X\|_*^2 &\leq \frac{1}{2\eta_t} \|X_t - X\|_*^2 - \frac{1}{2\eta_t} \|X_t - X_{t+1}\|_*^2 + \left( \frac{1}{2\eta_{t+1}} - \frac{1}{2\eta_t} \right) \|X\|_*^2 - B \\
&= \frac{1}{2\eta_t} \|X_t - X\|_*^2 - \frac{1}{2\eta_t} \|X_t - X_{t+1}\|_*^2 + \left( \frac{1}{2\eta_{t+1}} - \frac{1}{2\eta_t} \right) \|X\|_*^2 \\
&\quad + \frac{1}{K} \left\langle \sum_{k=1}^K \hat{V}_{k,t+1/2}, X_{t+1/2} - X_{t+1} \right\rangle - \frac{1}{K} \left\langle \sum_{k=1}^K \hat{V}_{k,t+1/2}, X_{t+1/2} - X \right\rangle.
\end{aligned}
$$
$$(*)$$

Next, also by the update rule (with $\gamma_t$), we have for any $X \in \mathbb{R}^d$

$$
\begin{aligned}
\frac{\eta_t}{K} \left\langle \sum_{k=1}^K \hat{V}_{k,t-1/2}, X_{t+1/2} - X \right\rangle &\leq \frac{\gamma_t}{K} \left\langle \sum_{k=1}^K \hat{V}_{k,t-1/2}, X_{t+1/2} - X \right\rangle \\
&= \langle X_t - X_{t+1/2}, X_{t+1/2} - X \rangle \\
&= \frac{1}{2} \|X_t - X\|_*^2 - \frac{1}{2} \|X_t - X_{t+1/2}\|_*^2 - \frac{1}{2} \|X_{t+1/2} - X\|_*^2.
\end{aligned}
$$

Substituting $X = X_{t+1}$ and dividing both sides of the inequality by $\eta_t$, we have

$$
\begin{aligned}
&\frac{1}{K} \left\langle \sum_{k=1}^K \hat{V}_{k,t-1/2}, X_{t+1/2} - X_{t+1} \right\rangle \\
&\leq \frac{1}{2\eta_t} \|X_t - X_{t+1}\|_*^2 - \frac{1}{2\eta_t} \|X_t - X_{t+1/2}\|_*^2 - \frac{1}{2\eta_t} \|X_{t+1/2} - X_{t+1}\|_*^2. \quad (**)
\end{aligned}
$$

Combining (*) with (**) and after some rearrangements, we obtain

$$\frac{1}{K}\left\langle \sum_{k=1}^{K} \hat{V}_{k,t+1/2}, X_{t+1/2} - X \right\rangle \leq \frac{1}{2\eta_t}\|X_t - X\|_*^2 - \frac{1}{2\eta_{t+1}}\|X_{t+1} - X\|_*^2$$

$$+ \left(\frac{1}{2\eta_{t+1}} - \frac{1}{2\eta_t}\right)\|X_1 - X\|_*^2$$

$$+ \frac{1}{K}\left\langle \sum_{k=1}^{K} \hat{V}_{k,t+1/2} - \hat{V}_{k,t-1/2}, X_{t+1/2} - X_{t+1} \right\rangle$$

$$- \frac{1}{2\eta_t}\|X_t - X_{t+1/2}\|_*^2 - \frac{1}{2\eta_t}\|X_{t+1/2} - X_{t+1}\|_*^2.$$

Then, by summing the above expression over $t = 1, 2, \ldots, T$ and with some telescoping terms, we obtain

$$\sum_{t=1}^{T}\frac{1}{K}\left\langle \sum_{k=1}^{K} \hat{V}_{k,t+1/2}, X_{t+1/2} - X \right\rangle \leq \frac{1}{2\eta_1}\|X_1 - X\|_*^2 - \frac{1}{2\eta_{T+1}}\|X_{T+1} - X\|_*^2$$

$$+ \left(\frac{1}{2\eta_{T+1}} - \frac{1}{2\eta_1}\right)\|X_1 - X\|_*^2$$

$$+ \sum_{t=1}^{T}\frac{1}{K}\left\langle \sum_{k=1}^{K}\left(\hat{V}_{k,t+1/2} - \hat{V}_{k,t-1/2}\right), X_{t+1/2} - X_{t+1} \right\rangle$$

$$- \sum_{t=1}^{T}\frac{1}{2\eta_t}\|X_t - X_{t+1/2}\|_*^2 - \sum_{t=1}^{T}\frac{1}{2\eta_t}\|X_{t+1/2} - X_{t+1}\|_*^2.$$

Next we consider the substitution $X_1 = 0$ which is just for notation simplicity and can be relaxed at the expense of obtaining a slightly more complicated expression. We can further drop the term $\frac{1}{2\eta_{T+1}}\|X_{T+1} - X\|_*^2$ to obtain

$$\frac{1}{K}\sum_{t=1}^{T}\left\langle \sum_{k=1}^{K} \hat{V}_{k,t+1/2}, X_{t+1/2} - X \right\rangle \leq \frac{1}{2\eta_{T+1}}\|X\|_*^2$$

$$+ \frac{1}{K}\sum_{t=1}^{T}\left\langle \sum_{k=1}^{K}\left(\hat{V}_{k,t+1/2} - \hat{V}_{k,t-1/2}\right), X_{t+1/2} - X_{t+1} \right\rangle$$

$$- \sum_{t=1}^{T}\frac{1}{2\eta_t}\|X_t - X_{t+1/2}\|_*^2 - \sum_{t=1}^{T}\frac{1}{2\eta_t}\|X_{t+1/2} - X_{t+1}\|_*^2.$$

$$(\dagger)$$

Note that by Cauchy-Schwarz and triangle inequalities, we have

$$\frac{1}{K}\left\langle \sum_{k=1}^{K}\left(\hat{V}_{k,t+1/2} - \hat{V}_{k,t-1/2}\right), X_{t+1/2} - X_{t+1} \right\rangle$$

$$= \frac{1}{K}\sum_{k=1}^{K}\left\langle \hat{V}_{k,t+1/2} - \hat{V}_{k,t-1/2}, X_{t+1/2} - X_{t+1} \right\rangle$$

$$\leq \sum_{k=1}^{K}\left\|\hat{V}_{k,t+1/2} - \hat{V}_{k,t-1/2}\right\|_* \left\|\frac{X_{t+1/2} - X_{t+1}}{K}\right\|_*.$$

Combining with the AM-GM inequality of the form

$$xy \leq \frac{\eta_t}{2K^2}x^2 + \frac{K^2}{2\eta_t}y^2,$$

we deduce from (†) further that

$$\frac{1}{K} \sum_{t=1}^{T} \left\langle \sum_{k=1}^{K} \left( \hat{V}_{k,t+1/2} - \hat{V}_{k,t-1/2} \right), X_{t+1/2} - X_{t+1} \right\rangle$$

$$\leq \sum_{t=1}^{T} \frac{\eta_t}{2K^2} \sum_{k=1}^{K} \left\| \hat{V}_{k,t+1/2} - \hat{V}_{k,t-1/2} \right\|_*^2 + \sum_{t=1}^{T} \frac{1}{2\eta_t} \| X_{t+1/2} - X_{t+1} \|_*^2. \tag{††}$$

Plugging (††) into (†), we obtain

$$\frac{1}{K} \sum_{t=1}^{T} \left\langle \sum_{k=1}^{K} \hat{V}_{k,t+1/2}, X_{t+1/2} - X \right\rangle \leq \frac{\|X\|_*^2}{2\eta_{T+1}} + \sum_{t=1}^{T} \sum_{k=1}^{K} \frac{\eta_t}{2K^2} \left\| \hat{V}_{k,t+1/2} - \hat{V}_{k,t-1/2} \right\|_*^2$$

$$- \sum_{t=1}^{T} \frac{1}{2\eta_t} \| X_t - X_{t+1/2} \|_*^2,$$

as desired. ■

## E.2 GAP ANALYSIS UNDER ABSOLUTE NOISE

We first introduce following two useful lemmas that will help to bound the (GAP):

**Lemma E.2.** *(Levy et al., 2018; McMahan & Streeter, 2010) For all non-negative numbers $\alpha_1, \ldots, \alpha_t$, it holds that*

$$\sqrt{\sum_{t=1}^{T} \alpha_t} \leq \sum_{t=1}^{T} \frac{\alpha_t}{\sqrt{\sum_{i=1}^{t} \alpha_i}} \leq 2\sqrt{\sum_{t=1}^{T} \alpha_t}.$$

**Lemma E.3.** *(Bach & Levy, 2019) Let $\mathcal{C} \in \mathbb{R}^d$ be a convex set and $h : \mathcal{C} \to \mathbb{R}$ be a 1-strongly convex w.r.t. a norm $\|\cdot\|$. Assume that $h(\boldsymbol{x}) - \min_{\boldsymbol{x} \in \mathcal{C}} h(\boldsymbol{x}) \leq D^2/2$ for all $\boldsymbol{x} \in \mathcal{C}$. Then, for any martingale difference $(\boldsymbol{z}_t)_{t=1}^{T} \in \mathbb{R}^d$ and any $\boldsymbol{x} \in \mathcal{C}$, we have*

$$\mathbb{E}\left[ \left\langle \sum_{t=1}^{T} \boldsymbol{z}_t, \boldsymbol{x} \right\rangle \right] \leq \frac{D^2}{2} \sqrt{\sum_{t=1}^{T} \mathbb{E}[\|\boldsymbol{z}_t\|^2]}. \tag{11}$$

Now we state and prove the complexity of Algorithm 1 under absolute noise and fixed compression scheme.

**Theorem 4.6** (Algorithm 1 under Absolute Noise). *Suppose the iterates $X_t$ of Algorithm 1 are updated with learning rate schedule given in (6) for all $t = 1/2, 1, \ldots, T$. Let $\mathcal{X} \subset \mathbb{R}^d$ be a compact neighborhood of a VI solution and $D^2 := \sup_{\boldsymbol{p} \in \mathcal{X}} \| X_1 - \boldsymbol{p} \|_2^2$. Under Assumptions 2.1, 2.2, 2.3, and 2.4, we have*

$$\mathbb{E}\left[ \mathrm{Gap}_{\mathcal{X}}\left( \overline{X}_{t+1/2} \right) \right] = \mathcal{O}\left( \left( (LD + \|A(X_1)\|_2 + \sigma) \widehat{\varepsilon_Q} + \sigma \right) D^2 L^2 / \sqrt{TK} \right).$$

*Proof.* Suppose first that no compression is applied, i.e., $\varepsilon_Q = 0$. Using the result of the template inequality Proposition E.1, we can drop the negative term to obtain

$$\frac{1}{K} \sum_{t=1}^{T} \left\langle \sum_{k=1}^{K} \hat{V}_{k,t+1/2}, X_{t+1/2} - X \right\rangle \leq \frac{\|X\|_*^2}{2\eta_{T+1}} + \sum_{t=1}^{T} \sum_{k=1}^{K} \frac{\eta_t}{2K^2} \| \hat{V}_{k,t+1/2} - \hat{V}_{k,t-1/2} \|_*^2.$$

Next we can expand the LHS with the absolute noise model Assumption 2.4 as follows

$$LHS = \frac{1}{K}\sum_{t=1}^{T}\left\langle\sum_{k=1}^{K}A_k(X_{t+1/2}), X_{t+1/2} - X\right\rangle + \frac{1}{K}\sum_{t=1}^{T}\left\langle\sum_{k=1}^{K}U_k(X_{t+1/2}), X_{t+1/2} - X\right\rangle$$

$$\geq \frac{1}{K}\sum_{t=1}^{T}\left\langle\sum_{k=1}^{K}A_k(X), X_{t+1/2} - X\right\rangle + \frac{1}{K}\sum_{t=1}^{T}\left\langle\sum_{k=1}^{K}U_k(X_{t+1/2}), X_{t+1/2} - X\right\rangle$$

$$= \frac{1}{K}\left\langle\sum_{k=1}^{K}A_k(X), \sum_{t=1}^{T}X_{t+1/2} - \sum_{t=1}^{T}X\right\rangle + \frac{1}{K}\sum_{t=1}^{T}\left\langle\sum_{k=1}^{K}U_k(X_{t+1/2}), X_{t+1/2} - X\right\rangle$$

$$= \frac{T}{K}\sum_{k=1}^{K}\left\langle A_k(X), \bar{X}_{T+1/2} - X\right\rangle + \frac{1}{K}\sum_{t=1}^{T}\left\langle\sum_{k=1}^{K}U_k(X_{t+1/2}), X_{t+1/2} - X\right\rangle,$$

where the second inequality follows from the monotonicity of $A$ and $\bar{X}_{T+1/2} = \sum_{t=1}^{T}X_{t+1/2}/T$. Plugging this back to the result from template inequality with some rearrangement, we obtain

$$\frac{1}{K}\sum_{k=1}^{K}\left\langle A_k(X), \bar{X}_{T+1/2} - X\right\rangle \leq \frac{1}{T}\left(\frac{\|X\|_*^2}{2\eta_{T+1}} + \sum_{t=1}^{T}\sum_{k=1}^{K}\frac{\eta_t}{2K^2}\|\hat{V}_{k,t+1/2} - \hat{V}_{k,t-1/2}\|_*^2\right.$$

$$\left. + \frac{1}{K}\sum_{t=1}^{T}\left\langle\sum_{k=1}^{K}U_k(X_{t+1/2}), X - X_{t+1/2}\right\rangle\right).$$

By taking the supremum over $X$, then dividing by T and then taking expectation on both sides, we get

$$\mathbb{E}\left[\sup_X \frac{1}{K}\sum_{k=1}^{K}\left\langle A_k(X), \bar{X}_{T+1/2} - X\right\rangle\right] \leq \frac{1}{T}(S_1 + S_2 + S_3),$$

where

$$S_1 = \mathbb{E}\left[\frac{D^2}{2\eta_{T+1}}\right]$$

$$S_2 = \mathbb{E}\left[\sum_{t=1}^{T}\sum_{k=1}^{K}\frac{\eta_t}{2K^2}\|\hat{V}_{k,t+1/2} - \hat{V}_{k,t-1/2}\|_*^2\right]$$

$$S_3 = \mathbb{E}\left[\sup_X \frac{1}{K}\sum_{t=1}^{T}\left\langle\sum_{k=1}^{K}U_k(X_{t+1/2}), X - X_{t+1/2}\right\rangle\right].$$

Here we make an important observation that

$$\mathbb{E}\left[\sum_{k=1}^{K}\left\|\hat{V}_{k,t+1/2} - \hat{V}_{k,t-1/2}\right\|_*^2\right] \leq 2\mathbb{E}\left[\sum_{k=1}^{K}\left\|A_k(X_{t+1/2}) - A_k(X_{t-1/2})\right\|_*^2\right]$$

$$+ 2\mathbb{E}\left[\sum_{k=1}^{K}\left\|U_k(X_{t+1/2}) - U_k(X_{t-1/2})\right\|_*^2\right]$$

$$\leq 2\sum_{k=1}^{K}L^2\mathbb{E}\left[\left\|X_{t+1/2} - X_{t-1/2}\right\|_*^2\right] + 4K\sigma^2$$

$$\leq 2KL^2D^2 + 4K\sigma^2, \tag{12}$$

where the second inequality comes from $L$-Lipschitzness the operator for the first summand and the absolute noise assumption for the second summand. Now we proceed to bound these terms one by

one. For $S_1$, from the choice of learning rates $\eta_t \leq 1$, with Equation (12)we obtain

$$S_1 = D^2 \mathbb{E}\left[\sqrt{1 + \sum_{t=1}^{T} \frac{1}{K^2} \sum_{k=1}^{K} \left\|\hat{V}_{k,t+1/2} - \hat{V}_{k,t-1/2}\right\|_*^2}\right]$$

$$\leq D^2 \sqrt{1 + \sum_{t=1}^{T} \mathbb{E}\left[\frac{1}{K^2} \sum_{k=1}^{K} \left\|\hat{V}_{k,t+1/2} - \hat{V}_{k,t-1/2}\right\|_*^2\right]}$$

$$\leq D^2 \sqrt{1 + \frac{2T(L^2 D^2 + 2\sigma^2)}{K}}.$$

Next, we proceed to bound $S_2$

$$S_2 = \mathbb{E}\left[\sum_{t=1}^{T} \sum_{k=1}^{K} \frac{\eta_t}{2K^2} \|\hat{V}_{k,t+1/2} - \hat{V}_{k,t-1/2}\|_*^2\right]$$

$$= \mathbb{E}\left[\sum_{t=1}^{T} \sum_{k=1}^{K} \left(\frac{\eta_t}{2K^2} - \frac{\eta_{t+1}}{2K^2}\right) \|\hat{V}_{k,t+1/2} - \hat{V}_{k,t-1/2}\|_*^2\right]$$

$$+ \mathbb{E}\left[\sum_{t=1}^{T} \sum_{k=1}^{K} \frac{\eta_{t+1}}{2K^2} \|\hat{V}_{k,t+1/2} - \hat{V}_{k,t-1/2}\|_*^2\right]$$

$$\leq \mathbb{E}\left[\sum_{t=1}^{T} \left(\frac{\eta_t}{2K^2} - \frac{\eta_{t+1}}{2K^2}\right)(2KL^2 D^2 + 4K\sigma^2)\right]$$

$$+ \frac{1}{2}\mathbb{E}\left[\sum_{t=1}^{T} \sum_{k=1}^{K} \frac{\|\hat{V}_{k,t+1/2} - \hat{V}_{k,t-1/2}\|_*^2 / K^2}{\sqrt{1 + \sum_{s=1}^{t} \sum_{k=1}^{K} \left\|\hat{V}_{k,s+1/2} - \hat{V}_{k,s-1/2}\right\|^2 / K^2}}\right] \quad \text{(from Equation (12))}$$

$$\leq 2L^2 D^2 + 4\sigma^2 + \frac{1}{2}\mathbb{E}\left[\sqrt{1 + \frac{1}{K^2} \sum_{t=1}^{T} \sum_{k=1}^{K} \left\|\hat{V}_{k,t+1/2} - \hat{V}_{k,t-1/2}\right\|^2}\right] \quad \text{(from Lemma E.2)}$$

$$\leq 2L^2 D^2 + 4\sigma^2 + \frac{1}{2}\sqrt{1 + \frac{2T(L^2 D^2 + 2\sigma^2)}{K}}.$$

Lastly, let's consider $S_3$

$$S_3 = \mathbb{E}\left[\sup_{X} \frac{1}{K} \sum_{t=1}^{T} \left\langle \sum_{k=1}^{K} U_k(X_{t+1/2}), X \right\rangle\right] - \mathbb{E}\left[\sup_{X} \frac{1}{K} \sum_{t=1}^{T} \left\langle \sum_{k=1}^{K} U_k(X_{t+1/2}), X_{t+1/2} \right\rangle\right]$$

We can bound the first term with Lemma E.3 as follows

$$\mathbb{E}\left[\sup_{X} \frac{1}{K} \sum_{t=1}^{T} \left\langle \sum_{k=1}^{K} U_k(X_{t+1/2}), X \right\rangle\right] \leq \frac{D^2}{2K} \sqrt{\mathbb{E}\left[\sum_{t=1}^{T} \sum_{k=1}^{K} \|U_{k,t+1/2}\|^2\right]} \leq \frac{D^2 \sigma \sqrt{T}}{2\sqrt{K}}$$

For the second term, we use law of total expectation

$$\mathbb{E}\left[\sum_{t=1}^{T} \left\langle \sum_{k=1}^{K} U_k(X_{t+1/2}), X_{t+1/2} \right\rangle\right] = \mathbb{E}\left[\sum_{t=1}^{T} \sum_{k=1}^{K} \mathbb{E}\left[\langle U_k(X_{t+1/2}), X_{t+1/2} \rangle | X_{t+1/2}\right]\right] = 0,$$

implying

$$S_3 \leq \frac{D^2 \sigma \sqrt{T}}{2\sqrt{K}}.$$

Combining the bounds of $S_1$, $S_2$ and $S_3$, we finally obtain the complexity without compression as

$$\mathbb{E}\left[\mathrm{Gap}_{\mathcal{X}}\left(\bar{X}_{t+1/2}\right)\right]$$

$$= \mathbb{E}\left[\sup_X \frac{1}{K}\sum_{k=1}^K \left\langle A_k(X), \bar{X}_{T+1/2} - X\right\rangle\right] \leq \frac{1}{T}\mathcal{O}\left(\frac{\sqrt{T}D^2L^2}{\sqrt{K}}\right) = \mathcal{O}\left(\frac{D^2L^2}{\sqrt{TK}}\right).$$

Now, we consider applying layer-wise compression to this bound. Firstly, recall that the average square root expected code-length bound is denoted as

$$\widehat{\varepsilon_Q} = \sum_{m=1}^M \sum_{j=1}^{J^m} \frac{T_{m,j}\sqrt{\varepsilon_{Q,m,j}}}{T}.$$

Finally, by applying compression bound Lemma D.7 along the ideas of (Faghri et al., 2020, Theorem 4) and (Ramezani-Kebrya et al., 2023, Theorem 3), we get the desired result

$$\mathbb{E}\left[\mathrm{Gap}_{\mathcal{X}}\left(\bar{X}_{t+1/2}\right)\right] = \mathcal{O}\left(\frac{((LD + \|A(X_1)\|_2 + \sigma)\widehat{\varepsilon_Q} + \sigma)D^2L^2}{\sqrt{TK}}\right)$$

∎

### E.3 GAP ANALYSIS UNDER RELATIVE NOISE

**Theorem 4.8** (Algorithm 1 under Relative Noise). *Suppose the iterates $X_t$ of Algorithm 1 are updated with learning rate schedule in (6) for all $t = 1/2, 1, \ldots, T$. Let $\mathcal{X} \subset \mathbb{R}^d$ be a compact neighborhood of a VI solution. Let $D^2 := \sup_{\boldsymbol{p}\in\mathcal{X}} \|X_1 - \boldsymbol{p}\|_2^2$. Under Assumptions 2.1, 2.2, 2.3, 2.5, and 4.7, we have*

$$\mathbb{E}\left[\mathrm{Gap}_{\mathcal{X}}\left(\overline{X}_{t+1/2}\right)\right] = \mathcal{O}\left((\sigma_R\overline{\varepsilon_Q} + \overline{\varepsilon_Q} + \sigma_R)D^2/(TK)\right).$$

*Proof.* Plugging $X^\star$ into part of the LHS of template inequality Proposition E.1 and then taking expectation, we obtain

$$\mathbb{E}\left[\left\langle \frac{1}{K}\sum_{k=1}^K \hat{V}_{k,t+1/2}, X_{t+1/2} - X^\star \right\rangle\right] = \mathbb{E}\left[\frac{1}{K}\sum_{k=1}^K \mathbb{E}\left[\langle \hat{V}_{k,t+1/2}, X_{t+1/2} - X^\star\rangle|X_{t+1/2}\right]\right]$$

$$= \mathbb{E}\left[\frac{1}{K}\sum_{k=1}^K \langle A_k(X_{t+1/2}), X_{t+1/2} - X^\star\rangle\right]$$

$$= \mathbb{E}\left[\langle A(X_{t+1/2}), X_{t+1/2} - X^\star\rangle\right]$$

$$\geq \mathbb{E}\left[\langle A(X_{t+1/2}) - A(X^\star), X_{t+1/2} - X^\star\rangle\right]$$

$$\geq \beta\mathbb{E}\left[\|A(X_{t+1/2})\|_*^2\right]$$

$$= \beta\mathbb{E}\left[\frac{1}{K}\sum_{k=1}^K \|A(X_{t+1/2})\|_*^2\right]$$

$$\geq \frac{\beta}{2\sigma_R + 2}\mathbb{E}\left[\frac{1}{K}\sum_{k=1}^K \|\hat{V}_{k,t+1/2}\|_*^2\right],$$

where the fifth step occurs due to the $\beta$-co-coercivity assumption and the last step follows from this inequality resulted from Assumption 2.5

$$\|\hat{V}_{k,t+1/2}\|_*^2 = \|V_{k,t+1/2} + U_{k,t+1/2}\|_*^2 \leq 2\|V_{k,t+1/2}\|_*^2 + 2\|U_{k,t+1/2}\|_*^2 \leq (2 + 2\sigma_R)\|V_{k,t+1/2}\|_*^2.$$

Plugging this back into the template inequality, we deduce

$$\frac{\beta}{2\sigma_R + 2}\sum_{t=1}^T \mathbb{E}\left[\frac{1}{K}\sum_{k=1}^K \|\hat{V}_{k,t+1/2}\|_*^2\right]$$

$$\leq \mathbb{E}\left[\frac{\|X^\star\|_*^2}{2\eta_{T+1}} + \sum_{t=1}^T \frac{\eta_t}{2K^2}\sum_{k=1}^K \left\|\hat{V}_{k,t+1/2} - \hat{V}_{k,t-1/2}\right\|_*^2 - \sum_{t=1}^T \frac{\|X_t - X_{t+1/2}\|_*^2}{2\eta_t}\right],$$

implying

$$\frac{\beta}{2\sigma_R + 2} \sum_{t=1}^{T} \mathbb{E}\left[\frac{1}{K} \sum_{k=1}^{K} \|\hat{V}_{k,t+1/2}\|_*^2\right] \leq \mathbb{E}\left[\frac{\|X^\star\|_*^2}{2\eta_{T+1}} + \sum_{t=1}^{T} \frac{\eta_t}{2K^2} \sum_{k=1}^{K} \left\|\hat{V}_{k,t+1/2} - \hat{V}_{k,t-1/2}\right\|_*^2\right]. \tag{Inq1}$$

On the other hand, we consider

$$\mathbb{E}\left[\sum_{t=1}^{T} \beta\|A(X_{t+1/2})\|_*^2 + \sum_{t=1}^{T} \frac{\|X_t - X_{t+1/2}\|_*^2}{2\eta_t}\right]$$

$$\geq \mathbb{E}\left[\sum_{t=1}^{T} \beta\|A(X_{t+1/2})\|_*^2 + \sum_{t=1}^{T} \frac{\beta^2}{2\eta_t}\|A(X_t) - A(X_{t+1/2})\|_*^2\right]$$

$$\geq \min\left\{\beta, \frac{\beta^2}{2\eta_0}\right\} \sum_{t=1}^{T} \mathbb{E}\left[\|A(X_{t+1/2})\|_*^2 + \|A(X_t) - A(X_{t+1/2})\|_*^2\right]$$

$$\geq \frac{1}{2} \min\left\{\beta, \frac{\beta^2}{2\eta_0}\right\} \sum_{t=1}^{T} \mathbb{E}\left[\|A(X_t)\|_*^2\right]$$

$$\geq \frac{1}{4 + 4\sigma_R} \min\left\{\beta, \frac{\beta^2}{2\eta_0}\right\} \sum_{t=1}^{T} \mathbb{E}\left[\frac{1}{K} \sum_{k=1}^{K} \|\hat{V}_{k,t}\|_*^2\right],$$

where the second step comes from the consequence of the co-coerceivity assumption. Plugging this back to template inequality, we obtain

$$\frac{1}{4 + 4\sigma_R} \min\left\{\beta, \frac{\beta^2}{2\eta_0}\right\} \sum_{t=1}^{T} \mathbb{E}\left[\frac{1}{K} \sum_{k=1}^{K} \|\hat{V}_{k,t}\|_*^2\right]$$

$$\leq \mathbb{E}\left[\frac{\|X^\star\|_*^2}{2\eta_{T+1}} + \sum_{t=1}^{T} \frac{\eta_t}{2K^2} \sum_{k=1}^{K} \left\|\hat{V}_{k,t+1/2} - \hat{V}_{k,t-1/2}\right\|_*^2\right]. \tag{Inq2}$$

Now summing the two above inequalties Inq1 and Inq2, we have

$$\frac{1}{4 + 4\sigma_R} \min\left\{\beta, \frac{\beta^2}{2\eta_0}\right\} \sum_{t=1}^{T} \mathbb{E}\left[\frac{1}{K} \sum_{k=1}^{K} \|\hat{V}_{k,t}\|_*^2\right] + \frac{\beta}{2\sigma_R + 2} \sum_{t=1}^{T} \mathbb{E}\left[\frac{1}{K} \sum_{k=1}^{K} \|\hat{V}_{k,t+1/2}\|_*^2\right]$$

$$\leq \mathbb{E}\left[\frac{\|X^\star\|_*^2}{\eta_{T+1}} + \sum_{t=1}^{T} \frac{\eta_t}{K^2} \sum_{k=1}^{K} \left\|\hat{V}_{k,t+1/2} - \hat{V}_{k,t-1/2}\right\|_*^2\right].$$

Next, from the bounding of $S_2$ from Theorem 4.6, we have

$$\mathbb{E}\left[\sum_{t=1}^{T} \frac{\eta_t}{K^2} \sum_{k=1}^{K} \left\|\hat{V}_{k,t+1/2} - \hat{V}_{k,t-1/2}\right\|_*^2\right] \leq \mathbb{E}\left[\frac{1}{\eta_{T+1}}\right],$$

yielding

$$\frac{1}{4 + 4\sigma_R} \min\left\{\beta, \frac{\beta^2}{2\eta_0}\right\} \sum_{t=1}^{T} \mathbb{E}\left[\frac{1}{K} \sum_{k=1}^{K} \|\hat{V}_{k,t}\|_*^2\right] + \frac{\beta}{2\sigma_R + 2} \sum_{t=1}^{T} \mathbb{E}\left[\frac{1}{K} \sum_{k=1}^{K} \|\hat{V}_{k,t+1/2}\|_*^2\right]$$

$$\leq \mathbb{E}\left[\frac{\|X^\star\|_*^2 + 1}{\eta_{T+1}}\right].$$

On the other hand, we can consider the lower bound for the LHS of this inequality

$$\frac{1}{4 + 4\sigma_R} \min\left\{\beta, \frac{\beta^2}{2\eta_0}\right\} \sum_{t=1}^{T} \mathbb{E}\left[\frac{1}{K} \sum_{k=1}^{K} \|\hat{V}_{k,t}\|_*^2\right] + \frac{\beta}{2\sigma_R + 2} \sum_{t=1}^{T} \mathbb{E}\left[\frac{1}{K} \sum_{k=1}^{K} \|\hat{V}_{k,t+1/2}\|_*^2\right]$$

$$\geq \frac{1}{4 + 4\sigma_R} \min\left\{\beta, \frac{\beta^2}{2\eta_0}\right\} \left(\sum_{t=1}^{T} \mathbb{E}\left[\frac{1}{K} \sum_{k=1}^{K} \|\hat{V}_{k,t}\|_*^2\right] + \sum_{t=1}^{T} \mathbb{E}\left[\frac{1}{K} \sum_{k=1}^{K} \|\hat{V}_{k,t+1/2}\|_*^2\right]\right)$$

$$\geq \frac{K}{2 + 2\sigma_R} \min\left\{\beta, \frac{\beta^2}{2\eta_0}\right\} \mathbb{E}\left[\sum_{t=1}^{T} \sum_{k=1}^{K} \frac{1}{K^2} \|\hat{V}_{k,t+1/2} - \hat{V}_{k,t}\|_*^2\right]$$

$$\geq \frac{K}{4 + 4\sigma_R} \min\left\{\beta, \frac{\beta^2}{2\eta_0}\right\} \mathbb{E}\left[\sum_{t=1}^{T} \left(\sum_{k=1}^{K} \frac{1}{K^2} \|\hat{V}_{k,t+1/2} - \hat{V}_{k,t}\|_*^2 + \sum_{k=2}^{K} \frac{1}{K^2} \|\hat{V}_{k,t} - \hat{V}_{k,t-1/2}\|_*^2\right)\right]$$

$$\geq \frac{K}{2 + 2\sigma_R} \min\left\{\beta, \frac{\beta^2}{2\eta_0}\right\} \mathbb{E}\left[\sum_{t=1}^{T} \sum_{k=2}^{K} \frac{1}{K^2} \|\hat{V}_{k,t+1/2} - \hat{V}_{k,t-1/2}\|_*^2\right]$$

$$\geq \frac{K}{2 + 2\sigma_R} \min\left\{\beta, \frac{\beta^2}{2\eta_0}\right\} \mathbb{E}\left[\frac{1}{\eta_{T+1}^2}\right].$$

Hence we have

$$\frac{K}{2 + 2\sigma_R} \min\left\{\beta, \frac{\beta^2}{2\eta_0}\right\} \left(\mathbb{E}\left[\frac{1}{\eta_{T+1}^2}\right]\right) \leq \mathbb{E}\left[\frac{\|X^\star\|_*^2 + 1}{\eta_{T+1}}\right]$$

$$= (\|X^\star\|_*^2 + 1)\mathbb{E}\left[\sqrt{\frac{1}{\eta_{T+1}^2}}\right]$$

$$\leq (\|X^\star\|_*^2 + 1)\sqrt{\mathbb{E}\left[\frac{1}{\eta_{T+1}^2}\right]},$$

where the last inequality follows from Jensen's inequality. Therefore, we obtain

$$\mathbb{E}\left[\frac{1}{\eta_{T+1}}\right] \leq \frac{2 + 2\sigma_R}{K} \max\left\{\frac{1}{\beta}, \frac{2\eta_0}{\beta^2}\right\}. \tag{13}$$

Similar to the proof of Theorem 4.6 for the absolute noise case, we consider

$$\mathbb{E}\left[\sup_X \frac{1}{K} \sum_{k=1}^{K} \langle A_k(X), \bar{X}_{T+1/2} - X\rangle\right] \leq \frac{1}{T}(S_1 + S_2 + S_3),$$

where

$$S_1 = \mathbb{E}\left[\frac{D^2}{2\eta_{T+1}}\right]$$

$$S_2 = \mathbb{E}\left[\sum_{t=1}^{T} \sum_{k=1}^{K} \frac{\eta_t}{2K^2} \|\hat{V}_{k,t+1/2} - \hat{V}_{k,t-1/2}\|_*^2\right]$$

$$S_3 = \mathbb{E}\left[\sup_X \frac{1}{K} \sum_{t=1}^{T} \left\langle \sum_{k=1}^{K} U_k(X_{t+1/2}), X - X_{t+1/2}\right\rangle\right].$$

Similar to the proof of Theorem 4.6, we have

$$S_2 \leq 2L^2 D^2 + 4\sigma^2 + \mathbb{E}\left[\frac{1}{\eta_{T+1}}\right].$$

Again, we decompose $S_3$ similarly to the proof of Theorem 4.6

$$S_3 = \mathbb{E}\left[\sup_X \frac{1}{K} \sum_{t=1}^{T} \left\langle \sum_{k=1}^{K} U_k(X_{t+1/2}), X\right\rangle\right] - \mathbb{E}\left[\sup_X \frac{1}{K} \sum_{t=1}^{T} \left\langle \sum_{k=1}^{K} U_k(X_{t+1/2}), X_{t+1/2}\right\rangle\right].$$

For the first term of the above expression, we note that

$$\mathbb{E}\left[\sup_X \frac{1}{K}\sum_{t=1}^T \left\langle \sum_{k=1}^K U_k(X_{t+1/2}), X \right\rangle\right] = \frac{1}{K}\mathbb{E}\left[\left\langle \sum_{t=1}^T \sum_{k=1}^K U_{k,t+1/2}, X^o \right\rangle\right]$$

$$= \frac{D^2}{2K}\sqrt{\mathbb{E}\left[\left\|\sum_{t=1}^T \sum_{k=1}^K U_{k,t+1/2}\right\|_*^2\right]}$$

$$\leq \frac{D^2}{2\sqrt{K}}\sqrt{\mathbb{E}\left[\sum_{t=1}^T \sigma_R \left\|A(X_{t+1/2})\right\|_*^2\right]}$$

$$\leq \frac{D^2}{2\sqrt{K}}\sqrt{\sigma_R \mathbb{E}\left[\frac{\|X^*\|_*^2}{2\gamma_{T+1}}\right]}$$

For the second term of $S_3$, we use law of total expectation

$$\mathbb{E}\left[\sum_{t=1}^T \left\langle \sum_{k=1}^K U_k(X_{t+1/2}), X_{t+1/2} \right\rangle\right] = \mathbb{E}\left[\sum_{t=1}^T \sum_{k=1}^K \mathbb{E}\left[\left\langle U_k(X_{t+1/2}), X_{t+1/2}\right\rangle | X_{t+1/2}\right]\right] = 0.$$

Therefore, from the bounds for $S_1, S_2, S_3$, we have the complexity for no compression is

$$\mathbb{E}\left[\mathrm{Gap}_{\mathcal{X}}\left(\bar{X}_{t+1/2}\right)\right] = \mathbb{E}\left[\sup_X \frac{1}{K}\sum_{k=1}^K \left\langle A_k(X), \bar{X}_{T+1/2} - X \right\rangle\right] \leq \mathcal{O}\left(\frac{D^2}{T}\right).$$

Now, we consider layer-wise compression. Firstly, recall that the average variance upper bound is

$$\overline{\varepsilon_Q} = \sum_{m=1}^M \sum_{j=1}^{J^m} \frac{T_{m,j}\varepsilon_{Q,m,j}}{T}.$$

Now with the bound from Lemma D.7, we can follow along the line of (Faghri et al., 2020, Theorem 4) and (Ramezani-Kebrya et al., 2023, Theorem 4) to obtain the final computation complexity with layer-wise compression

$$\mathbb{E}\left[\mathrm{Gap}_{\mathcal{X}}\left(\bar{X}_{t+1/2}\right)\right] = \mathcal{O}\left(\frac{(\sigma_R\overline{\varepsilon_Q} + \overline{\varepsilon_Q} + \sigma_R)D^2}{T}\right).$$

∎

# F   ANALYSIS IN ALMOST SURE BOUNDEDNESS MODEL

## F.1   USEFUL LEMMAS

For the sake of convenience, we introduce the following new notations: [6]

$$\lambda_t = \frac{1}{K^2}\sum_{s=1}^t \left\|\sum_{k=1}^K \hat{V}_{k,s+1/2}\right\|^2, \mu_t = \sum_{s=1}^t \|X_s - X_{s+1}\|^2,$$

yielding

$$\gamma_t = \frac{1}{(1+\lambda_{t-2})^{1/2-\hat{q}}}, \eta_t = \frac{1}{\sqrt{1+\lambda_{t-2}+\mu_{t-2}}}.$$

We now establish some basic lemmas that will be reused through out this theoretical analysis.

**Lemma F.1.** *Let Assumption 2.4 holds. Then for $T \in \mathbb{N}$, we have*

$$\lambda_T \leq 2T(J^2 + \sigma^2).$$

___
[6]For $t \leq 0$, $\lambda_t = \mu_t = 0$.

*Proof.* Using Assumption 2.4, we note that

$$\frac{1}{K^2}\left\|\sum_{k=1}^{K}\hat{V}_{k,t+1/2}\right\|^2 = \left\|\frac{1}{K}\sum_{k=1}^{K}\left(V_{k,t+1/2}+U_{k,t+1/2}\right)\right\|^2$$

$$\leq 2\left\|\frac{1}{K}\sum_{k=1}^{K}V_{k,t+1/2}\right\|^2 + 2\left\|\frac{1}{K}\sum_{k=1}^{K}U_{k,t+1/2}\right\|^2$$

$$\leq \frac{2}{K}\sum_{k=1}^{K}\left\|V_{k,t+1/2}\right\|^2 + \frac{2}{K}\sum_{k=1}^{K}\left\|U_{k,t+1/2}\right\|^2$$

$$\leq J^2 + 2\sigma^2,$$

implying $\lambda_T \leq 2TJ^2 + 2T\sigma^2$. ∎

**Lemma F.2.** *(Hsieh et al., 2022, Lemma 14), a generalization of (Auer et al., 2002, Lemma 3.5) Let $T \in \mathbb{N}, \varepsilon > 0$, and $q \in [0,1)$. For any sequence of non-negative real numbers $a_1, \ldots, a_T$, we have*

$$\sum_{t=1}^{T}\frac{a_t}{\left(\varepsilon + \sum_{s=1}^{t}a_s\right)^q} \leq \frac{1}{1-q}\left(\sum_{t=1}^{T}a_t\right)^{1-q}.$$

Combining the above two lemmas, we deduce the following useful bound

**Lemma F.3.** *Suppose that Assumption 2.4 holds, let $s \in \mathbb{N}$, and $r \in [0,1)$, then for $T \in \mathbb{N}$, we obtain*

$$\sum_{t=1}^{T}\frac{\|\sum_{k=1}^{K}\hat{V}_{k,t+1/2}/K\|^2}{(1+\lambda_{t-s})^r} \leq \frac{\lambda_T^{1-r}}{1-r} + 2s(J^2+\sigma^2).$$

*Proof.* Note that

$$\frac{1}{(1+\lambda_t)^r} \leq \frac{1}{(1+\lambda_{t-s})^r}.$$

Combining the above inequality with bound of $\left\|\sum_{k=1}^{K}\hat{V}_{k,t+1/2}/K\right\|^2$ in Lemma F.1, we deduce

$$\left(\frac{1}{(1+\lambda_{t-s})^r} - \frac{1}{(1+\lambda_t)^r}\right)\left\|\sum_{k=1}^{K}\hat{V}_{k,t+1/2}/K\right\|^2 \leq \left(\frac{1}{(1+\lambda_{t-s})^r} - \frac{1}{(1+\lambda_t)^r}\right)2(J^2+\sigma^2).$$

Combining this inequality with Lemma F.2, we derive

$$\sum_{t=1}^{T}\frac{\|\sum_{k=1}^{K}\hat{V}_{k,t+1/2}/K\|^2}{(1+\lambda_{t-s})^r}$$

$$= \sum_{t=1}^{T}\left(\frac{\|\sum_{k=1}^{K}\hat{V}_{k,t+1/2}/K\|^2}{(1+\lambda_t)^r} + \left(\frac{1}{(1+\lambda_{t-s})^r} - \frac{1}{(1+\lambda_t)^r}\right)\left\|\sum_{k=1}^{K}\hat{V}_{k,t+1/2}/K\right\|^2\right)$$

$$\leq \sum_{t=1}^{T}\frac{\|\sum_{k=1}^{K}\hat{V}_{k,t+1/2}/K\|^2}{(1+\lambda_t)^r} + \sum_{t=1}^{T}\left(\frac{1}{(1+\lambda_{t-s})^r} - \frac{1}{(1+\lambda_t)^r}\right)2(J^2+\sigma^2)$$

$$\leq \frac{\lambda_T^{1-r}}{1-r} + \sum_{t=1-s}^{0}\frac{2(J^2+\sigma^2)}{(1+\lambda_t)^r}$$

$$= \frac{\lambda_T^{1-r}}{1-r} + 2s(J^2+\sigma^2).$$

∎

We also establish the following lemma to bound the inverse of $\eta_t$

**Lemma F.4.** *(Hsieh et al., 2022, Lemma 17) For $T \in \mathbb{N}$, and $a, b \in \mathbb{R}_+$, it occurs that*

$$\frac{a}{\eta_{T+1}} - b \sum_{t=1}^{T} \frac{\|X_t - X_{t+1}\|^2}{\eta_t} \leq a\sqrt{1 + \lambda_{T-1}} + \frac{a^2}{4b}.$$

*Proof.* Note that

$$\frac{a}{\eta_{T+1}} = a\sqrt{1 + \lambda_{T-1} + \mu_{T-1}}$$

$$\leq a\sqrt{1 + \lambda_{T-1}} + a\sqrt{\mu_{T-1}}.$$

And we also have

$$b \sum_{t=1}^{T} \frac{\|X_t - X_{t+1}\|^2}{\eta_t} \geq b \sum_{t=1}^{T} \|X_t - X_{t+1}\|^2$$

$$\geq b\mu_{T-1}.$$

Define function $h : \mathbb{R} \to \mathbb{R}, h(x) = ax - bx^2$. We notice $a\sqrt{\mu_{T-1}} - b\mu_{T-1} \leq \max_{x \in \mathbb{R}} f(x) = a/4b^2$. This concludes the proof. ∎

### F.2 IMPORTANT INEQUALITIES

We start with constructing an energy inequality for (5) (without quantization).

**Proposition F.5.** *[Energy Inequality] Let $(X_t)_{t \in \mathbb{N}}$ and $(X_{t+1/2})_{t \in \mathbb{N}}$ be generated by (5) with non-increasing learning rates. For any $p \in \mathcal{X}$ and $t \geq 2$, it holds*

$$\frac{\|X_{t+1} - p\|^2}{\eta_{t+1}} = \frac{\|X_t - p\|^2}{\eta_t} - \frac{\|X_t - X_{t+1}\|^2}{\eta_t} + \left(\frac{1}{\eta_{t+1}} - \frac{1}{\eta_t}\right)\left(\|X_1 - p\|^2 - \|X_1 - X_{t+1}\|^2\right)$$

$$- \frac{2}{K}\left\langle \sum_{k=1}^{K} \hat{V}_{k,t+1/2}, X_{t+1/2} - p \right\rangle - \frac{2\gamma_t}{K^2}\left\langle \sum_{k=1}^{K} \hat{V}_{k,t+1/2}, \sum_{k=1}^{K} \hat{V}_{k,t-1/2} \right\rangle$$

$$+ \frac{2}{K}\left\langle \sum_{k=1}^{K} \hat{V}_{k,t+1/2}, X_t - X_{t+1} \right\rangle.$$

*Proof.* Using the fact that $\sum_{k=1}^{K} \hat{V}_{k,t+1/2}/K = (X_t - X_1)/\eta_t - (X_{t+1} - X_1)/\eta_{t+1}$, we have

$$\left\langle \sum_{k=1}^{K} \frac{\hat{V}_{k,t+1/2}}{K}, X_{t+1} - p \right\rangle = \left\langle \frac{X_t - X_1}{\eta_t} - \frac{X_{t+1} - X_1}{\eta_{t+1}}, X_{t+1} - p \right\rangle$$

$$= \frac{1}{\eta_t}\langle X_t - X_{t+1}, X_{t+1} - p \rangle$$

$$+ \left(\frac{1}{\eta_{t+1}} - \frac{1}{\eta_t}\right)\langle X_1 - X_{t+1}, X_{t+1} - p \rangle$$

$$= \frac{1}{2\eta_t}(\|X_t - p\|^2 - \|X_{t+1} - p\|^2 - \|X_t - X_{t+1}\|^2)$$

$$+ \left(\frac{1}{2\eta_{t+1}} - \frac{1}{2\eta_t}\right)(\|X_1 - p\|^2 - \|X_{t+1} - p\|^2 - \|X_1 - X_{t+1}\|^2).$$

Multiplying both sides by 2 and rearranging, we obtain

$$\frac{\|X_{t+1} - p\|^2}{\eta_{t+1}} = \frac{\|X_t - p\|^2}{\eta_t} - \frac{\|X_t - X_{t+1}\|^2}{\eta_t} + \left(\frac{1}{\eta_{t+1}} - \frac{1}{\eta_t}\right)\left(\|X_1 - p\|^2 - \|X_1 - X_{t+1}\|^2\right)$$

$$- \frac{2}{K}\left\langle \sum_{k=1}^{K} \hat{V}_{k,t+1/2}, X_{t+1} - p \right\rangle.$$

Lastly, note that

$$
\left\langle \sum_{k=1}^{K} \hat{V}_{k,t+1/2}, X_{t+1} - p \right\rangle = \left\langle \sum_{k=1}^{K} \hat{V}_{k,t+1/2}, X_{t+1/2} - p \right\rangle + \left\langle \sum_{k=1}^{K} \hat{V}_{k,t+1/2}, X_t - X_{t+1/2} \right\rangle
$$

$$
- \left\langle \sum_{k=1}^{K} \hat{V}_{k,t+1/2}, X_t - X_{t+1} \right\rangle
$$

$$
= \left\langle \sum_{k=1}^{K} \hat{V}_{k,t+1/2}, X_{t+1/2} - p \right\rangle + \frac{\gamma_k}{K} \left\langle \sum_{k=1}^{K} \hat{V}_{k,t+1/2}, \sum_{k=1}^{K} \hat{V}_{k,t-1/2} \right\rangle
$$

$$
- \left\langle \sum_{k=1}^{K} \hat{V}_{k,t+1/2}, X_t - X_{t+1} \right\rangle,
$$

yielding the desired expression. ∎

**Corollary F.6** (Energy inequality). *Let $(X_t)_{t\in\mathbb{N}}$ and $(X_{t+1/2})_{t\in\mathbb{N}}$ be generated by (5) with non-increasing learning rates. For any $p \in \mathcal{X}$ and $t \in \mathbb{N}$, it holds that*

$$
\frac{\|X_{t+1} - p\|^2}{\eta_{t+1}} \leq \frac{\|X_t - p\|^2}{\eta_t} + \left( \frac{1}{\eta_{t+1}} - \frac{1}{\eta_t} \right) \|X_1 - p\|^2 - \frac{2}{K} \left\langle \sum_{k=1}^{K} \hat{V}_{k,t+1/2}, X_{t+1/2} - p \right\rangle
$$

$$
- \frac{2\gamma_t}{K^2} \left\langle \sum_{k=1}^{K} \hat{V}_{k,t+1/2}, \sum_{k=1}^{K} \hat{V}_{k,t-1/2} \right\rangle + \frac{\eta_t}{K^2} \left\| \sum_{k=1}^{K} \hat{V}_{k,t+1/2} \right\|^2
$$

$$
+ \min \left( \frac{\eta_t}{K^2} \left\| \sum_{k=1}^{K} \hat{V}_{k,t+1/2} \right\|^2 - \frac{\|X_t - X_{t+1}\|^2}{2\eta_t}, 0 \right).
$$

*Proof.* By Young's inequality,

$$
\frac{2}{K} \left\langle \sum_{k=1}^{K} \hat{V}_{t+1/2}, X_t - X_{t+1} \right\rangle
$$

$$
\leq \min \left( \frac{\eta_t}{K^2} \left\| \sum_{k=1}^{K} \hat{V}_{t+1/2} \right\|^2 + \frac{\|X_t - X_{t+1}\|^2}{\eta_t}, \ \frac{2\eta_t}{K^2} \left\| \sum_{k=1}^{K} \hat{V}_{t+1/2} \right\|^2 + \frac{\|X_t - X_{t+1}\|^2}{2\eta_t} \right)
$$

$$
= \frac{\eta_t}{K^2} \left\| \sum_{k=1}^{K} \hat{V}_{t+1/2} \right\|^2 + \frac{\|X_t - X_{t+1}\|^2}{\eta_t} + \min \left( 0, \frac{\eta_t}{K^2} \left\| \sum_{k=1}^{K} \hat{V}_{t+1/2} \right\|^2 - \frac{\|X_t - X_{t+1}\|^2}{2\eta_t} \right)
$$

Using this inequality and dropping the non-positive term $-\left( \frac{1}{\eta_{t+1}} - \frac{1}{\eta_t} \right) \|X_1 - X_{t+1}\|^2$ from the result of Proposition F.5, we can obtain the required inequality. ∎

Next, we can evaluate the noise and further expand the energy inequality (Corollary F.6) in the following lemma

**Lemma F.7.** *For $t \geq 2$, it holds that*

$$
\mathbb{E} \left[ \frac{-2\gamma_t}{K^2} \left\langle \sum_{k=1}^{K} \hat{V}_{k,t+1/2}, \sum_{k=1}^{K} \hat{V}_{k,t-1/2} \right\rangle \right] \leq \mathbb{E} \left[ \frac{-\gamma_t}{K^2} \left\| \sum_{k=1}^{K} V_{k,t+1/2} \right\|^2 + \frac{-\gamma_t}{K^2} \left\| \sum_{k=1}^{K} V_{k,t-1/2} \right\|^2 \right.
$$

$$
+ \frac{\gamma_t}{K^2} \left\| \sum_{k=1}^{K} V_{k,t+1/2} - \sum_{k=1}^{K} V_{k,t-1/2} \right\|^2
$$

$$
\left. + L(\gamma_t^2 + (\gamma_t + \eta_t)^2) \|\mathbf{U}_{t-1/2}\|^2 \right].
$$

*Proof.* We use $V_{k,t}$ as a shorthand for $A_k(X_t)$ and $\hat{V}_{k,t} = V_{k,t} + U_{k,t}$, where $U_{k,t}$ is the zero mean noise. By the law of total expectation

$$
\begin{aligned}
\mathbb{E}\left[\frac{-2\gamma_t}{K^2}\left\langle\sum_{k=1}^{K}\hat{V}_{k,t+1/2}, \sum_{k=1}^{K}\hat{V}_{k,t-1/2}\right\rangle\right] &= \mathbb{E}\left[\frac{-2\gamma_t}{K^2}\left\langle\mathbb{E}\left[\sum_{k=1}^{K}\hat{V}_{k,t+1/2}\right], \sum_{k=1}^{K}\hat{V}_{k,t-1/2}\right\rangle\right] \\
&= \mathbb{E}\left[\frac{-2\gamma_t}{K^2}\left\langle\sum_{k=1}^{K}V_{k,t+1/2}, \sum_{k=1}^{K}V_{k,t-1/2}\right\rangle\right. \\
&\quad \left. + \frac{-2\gamma_t}{K^2}\left\langle\sum_{k=1}^{K}V_{k,t+1/2}, \sum_{k=1}^{K}U_{k,t-1/2}\right\rangle\right].
\end{aligned}
$$

First, note that

$$
\begin{aligned}
\frac{-2\gamma_t}{K^2}\left\langle\sum_{k=1}^{K}V_{k,t+1/2}, \sum_{k=1}^{K}V_{k,t-1/2}\right\rangle &= \frac{-\gamma_t}{K^2}\left\|\sum_{k=1}^{K}V_{k,t+1/2}\right\|^2 + \frac{-\gamma_t}{K^2}\left\|\sum_{k=1}^{K}V_{k,t-1/2}\right\|^2 \\
&\quad + \frac{\gamma_t}{K^2}\left\|\sum_{k=1}^{K}V_{k,t+1/2} - \sum_{k=1}^{K}V_{k,t-1/2}\right\|^2,
\end{aligned}
$$

implying

$$
\begin{aligned}
\mathbb{E}\left[\frac{-2\gamma_t}{K^2}\left\langle\sum_{k=1}^{K}\hat{V}_{k,t+1/2}, \sum_{k=1}^{K}\hat{V}_{k,t-1/2}\right\rangle\right] &= \mathbb{E}\left[-\frac{\gamma_t}{K^2}\left\|\sum_{k=1}^{K}V_{k,t+1/2}\right\|^2 - \frac{\gamma_t}{K^2}\left\|\sum_{k=1}^{K}V_{k,t-1/2}\right\|^2\right. \\
&\quad + \frac{\gamma_t}{K^2}\left\|\sum_{k=1}^{K}V_{k,t+1/2} - \sum_{k=1}^{K}V_{k,t-1/2}\right\|^2 \\
&\quad \left. - \frac{2\gamma_t}{K^2}\left\langle\sum_{k=1}^{K}V_{k,t+1/2}, \sum_{k=1}^{K}U_{k,t-1/2}\right\rangle\right].
\end{aligned}
$$

$$\text{(\ddag)}$$

From the update rules of (5), we have

$$
X_{t+1/2} = X_t - \frac{\gamma_t}{K}\sum_{k=1}^{K}\hat{V}_{k,t-1/2}, \quad X_t = X_1 - \frac{\eta_t}{K}\sum_{s=1}^{t-1}\sum_{k=1}^{K}\hat{V}_{k,s+1/2}.
$$

Combining these two equations, we get

$$
\begin{aligned}
X_{t+1/2} &= X_1 - \frac{\eta_t}{K}\sum_{s=1}^{t-1}\sum_{k=1}^{K}\hat{V}_{k,s+1/2} - \frac{\gamma_t}{K}\sum_{k=1}^{K}\hat{V}_{k,t-1/2} \\
&= X_1 - \frac{\eta_t}{K}\sum_{s=1}^{t-2}\sum_{k=1}^{K}\hat{V}_{k,s+1/2} - \frac{\gamma_t + \eta_t}{K}\sum_{k=1}^{K}\hat{V}_{k,t-1/2} \\
&= X_1 - \frac{\eta_t}{K}\sum_{s=1}^{t-2}\sum_{k=1}^{K}\hat{V}_{k,s+1/2} - \frac{\gamma_t + \eta_t}{K}\sum_{k=1}^{K}\left(V_{k,t-1/2} + U_{k,t-1/2}\right).
\end{aligned}
$$

Now, let $\sum_{k=1}^{K}U_{k,t}/K = \mathbf{U}_t$ as the sum of all the noises from K nodes at time t. It is clear that $\mathbf{U}_t$ also has zero mean. Let $\tilde{X}_{t+1/2} = X_{t+1/2} + (\eta_t + \gamma_t)\mathbf{U}_{t-1/2}$ to be a surrogate for $X_{t+1/2}$ when removing the noise of time $t-1$. We then obtain

$$
\tilde{X}_{t+1/2} = X_1 - \frac{\eta_t}{K}\sum_{s=1}^{t-2}\sum_{k=1}^{K}\hat{V}_{k,s+1/2} - \frac{\gamma_t + \eta_t}{K}\sum_{k=1}^{K}V_{k,t-1/2}.
$$

Applying the notations $\mathbf{U}_{t-1/2} = \sum_{k=1}^K U_{k,t-1/2}/K$ and $A_k(X_{t+1/2}) = V_{k,t+1/2}$ into (‡), we have

$$\mathbb{E}\left[\frac{-2\gamma_t}{K^2}\left\langle\sum_{k=1}^K \hat{V}_{k,t+1/2}, \sum_{k=1}^K \hat{V}_{k,t-1/2}\right\rangle\right] = \mathbb{E}\left[-\frac{\gamma_t}{K^2}\left\|\sum_{k=1}^K V_{k,t+1/2}\right\|^2 - \frac{\gamma_t}{K^2}\left\|\sum_{k=1}^K V_{k,t-1/2}\right\|^2\right.$$

$$+ \frac{\gamma_t}{K^2}\left\|\sum_{k=1}^K V_{k,t+1/2} - \sum_{k=1}^K V_{k,t-1/2}\right\|^2$$

$$\left. - \frac{2\gamma_t}{K}\left\langle\sum_{k=1}^K A_k(X_{t+1/2}), \mathbf{U}_{t-1/2}\right\rangle\right].$$

We now bound the last term of the RHS of the above expression. First, notice that

$$\mathbb{E}\left[\left\langle\sum_{k=1}^K A_k(\tilde{X}_{t+1/2}), \mathbf{U}_{t-1/2}\right\rangle\right] = \left\langle\sum_{k=1}^K A_k(\tilde{X}_{t+1/2}), \mathbb{E}[\mathbf{U}_{t-1/2}]\right\rangle = 0$$

With that and the L-Lipschitz of $A_k$, we deduce

$$-\mathbb{E}\left[\left\langle\sum_{k=1}^K A_k(X_{t+1/2}), \mathbf{U}_{t-1/2}\right\rangle\right] = -\mathbb{E}\left[\left\langle\sum_{k=1}^K A_k(X_{t+1/2}) - A_k(\tilde{X}_{t+1/2}), \mathbf{U}_{t-1/2}\right\rangle\right]$$

$$- \mathbb{E}\left[\left\langle\sum_{k=1}^K A_k(\tilde{X}_{t+1/2}), \mathbf{U}_{t-1/2}\right\rangle\right]$$

$$= \mathbb{E}\left[\left\langle\sum_{k=1}^K A_k(\tilde{X}_{t+1/2}) - A_k(X_{t+1/2}), \mathbf{U}_{t-1/2}\right\rangle\right]$$

$$\leq \mathbb{E}\left[KL\|\tilde{X}_{t+1/2} - X_{t+1/2}\|\|\mathbf{U}_{t-1/2}\|\right]$$

$$\leq \mathbb{E}\left[KL\left(\frac{\|\tilde{X}_{t+1/2} - X_{t+1/2}\|^2}{2\gamma_t} + \frac{\gamma_t\|\mathbf{U}_{t-1/2}\|^2}{2}\right)\right]$$

$$= \mathbb{E}\left[KL\left(\frac{(\gamma_t + \eta_t)^2\|\mathbf{U}_{t-1/2}\|^2}{2\gamma_t} + \frac{\gamma_t\|\mathbf{U}_{t-1/2}\|^2}{2}\right)\right],$$

yielding

$$\frac{-2\gamma_t}{K}\mathbb{E}\left[\left\langle\sum_{k=1}^K A_k(X_{t+1/2}), \mathbf{U}_{t-1/2}\right\rangle\right] \leq \mathbb{E}\left[L\left((\gamma_t + \eta_t)^2\|\mathbf{U}_{t-1/2}\|^2 + \gamma_t^2\|\mathbf{U}_{t-1/2}\|^2\right)\right].$$

In brief, we get

$$\mathbb{E}\left[\frac{-2\gamma_t}{K^2}\left\langle\sum_{k=1}^K \hat{V}_{k,t+1/2}, \sum_{k=1}^K \hat{V}_{k,t-1/2}\right\rangle\right] \leq \mathbb{E}\left[\frac{-\gamma_t}{K^2}\left\|\sum_{k=1}^K V_{k,t+1/2}\right\|^2 + \frac{-\gamma_t}{K^2}\left\|\sum_{k=1}^K V_{k,t-1/2}\right\|^2\right.$$

$$+ \frac{\gamma_t}{K^2}\left\|\sum_{k=1}^K V_{k,t+1/2} - \sum_{k=1}^K V_{k,t-1/2}\right\|^2$$

$$\left. + L(\gamma_t^2 + (\gamma_t + \eta_t)^2)\|\mathbf{U}_{t-1/2}\|^2\right],$$

as desired. ∎

Now we can establish the quasi-descent inequality for (5) as follows

**Theorem F.8** (Quasi-descent Inequality). *For $t \geq 2$, it holds that*

$$\mathbb{E}\left[\frac{\|X_{t+1} - p\|^2}{\eta_{t+1}}\right]$$

$$\leq \mathbb{E}\left[\frac{\|X_t - p\|^2}{\eta_t} + \left(\frac{1}{\eta_{t+1}} - \frac{1}{\eta_t}\right)\|X_1 - p\|^2 - \frac{2}{K}\left\langle \sum_{k=1}^{K} V_{k,t+1/2}, X_{t+1/2} - p \right\rangle\right.$$

$$- \frac{\gamma_t}{K^2}\left\|\sum_{k=1}^{K} V_{k,t+1/2}\right\|^2 - \frac{\gamma_t}{K^2}\left\|\sum_{k=1}^{K} V_{k,t-1/2}\right\|^2 + \frac{\gamma_t}{K^2}\left\|\sum_{k=1}^{K} V_{k,t+1/2} - \sum_{k=1}^{K} V_{k,t-1/2}\right\|^2$$

$$+ \min\left(\frac{\eta_t}{K^2}\left\|\sum_{k=1}^{K} \hat{V}_{k,t+1/2}\right\|^2 - \frac{\|X_t - X_{t+1}\|^2}{2\eta_t}, 0\right)$$

$$\left. + \frac{\eta_t}{K^2}\left\|\sum_{k=1}^{K} \hat{V}_{k,t+1/2}\right\|^2 + L\left((\gamma_t + \eta_t)^2 + \gamma_t^2\right)\|\mathbf{U}_{t-1/2}\|^2\right].$$

*Proof.* This result immediately follows from plugging Lemma F.7 into Corollary F.6. ∎

With this quasi-descent inequality, we pick the learning rates as follows

$$\gamma_t = \left(1 + \sum_{s=1}^{t-2}\sum_{k=1}^{K}\left\|\frac{\hat{V}_{k,s+1/2}}{K}\right\|^2\right)^{\hat{q}-\frac{1}{2}}, \eta_t = \left(1 + \sum_{s=1}^{t-2}\sum_{k=1}^{K}\left\|\frac{\hat{V}_{k,s+1/2}}{K}\right\|^2 + \|X_s - X_{s+1}\|^2\right)^{-\frac{1}{2}}.$$

Similar to AdaGrad (Duchi et al., 2011), we include the the sum of the squared norm of the feedback in the denominators, helping to control the various positive terms appearing in the quasi-descent inequality, like $\frac{\eta_t}{K^2}\left\|\sum_{k=1}^{K} \hat{V}_{k,t+1/2}\right\|^2$ and $L\left((\gamma_t + \eta_t)^2 + \gamma_t^2\right)\|\mathbf{U}_{t-1/2}\|^2$. Nonetheless, this sum is not taken to the same exponent in the definition of the two learning rates. This scale separation ensures that the contribution of the term $-\frac{\gamma_t}{K^2}\left\|\sum_{k=1}^{K} V_{k,t+1/2}\right\|^2$ remains negative, which is crucial for deriving constant regret under multiplicative noise. As a technical detail, the term $\sum_{s=1}^{t-2}\|X_s - X_{s+1}\|^2$ is included in the definition of $\eta_t$ for controlling the difference

$$\frac{\gamma_t}{K^2}\left\|\sum_{k=1}^{K} V_{k,t+1/2} - \sum_{k=1}^{K} V_{k,t-1/2}\right\|^2 - \frac{\|X_t - X_{t+1}\|^2}{2\eta_t}.$$

Some technical insight is that $\gamma_t$ and $\eta_t$ should at least be in the order of $\Omega\left(1/t^{\frac{1}{2}-\hat{q}}\right)$ and $\Omega\left(1/t^{\frac{1}{2}}\right)$.

We can restructure the quasi-descent inequality Theorem F.8 as follows.

**Lemma F.9** (Alt Template Inequality). *Let $(X_t)_{t\in\mathbb{N}}$ and $(X_{t+1/2})_{t\in\mathbb{N}}$ be generated by (5) with non-increasing learning rates $\eta_t$ and $\gamma_t$ from the Alt schedule, such that $\eta_t \leq \gamma_t$ for all $t \in \mathbb{N}$. For any $p \in \mathcal{X}$ and $T \in \mathbb{N}$, it holds*

$$\mathbb{E}\left[\sum_{t=1}^{T}\left\langle\frac{1}{K}\sum_{k=1}^{K} V_{k,t+1/2}, X_{t+1/2} - p\right\rangle\right] \leq \mathbb{E}\left[\frac{\|X_1 - p\|^2}{2\eta_{T+1}} + \sum_{t=1}^{T}\frac{\eta_t}{2K^2}\left\|\sum_{k=1}^{K} \hat{V}_{k,t+1/2}\right\|^2\right.$$

$$+ \frac{3L^2}{K^2}\sum_{t=2}^{T}\gamma_t^3\left\|\sum_{k=1}^{K} \hat{V}_{k,t-1/2}\right\|^2$$

$$\left. + \frac{3L^2}{2}\sum_{t=2}^{T}\gamma_t\|X_t - X_{t-1}\|^2 + \frac{5L}{2}\sum_{t=2}^{T}\gamma_t^2\|\mathbf{U}_{t-1/2}\|^2\right].$$

*Proof.* From Theorem F.8, by dropping non-positive terms and using the fact that

$$\min\left(\frac{\eta_t}{K^2}\left\|\sum_{k=1}^{K}\hat{V}_{k,t+1/2}\right\|^2 - \frac{\|X_t - X_{t+1}\|^2}{2\eta_t}, 0\right) \le 0,$$

we obtain

$$\mathbb{E}\left[\frac{\|X_{t+1} - p\|^2}{\eta_{t+1}}\right] \le \mathbb{E}\left[\frac{\|X_t - p\|^2}{\eta_t} + \left(\frac{1}{\eta_{t+1}} - \frac{1}{\eta_t}\right)\|X_1 - p\|^2\right.$$

$$- \frac{2}{K}\left\langle\sum_{k=1}^{K}V_{k,t+1/2}, X_{t+1/2} - p\right\rangle + \frac{\gamma_t}{K^2}\left\|\sum_{k=1}^{K}V_{k,t+1/2} - \sum_{k=1}^{K}V_{k,t-1/2}\right\|^2$$

$$\left.+ \frac{\eta_t}{K^2}\left\|\sum_{k=1}^{K}\hat{V}_{k,t+1/2}\right\|^2 + L\left((\gamma_t + \eta_t)^2 + \gamma_t^2\right)\|\mathbf{U}_{t-1/2}\|^2\right].$$

Rearranging the terms, and multiplying both sides by $1/2$, we obtain

$$\mathbb{E}\left[\left\langle\frac{1}{K}\sum_{k=1}^{K}V_{k,t+1/2}, X_{t+1/2} - p\right\rangle\right]$$

$$\le \mathbb{E}\left[\frac{\|X_t - p\|^2}{2\eta_t} - \frac{\|X_{t+1} - p\|^2}{2\eta_{t+1}} + \left(\frac{1}{2\eta_{t+1}} - \frac{1}{2\eta_t}\right)\|X_1 - p\|^2 + \frac{\eta_t}{2K^2}\left\|\sum_{k=1}^{K}\hat{V}_{k,t+1/2}\right\|^2\right. \quad (\star)$$

$$\left.+ \frac{\gamma_t}{2K^2}\left\|\sum_{k=1}^{K}V_{k,t+1/2} - \sum_{k=1}^{K}V_{k,t-1/2}\right\|^2 + \frac{L\left((\gamma_t + \eta_t)^2 + \gamma_t^2\right)}{2}\|\mathbf{U}_{t-1/2}\|^2\right].$$

Note that this inequality holds for $t \ge 2$ as suggested by Theorem F.8. If $t = 1$, then we know

$$\|X_2 - p\|^2 = \|X_1 - p\|^2 - \frac{2\eta_2}{K}\left\langle\sum_{k=1}^{K}\hat{V}_{k,3/2}, X_1 - p\right\rangle + \frac{\eta_2^2}{K^2}\left\|\sum_{k=1}^{K}\hat{V}_{k,3/2}\right\|^2.$$

Setting $X_{3/2} = X_1 = 0$ and $\eta_1 = \eta_2$, we can obtain

$$\mathbb{E}\left[\left\langle\frac{1}{K}\sum_{k=1}^{K}\hat{V}_{k,3/2}, X_1 - p\right\rangle\right] = \mathbb{E}\left[\frac{\|X_1 - p\|^2}{2\eta_2} - \frac{\|X_2 - p\|^2}{2\eta_2} + \frac{\eta_1\left\|\sum_{k=1}^{K}\hat{V}_{k,3/2}\right\|^2}{2K^2}\right] \quad (\star\star)$$

Now, we sum the inequality $(\star)$ over $t$ from 2 to $T$ and then add $(\star\star)$, yielding

$$\mathbb{E}\left[\sum_{t=1}^{T}\left\langle\frac{1}{K}\sum_{k=1}^{K}V_{k,t+1/2}, X_{t+1/2} - p\right\rangle\right]$$

$$\le \mathbb{E}\left[\frac{\|X_1 - p\|^2}{2\eta_{T+1}} + \sum_{t=1}^{T}\frac{\eta_t}{2K^2}\left\|\sum_{k=1}^{K}\hat{V}_{k,t+1/2}\right\|^2 + \sum_{t=2}^{T}\frac{\gamma_t}{2K^2}\left\|\sum_{k=1}^{K}V_{k,t+1/2} - \sum_{k=1}^{K}V_{k,t-1/2}\right\|^2\right.$$

$$\left.+ \sum_{t=2}^{T}\frac{L\left((\gamma_t + \eta_t)^2 + \gamma_t^2\right)}{2}\|\mathbf{U}_{t-1/2}\|^2\right]$$

$$\le \mathbb{E}\left[\frac{\|X_1 - p\|^2}{2\eta_{T+1}} + \sum_{t=1}^{T}\frac{\eta_t}{2K^2}\left\|\sum_{k=1}^{K}\hat{V}_{k,t+1/2}\right\|^2 + \sum_{t=2}^{T}\frac{\gamma_t}{2K^2}\left\|\sum_{k=1}^{K}V_{k,t+1/2} - \sum_{k=1}^{K}V_{k,t-1/2}\right\|^2\right.$$

$$\left.+ \sum_{t=2}^{T}\frac{5L\gamma_t^2}{2}\|\mathbf{U}_{t-1/2}\|^2\right],$$

$$(\ddagger\ddagger)$$

where the last step follows $\eta_t \leq \gamma_t$. We also can bound the difference term as follows

$$\left\| \sum_{k=1}^K V_{k,t+1/2} - \sum_{k=1}^K V_{k,t-1/2} \right\|^2 \leq 3 \left\| \sum_{k=1}^K V_{k,t+1/2} - \sum_{k=1}^K V_{k,t} \right\|^2 + 3 \left\| \sum_{k=1}^K V_{k,t} - \sum_{k=1}^K V_{k,t-1} \right\|^2$$

$$+ 3 \left\| \sum_{k=1}^K V_{k,t-1} - \sum_{k=1}^K V_{k,t-1/2} \right\|^2 .$$

Note that by the $L$-Lipschitz continuity and the update rule of (5), we have

$$3 \left\| \sum_{k=1}^K V_{k,t+1/2} - \sum_{k=1}^K V_{k,t} \right\|^2 = 3 \left\| \sum_{k=1}^K (A_k(X_{t+1/2}) - A_k(X_t)) \right\|^2$$

$$\leq 3 \left\| \sum_{k=1}^K L\|X_{t+1/2} - X_t\| \right\|^2$$

$$= 3K^2 L^2 \|X_{t+1/2} - X_t\|^2$$

$$= 3L^2 \gamma_t^2 \left\| \sum_{k=1}^K \hat{V}_{k,t-1/2} \right\|^2 .$$

After bounding the second and third terms in a similar manner, we obtain

$$\left\| \sum_{k=1}^K V_{k,t+1/2} - \sum_{k=1}^K V_{k,t-1/2} \right\|^2$$

$$\leq 3L^2 \gamma_t^2 \left\| \sum_{k=1}^K \hat{V}_{k,t-1/2} \right\|^2 + 3K^2 L^2 \|X_t - X_{t-1}\|^2 + 3L^2 \gamma_{t-1}^2 \left\| \sum_{k=1}^K \hat{V}_{k,t-3/2} \right\|^2 . \qquad \text{(D.1.1)}$$

Using the initialization that $\hat{V}_{k,1/2} = 0 \; \forall \, k \in [K]$, we have

$$\sum_{t=2}^T \frac{\gamma_t}{2K^2} \left\| \sum_{k=1}^K V_{k,t+1/2} - \sum_{k=1}^K V_{k,t-1/2} \right\|^2$$

$$\leq \sum_{t=2}^T \frac{3L^2 \gamma_t^3}{K^2} \left\| \sum_{k=1}^K \hat{V}_{k,t-1/2} \right\|^2 + \sum_{t=2}^T \frac{3L^2 \gamma_t}{2} \|X_t - X_{t-1}\|^2 . \qquad \text{(D.1.2)}$$

Combining this with the inequality (‡‡), we finally obtain

$$\mathbb{E}\left[ \sum_{t=1}^T \left\langle \frac{1}{K} \sum_{k=1}^K V_{k,t+1/2}, X_{t+1/2} - p \right\rangle \right] \leq \mathbb{E}\left[ \frac{\|X_1 - p\|^2}{2\eta_{T+1}} + \sum_{t=1}^T \frac{\eta_t}{2K^2} \left\| \sum_{k=1}^K \hat{V}_{k,t+1/2} \right\|^2 \right.$$

$$+ \frac{3L^2}{K^2} \sum_{t=2}^T \gamma_t^3 \left\| \sum_{k=1}^K \hat{V}_{k,t-1/2} \right\|^2$$

$$\left. + \frac{3L^2}{2} \sum_{t=2}^T \gamma_t \|X_t - X_{t-1}\|^2 + \frac{5L}{2} \sum_{t=2}^T \gamma_t^2 \|\mathbf{U}_{t-1/2}\|^2 \right] .$$

$$\blacksquare$$

### F.3 BOUND ON SUM OF SQUARED NORMS

We start to bound the sum of squared norms by first revamping the quasi-descent inequality Theorem F.8 in a different way.

**Lemma F.10.** *Let $(X_t)_{t \in \mathbb{N}}$ and $(X_{t+1/2})_{t \in \mathbb{N}}$ be generated by (5) with non-increasing learning rates $\eta_t$ and $\gamma_t$ from Alt schedule, such that $\eta_t \le \gamma_t$ for all $t \in \mathbb{N}$. For $T \in \mathbb{N}$ and $x^\star \in \mathcal{X}^\star$, we have*

$$\sum_{t=2}^{T} \mathbb{E}\left[ \frac{\gamma_t}{K^2} \left\| \sum_{k=1}^{K} V_{k,t+1/2} \right\|^2 + \frac{\gamma_t}{K^2} \left\| \sum_{k=1}^{K} V_{k,t-1/2} \right\|^2 \right]$$

$$\le \mathbb{E}\left[ \frac{\|X_1 - x^\star\|^2}{\eta_{T+1}} + \sum_{t=2}^{T} \frac{6L^2 \gamma_t^3}{K^2} \left\| \sum_{k=1}^{K} \hat{V}_{k,t-1/2} \right\|^2 + \sum_{t=1}^{T} \frac{3\gamma_t}{K^2} \left\| \sum_{k=1}^{K} \hat{V}_{k,t} - \sum_{k=1}^{K} \hat{V}_{k,t+1} \right\|^2 \right.$$

$$\left. - \sum_{t=1}^{T} \frac{\|X_t - X_{t+1}\|^2}{2\eta_t} + \sum_{t=2}^{T} \frac{2\eta_t}{K^2} \left\| \sum_{k=1}^{K} \hat{V}_{k,t+1/2} \right\|^2 + 5L \sum_{t=2}^{T} \gamma_t^2 \|\mathbf{U}_{t-1/2}\|^2 \right],$$

*Proof.* It is straightforwards that

$$\min\left( \frac{\eta_t}{K^2} \left\| \sum_{k=1}^{K} \hat{V}_{k,t+1/2} \right\|^2 - \frac{\|X_t - X_{t+1}\|^2}{2\eta_t}, 0 \right) \le \frac{\eta_t}{K^2} \left\| \sum_{k=1}^{K} \hat{V}_{k,t+1/2} \right\|^2 - \frac{\|X_t - X_{t+1}\|^2}{2\eta_t}.$$

Next, similar to (D.1.1), we have

$$\left\| \sum_{k=1}^{K} V_{k,t+1/2} - \sum_{k=1}^{K} V_{k,t-1/2} \right\|^2$$

$$\le 3L^2 \gamma_t^2 \left\| \sum_{k=1}^{K} \hat{V}_{k,t-1/2} \right\|^2 + 3 \left\| \sum_{k=1}^{K} \hat{V}_{k,t} - \sum_{k=1}^{K} \hat{V}_{k,t-1} \right\|^2 + 3L^2 \gamma_{t-1}^2 \left\| \sum_{k=1}^{K} \hat{V}_{k,t-3/2} \right\|^2.$$

And since $\eta_t \le \gamma_t$, note that

$$L\left( (\gamma_t + \eta_t)^2 + \gamma_t^2 \right) \|\mathbf{U}_{t-1/2}\|^2 \le 5L\gamma^2 \|\mathbf{U}_{t-1/2}\|^2$$

With these inequalities, we can rewrite quasi-descent inequality Theorem F.8 as

$$\mathbb{E}\left[ \frac{\|X_{t+1} - x^\star\|^2}{\eta_{t+1}} \right]$$

$$\le \mathbb{E}\left[ \frac{\|X_t - x^\star\|^2}{\eta_t} + \left( \frac{1}{\eta_{t+1}} - \frac{1}{\eta_t} \right) \|X_1 - x^\star\|^2 - \frac{\gamma_t}{K^2} \left\| \sum_{k=1}^{K} V_{k,t+1/2} \right\|^2 - \frac{\gamma_t}{K^2} \left\| \sum_{k=1}^{K} V_{k,t-1/2} \right\|^2 \right.$$

$$+ \frac{3L^2 \gamma_t^3}{K^2} \left\| \sum_{k=1}^{K} \hat{V}_{k,t-1/2} \right\|^2 + \frac{3L^2 \gamma_t \gamma_{t-1}^2}{K^2} \left\| \sum_{k=1}^{K} \hat{V}_{k,t-3/2} \right\|^2 + \frac{3\gamma_t}{K^2} \left\| \sum_{k=1}^{K} \hat{V}_{k,t} - \sum_{k=1}^{K} \hat{V}_{k,t-1} \right\|^2$$

$$\left. + \frac{2\eta_t}{K^2} \left\| \sum_{k=1}^{K} \hat{V}_{k,t+1/2} \right\|^2 - \frac{\|X_t - X_{t+1}\|^2}{2\eta_t} + 5L\gamma_t^2 \|\mathbf{U}_{t-1/2}\|^2 \right].$$

Summing from $t = 2$ to $T$ of the above, we obtain the following after some rearrangements

$$\sum_{t=2}^{T} \mathbb{E}\left[ \frac{\gamma_t}{K^2} \left\| \sum_{k=1}^{K} V_{k,t+1/2} \right\|^2 + \frac{\gamma_t}{K^2} \left\| \sum_{k=1}^{K} V_{k,t-1/2} \right\|^2 \right]$$

$$\le \mathbb{E}\left[ \frac{\|X_2 - x^\star\|^2}{\eta_2} + \left( \frac{1}{\eta_{T+1}} - \frac{1}{\eta_2} \right) \|X_1 - x^\star\|^2 \right.$$

$$+ \sum_{t=2}^{T} \frac{6L^2 \gamma_t^3}{K^2} \left\| \sum_{k=1}^{K} \hat{V}_{k,t-1/2} \right\|^2 - \sum_{t=2}^{T} \frac{\|X_t - X_{t+1}\|^2}{2\eta_t}$$

$$\left. + \sum_{t=2}^{T} \frac{3\gamma_t}{K^2} \left\| \sum_{k=1}^{K} \hat{V}_{k,t} - \sum_{k=1}^{K} \hat{V}_{k,t-1} \right\|^2 + \sum_{t=2}^{T} \frac{2\eta_t}{K^2} \left\| \sum_{k=1}^{K} \hat{V}_{k,t+1/2} \right\|^2 + 5L \sum_{t=2}^{T} \gamma_t^2 \|\mathbf{U}_{t-1/2}\|^2 \right],$$

$$\tag{D.2.1}$$

in which we use the fact that $\hat{V}_{k,1/2} = 0 \; \forall \, k \in [K]$ and get the bound similar to (D.1.2). Next, note that

$$\sum_{t=2}^{T} \frac{3\gamma_t}{K^2} \left\| \sum_{k=1}^{K} \hat{V}_{k,t} - \sum_{k=1}^{K} \hat{V}_{k,t-1} \right\|^2 = \sum_{t=1}^{T} \frac{3\gamma_{t+1}}{K^2} \left\| \sum_{k=1}^{K} \hat{V}_{k,t} - \sum_{k=1}^{K} \hat{V}_{k,t+1} \right\|^2$$

$$\leq \sum_{t=1}^{T} \frac{3\gamma_t}{K^2} \left\| \sum_{k=1}^{K} \hat{V}_{k,t} - \sum_{k=1}^{K} \hat{V}_{k,t+1} \right\|^2, \qquad (D.2.2)$$

where the last step stems from $\gamma_t \geq \gamma_{t+1}$. If $t = 1$, then we know

$$\|X_2 - x^\star\|^2 = \|X_1 - x^\star\|^2 - \frac{2\eta_2}{K} \left\langle \sum_{k=1}^{K} \hat{V}_{k,3/2}, X_1 - x^\star \right\rangle + \frac{\eta_2^2}{K^2} \left\| \sum_{k=1}^{K} \hat{V}_{k,3/2} \right\|^2$$

$$\leq \|X_1 - x^\star\|^2 + \frac{\eta_2^2}{K^2} \left\| \sum_{k=1}^{K} \hat{V}_{k,3/2} \right\|^2 .$$

This implies

$$\mathbb{E}\left[ \frac{\|X_2 - x^\star\|^2}{\eta_2} \right] \leq \mathbb{E}\left[ \frac{\|X_1 - x^\star\|^2}{\eta_2} + \frac{\eta_2}{K^2} \left\| \sum_{k=1}^{K} \hat{V}_{k,3/2} \right\|^2 \right]$$

$$\leq \mathbb{E}\left[ \frac{\|X_1 - x^\star\|^2}{\eta_2} + \frac{2\eta_2}{K^2} \left\| \sum_{k=1}^{K} \hat{V}_{k,3/2} \right\|^2 - \frac{\|X_1 - X_2\|^2}{2\eta_1} \right]. \qquad (D.2.3)$$

Now plugging (D.2.2) into (D.2.1), and adding (D.2.3), we eventually obtain

$$\sum_{t=2}^{T} \mathbb{E}\left[ \frac{\gamma_t}{K^2} \left\| \sum_{k=1}^{K} V_{k,t+1/2} \right\|^2 + \frac{\gamma_t}{K^2} \left\| \sum_{k=1}^{K} V_{k,t-1/2} \right\|^2 \right]$$

$$\leq \mathbb{E}\left[ \frac{\|X_1 - x^\star\|^2}{\eta_{T+1}} + \sum_{t=2}^{T} \frac{6L^2\gamma_t^3}{K^2} \left\| \sum_{k=1}^{K} \hat{V}_{k,t-1/2} \right\|^2 + \sum_{t=1}^{T} \frac{3\gamma_t}{K^2} \left\| \sum_{k=1}^{K} \hat{V}_{k,t} - \sum_{k=1}^{K} \hat{V}_{k,t+1} \right\|^2 \right.$$

$$\left. - \sum_{t=1}^{T} \frac{\|X_t - X_{t+1}\|^2}{2\eta_t} + \sum_{t=2}^{T} \frac{2\eta_t}{K^2} \left\| \sum_{k=1}^{K} \hat{V}_{k,t+1/2} \right\|^2 + 5L \sum_{t=2}^{T} \gamma_t^2 \|\mathbf{U}_{t-1/2}\|^2 \right].$$

$\blacksquare$

Next, we establish the following lemma to control the sum of some differences

**Lemma F.11.** *Let $(X_t)_{t\in\mathbb{N}}$ and $(X_{t+1/2})_{t\in\mathbb{N}}$ be generated by (5) with non-increasing learning rates $\eta_t$ and $\gamma_t$ from Alt schedule, such that $\eta_t \leq \gamma_t$ for all $t \in \mathbb{N}$. For all $T \in \mathbb{N}$, with almost sure boundedness assumptions from either Assumption 2.4 or 2.5 it holds that*

$$\sum_{t=1}^{T} \frac{3\gamma_t}{K^2} \left\| \sum_{k=1}^{K} \hat{V}_{k,t} - \sum_{k=1}^{K} \hat{V}_{k,t+1} \right\|^2 - \sum_{t=1}^{T} \frac{\|X_t - X_{t+1}\|^2}{4\eta_t} \leq 432L^4 + 24J^2.$$

*Proof.* Define $\bar{t} := \max\left\{ s \in \{0, \ldots, T\} : \eta_s \geq \frac{1}{12L^2} \right\}$. So as to ensure $\bar{t}$ is always well-defined, we can set $\eta_0 \geq \frac{1}{12L^2}$. By definition of $\mu_t$ and $\eta_{\bar{t}}$, we can deduce that $\mu_{\bar{t}-2} \leq 114L^2$. Now since

$\gamma_t \leq 1$, we have

$$\sum_{t=1}^{T} \frac{3\gamma_t}{K^2} \left\| \sum_{k=1}^{K} \hat{V}_{k,t} - \sum_{k=1}^{K} \hat{V}_{k,t+1} \right\|^2$$

$$\leq \sum_{t=1}^{T} \frac{3}{K^2} \left\| \sum_{k=1}^{K} \hat{V}_{k,t} - \sum_{k=1}^{K} \hat{V}_{k,t+1} \right\|^2$$

$$\leq \sum_{t\in[T]/\{\bar{t}-1,\bar{t}\}} \frac{3}{K^2} \left\| \sum_{k=1}^{K} \hat{V}_{k,t} - \sum_{k=1}^{K} \hat{V}_{k,t+1} \right\|^2 + \sum_{t\in\{\bar{t}-1,\bar{t}\}} \frac{3}{K^2} \left\| \sum_{k=1}^{K} \hat{V}_{k,t} - \sum_{k=1}^{K} \hat{V}_{k,t+1} \right\|^2$$

$$\leq \sum_{t\in[T]/\{\bar{t}-1,\bar{t}\}} \frac{3}{K^2} \left( \sum_{k=1}^{K} L\|X_t - X_{t+1}\| \right)^2 + \sum_{t\in\{\bar{t}-1,\bar{t}\}} \frac{6}{K^2} \left( \left\| \sum_{k=1}^{K} \hat{V}_{k,t} \right\|^2 + \left\| \sum_{k=1}^{K} \hat{V}_{k,t+1} \right\|^2 \right)$$

$$\leq \sum_{t\in[T]/\{\bar{t}-1,\bar{t}\}} 3L^2 \|X_t - X_{t+1}\|^2 + \sum_{t\in\{\bar{t}-1,\bar{t}\}} 12J^2$$

$$\leq \sum_{t\in[T]/\{\bar{t}-1,\bar{t}\}} 3L^2 \|X_t - X_{t+1}\|^2 + 24J^2$$

$$= \sum_{t=1}^{\bar{t}-2} 3L^2 \|X_t - X_{t+1}\|^2 + \sum_{t=\bar{t}+1}^{T} 3L^2 \|X_t - X_{t+1}\|^2 + 24J^2$$

$$= 3L^2 \mu_{\bar{t}-2} + \sum_{t=\bar{t}+1}^{T} 3L^2 \|X_t - X_{t+1}\|^2 + 24J^2$$

$$\leq 432L^4 + \sum_{t=\bar{t}+1}^{T} 3L^2 \|X_t - X_{t+1}\|^2 + 24J^2.$$

As $\eta_t \leq \dfrac{1}{12L^2}$ for $t \geq \bar{t}+1$, note that

$$\sum_{t=1}^{T} \frac{\|X_t - X_{t+1}\|^2}{4\eta_t} \geq \sum_{t=\bar{t}+1}^{T} \frac{\|X_t - X_{t+1}\|^2}{4\eta_t} \geq \sum_{t=\bar{t}+1}^{T} 3L^2 \|X_t - X_{t+1}\|^2,$$

yielding

$$\sum_{t=1}^{T} \frac{3\gamma_t}{K^2} \left\| \sum_{k=1}^{K} \hat{V}_{k,t} - \sum_{k=1}^{K} \hat{V}_{k,t+1} \right\|^2 \leq 432L^4 + \sum_{t=1}^{T} \frac{\|X_t - X_{t+1}\|^2}{4\eta_t} + 24J^2.$$

A simple rearrangment of the term $\sum_{t=1}^{T} \|X_t - X_{t+1}\|^2/(4\eta_t)$ will give the desired expression. ∎

Finally, we can establish the bound on sum of squared norms.

**Lemma F.12** (Bound on Sum of Square Norms). *Let $(X_t)_{t\in\mathbb{N}}$ and $(X_{t+1/2})_{t\in\mathbb{N}}$ be generated by (5) with non-increasing learning rates $\eta_t$ and $\gamma_t$ from Alt schedule, such that $\eta_t \leq \gamma_t$ for all $t \in \mathbb{N}$. Denote $D^2 = \sup_{p\in\mathcal{X}} \|X_1 - p\|^2$. For all $T \in \mathbb{N}$, we have*

$$\mathbb{E}\left[ \sum_{t=1}^{T} \frac{\gamma_t}{K^2} \left\| \sum_{k=1}^{K} V_{k,t+1/2} \right\|^2 + \sum_{t=1}^{T} \frac{\|X_t - X_{t+1}\|^2}{8\eta_t} \right] \leq a\mathbb{E}\left[ \sqrt{\lambda_{T-1}} \right] + b,$$

*where $a$ and $b$ are constants with the following values*

$$a = 12L^2 + 10L + 4 + D^2;$$
$$b = (12L^2 + 10L + 8)(J^2 + \sigma^2) + 432L^4 + 24J^2 + D^2 + 2D^4.$$

*Proof.* From Lemma F.10 and Lemma F.11, we have

$$\mathbb{E}\left[\sum_{t=2}^{T}\frac{\gamma_t}{K^2}\left\|\sum_{k=1}^{K}V_{k,t+1/2}\right\|^2 + \sum_{t=2}^{T}\frac{\gamma_t}{K^2}\left\|\sum_{k=1}^{K}V_{k,t-1/2}\right\|^2 + \sum_{t=1}^{T}\frac{\|X_t - X_{t+1}\|^2}{8\eta_t}\right]$$

$$\leq \mathbb{E}\left[\sum_{t=2}^{T}\frac{6L^2\gamma_t^3}{K^2}\left\|\sum_{k=1}^{K}\hat{V}_{k,t-1/2}\right\|^2 + \sum_{t=1}^{T}\frac{3\gamma_t}{K^2}\left\|\sum_{k=1}^{K}\hat{V}_{k,t} - \sum_{k=1}^{K}\hat{V}_{k,t+1}\right\|^2 - \sum_{t=1}^{T}\frac{\|X_t - X_{t+1}\|^2}{4\eta_t}\right.$$

$$\left.+ \frac{\|X_1 - x^\star\|^2}{\eta_{T+1}} - \sum_{t=1}^{T}\frac{\|X_t - X_{t+1}\|^2}{8\eta_t} + \sum_{t=2}^{T}\frac{2\eta_t}{K^2}\left\|\sum_{k=1}^{K}\hat{V}_{k,t+1/2}\right\|^2 + 5L\sum_{t=2}^{T}\gamma_t^2\|\mathbf{U}_{t-1/2}\|^2\right]$$

$$\leq \mathbb{E}\left[\frac{\|X_1 - x^\star\|^2}{\eta_{T+1}} + \sum_{t=2}^{T}\frac{6L^2\gamma_t^3}{K^2}\left\|\sum_{k=1}^{K}\hat{V}_{k,t-1/2}\right\|^2 + 432L^4 + 24J^2\right.$$

$$\left.- \sum_{t=1}^{T}\frac{\|X_t - X_{t+1}\|^2}{8\eta_t} + \sum_{t=2}^{T}\frac{2\eta_t}{K^2}\left\|\sum_{k=1}^{K}\hat{V}_{k,t+1/2}\right\|^2 + 5L\sum_{t=2}^{T}\gamma_t^2\|\mathbf{U}_{t-1/2}\|^2\right].$$

$$\text{(D.4.1)}$$

Now, since $\gamma_t \leq 1$, $\|\mathbf{U}_{t-1/2}\|^2 \leq \left\|\sum_{k=1}^{K}\hat{V}_{k,t-1/2}\right\|^2/K^2$, and $\gamma_{t-1}^2 \leq 1/\sqrt{1+\lambda_{t+1}}$, we have

$$\mathbb{E}\left[\sum_{t=2}^{T}\frac{6L^2\gamma_t^3}{K^2}\left\|\sum_{k=1}^{K}\hat{V}_{k,t-1/2}\right\|^2 + 5L\sum_{t=2}^{T}\gamma_t^2\|\mathbf{U}_{t-1/2}\|^2\right]$$

$$\leq \mathbb{E}\left[\sum_{t=2}^{T}\left(\frac{6L^2\gamma_t^3}{K^2}\left\|\sum_{k=1}^{K}\hat{V}_{k,t-1/2}\right\|^2 + \frac{5L\gamma_t^2}{K^2}\left\|\sum_{k=1}^{K}\hat{V}_{k,t-1/2}\right\|^2\right)\right]$$

$$\leq \mathbb{E}\left[\sum_{t=2}^{T}\left(\frac{6L^2\gamma_t^2}{K^2} + \frac{5L\gamma_t^2}{K^2}\right)\left\|\sum_{k=1}^{K}\hat{V}_{k,t-1/2}\right\|^2\right] = \mathbb{E}\left[\sum_{t=1}^{T-1}\left(6L^2 + 5L\right)\gamma_t^2\left\|\sum_{k=1}^{K}\hat{V}_{k,t+1/2}/K\right\|^2\right]$$

$$\leq (6L^2 + 5L)\mathbb{E}\left[\sum_{t=1}^{T-1}\frac{\left\|\sum_{k=1}^{K}\hat{V}_{k,t+1/2}/K\right\|^2}{\sqrt{1+\lambda_{t-1}}}\right] \leq (6L^2 + 5L)\left(2\mathbb{E}\left[\sqrt{\lambda_{T-1}}\right] + 2(J^2 + \sigma^2)\right).$$

In a similar manner, we can bound

$$\sum_{t=2}^{T}\frac{2\eta_t}{K^2}\left\|\sum_{k=1}^{K}\hat{V}_{k,t+1/2}\right\|^2 \leq 4\mathbb{E}\left[\sqrt{\lambda_{T-1}}\right] + 8(J^2 + \sigma^2).$$

With these two inequality, we can rewrite (D.4.1) as

$$\mathbb{E}\left[\sum_{t=2}^{T}\frac{\gamma_t}{K^2}\left\|\sum_{k=1}^{K}V_{k,t+1/2}\right\|^2 + \sum_{t=2}^{T}\frac{\gamma_t}{K^2}\left\|\sum_{k=1}^{K}V_{k,t-1/2}\right\|^2 + \sum_{t=1}^{T}\frac{\|X_t - X_{t+1}\|^2}{8\eta_t}\right]$$

$$\leq (6L^2 + 5L)(2\mathbb{E}\left[\sqrt{\lambda_{T-1}}\right] + 2(J^2 + \sigma^2)) + 432L^4 + 24J^2 + 4\mathbb{E}\left[\sqrt{\lambda_{T-1}}\right] + 8(J^2 + \sigma^2)$$

$$+ \mathbb{E}\left[\frac{\|X_1 - x^\star\|^2}{\eta_{T+1}} - \sum_{t=1}^{T}\frac{\|X_t - X_{t+1}\|^2}{8\eta_t}\right]$$

$$= (12L^2 + 10L + 4)\mathbb{E}\left[\sqrt{\lambda_{T-1}}\right] + (12L^2 + 10L + 8)(J^2 + \sigma^2) + 432L^4 + 24J^2$$

$$+ \mathbb{E}\left[\frac{\|X_1 - x^\star\|^2}{\eta_{T+1}} - \sum_{t=1}^{T}\frac{\|X_t - X_{t+1}\|^2}{8\eta_t}\right].$$

Note that by the initialization $X_{3/2} = X_1$ and $\gamma_2 = \gamma_1$, we can further simplify the LHS of the above inequality as follows.

$$\mathbb{E}\left[\sum_{t=2}^T \frac{\gamma_t}{K^2}\left\|\sum_{k=1}^K V_{k,t+1/2}\right\|^2 + \sum_{t=2}^T \frac{\gamma_t}{K^2}\left\|\sum_{k=1}^K V_{k,t-1/2}\right\|^2 + \sum_{t=1}^T \frac{\|X_t - X_{t+1}\|^2}{8\eta_t}\right]$$

$$\geq \mathbb{E}\left[\sum_{t=1}^T \frac{\gamma_t}{K^2}\left\|\sum_{k=1}^K V_{k,t+1/2}\right\|^2 + \sum_{t=1}^T \frac{\|X_t - X_{t+1}\|^2}{8\eta_t}\right]$$

Now, we just have to deal with the last term of the sum. With Lemma F.4, we have

$$\mathbb{E}\left[\frac{\|X_1 - x^\star\|^2}{\eta_{T+1}} - \sum_{t=1}^T \frac{\|X_t - X_{t+1}\|^2}{8\eta_t}\right] \leq \mathbb{E}\left[\|X_1 - x^\star\|^2\sqrt{1 + \lambda_{T-1}} + 2\|X_1 - x^\star\|^4\right]$$

$$= D^2\mathbb{E}\left[\sqrt{1 + \lambda_{T-1}}\right] + 2D^4$$

$$\leq D^2\mathbb{E}\left[\sqrt{\lambda_{T-1}}\right] + D^2 + 2D^4,$$

yielding the desired result. ∎

We now establish an useful bound for $\sum_{t=1}^T \mathbb{E}\left[\left\|\sum_{k=1}^K V_{k,t+1/2}/K\right\|^2\right]$.

**Lemma F.13.** *With the Alt learning rate updating schedule and for $T \in \mathbb{N}$, we have*

$$\sum_{t=1}^T \mathbb{E}\left[\left\|\sum_{k=1}^K V_{k,t+1/2}/K\right\|^2\right] = \mathcal{O}(T^{1-\hat{q}}).$$

*Proof.* For $t \in [T]$, note that

$$\gamma_t = \frac{1}{(1 + \lambda_{t-2})^{1/2-\hat{q}}} \leq \frac{1}{(1 + 2\max\{0, t-2\}(J^2 + \sigma^2))^{1/2-\hat{q}}} \leq \frac{1}{(1 + 2T(J^2 + \sigma^2))^{1/2-\hat{q}}},$$

where the second steps follows from Lemma F.1. Now plugging this bound to Lemma F.12, we obtain

$$\frac{\sum_{t=1}^T \mathbb{E}\left[\left\|\sum_{k=1}^K V_{k,t+1/2}/K\right\|^2\right]}{(1 + 2T(J^2 + \sigma^2))^{1/2-\hat{q}}} \leq a\mathbb{E}\left[\sqrt{\lambda_T}\right] + b,$$

where $a$ and $b$ are constants defined similarly to Lemma F.12. By using Lemma F.1 again to get $\sqrt{\lambda_T}$ is of order $\mathcal{O}(\sqrt{T})$, we obtain

$$\sum_{t=1}^T \mathbb{E}\left[\left\|\sum_{k=1}^K V_{k,t+1/2}/K\right\|^2\right] \leq \left(a\mathbb{E}\left[\sqrt{\lambda_T}\right] + b\right)(1 + 2T(J^2 + \sigma^2))^{1/2-\hat{q}}$$

$$= \mathcal{O}\left(\sqrt{T}\right)(1 + 2T(J^2 + \sigma^2))^{1/2-\hat{q}},$$

which equates to $\mathcal{O}\left(T^{1-\hat{q}}\right)$ as desired. ∎

### F.4 GAP ANALYSIS UNDER ABSOLUTE NOISE

**Lemma F.14** (General Bound for GAP). *Let $\mathcal{X} \subset \mathbb{R}^d$ denote a compact neighborhood of a solution for (VI). Let $D^2 := \sup_{p \in \mathcal{X}} \|X_1 - p\|^2$. Suppose that the oracle and the problem (VI) satisfy*

*Assumptions 2.1, 2.2 and 2.3. Let $(X_t)_{t\in\mathbb{N}}$ and $(X_{t+1/2})_{t\in\mathbb{N}}$ be generated by (5) with non-increasing learning rates $\eta_t$ and $\gamma_t$ from Alt schedule, such that $\eta_t \leq \gamma_t$ for all $t \in \mathbb{N}$. It holds*

$$\mathbb{E}\left[\sup_{p\in\mathcal{X}}\left\langle A(p), \bar{X}_{t+1/2} - p\right\rangle\right] \leq \frac{1}{T}\mathbb{E}\left[\left(6L^2 + 5L + \frac{D^2}{2}\right)\sqrt{\lambda_{T-1}} + \sqrt{\lambda_T}\right.$$

$$+\frac{D^2\sqrt{\mu_{T-1}}}{2} + (6L^2 + 5L)(J^2 + \sigma^2)$$

$$\left.+\frac{D^2}{2} + 2(J^2 + \sigma^2) + \frac{3L^2}{2}\sum_{t=1}^{T-1}\|X_{t+1} - X_t\|^2\right].$$

*Proof.* First note that

$$\sup_{p\in\mathcal{X}}\mathbb{E}\left[\sum_{t=1}^{T}\left\langle\frac{1}{K}\sum_{k=1}^{K}V_{k,t+1/2}, X_{t+1/2} - p\right\rangle\right] = \sup_{p\in\mathcal{X}}\mathbb{E}\left[\frac{1}{K}\sum_{k=1}^{K}\left\langle V_{k,t+1/2}, \sum_{t=1}^{T}X_{t+1/2} - p\right\rangle\right]$$

$$\geq \sup_{p\in\mathcal{X}}\mathbb{E}\left[\frac{1}{K}\sum_{k=1}^{K}\left\langle A_k(p), \sum_{t=1}^{T}X_{t+1/2} - p\right\rangle\right]$$

$$= \sup_{p\in\mathcal{X}}\mathbb{E}\left[\frac{T}{K}\sum_{k=1}^{K}\left\langle A_k(p), \bar{X}_{t+1/2} - p\right\rangle\right]$$

$$= T\mathbb{E}\left[\sup_{p\in\mathcal{X}}\left\langle A(p), \bar{X}_{t+1/2} - p\right\rangle\right].$$

where the second inequality stems from the monotonicity of operators $A_k$ for $k \in [K]$. From the template inequality (Lemma F.9) and the two facts that $\gamma_t \leq 1$ and $\sum_{k=1}^{K}\hat{V}_{k,t-1/2}/K \geq \mathbf{U}_{t-1/2}$, we deduce

$$\sup_{p\in\mathcal{X}}\mathbb{E}\left[\frac{1}{K}\sum_{k=1}^{K}\left\langle V_{k,t+1/2}, \bar{X}_{t+1/2} - p\right\rangle\right]$$

$$\leq \mathbb{E}\left[\frac{\|X_1 - p\|^2}{2\eta_{T+1}} + \frac{3L^2}{K^2}\sum_{t=2}^{T}\gamma_t^3\left\|\sum_{k=1}^{K}\hat{V}_{k,t-1/2}\right\|^2 + \frac{3L^2}{2}\sum_{t=2}^{T}\gamma_t\|X_t - X_{t-1}\|^2\right.$$

$$\left.+\sum_{t=1}^{T}\frac{\eta_t}{2K^2}\left\|\sum_{k=1}^{K}\hat{V}_{k,t+1/2}\right\|^2 + \frac{5L}{2K^2}\sum_{t=2}^{T}\gamma_t^2\left\|\sum_{k=1}^{K}\hat{V}_{t-1/2}\right\|^2\right]$$

$$\leq \mathbb{E}\left[\frac{D^2\sqrt{1 + \lambda_{T-1} + \mu_{T-1}}}{2} + \frac{6L^2 + 5L}{2K^2}\sum_{t=1}^{T-1}\gamma_{t+1}^2\left\|\sum_{k=1}^{K}\hat{V}_{k,t+1/2}\right\|^2\right.$$

$$\left.+\frac{3L^2}{2}\sum_{t=1}^{T-1}\|X_{t+1} - X_t\|^2 + \sum_{t=1}^{T}\frac{\eta_t}{2K^2}\left\|\sum_{k=1}^{K}\hat{V}_{k,t+1/2}\right\|^2\right].$$

Now we can analyze three terms of this sum in the following three inequalities.

$$\frac{D^2\sqrt{1 + \lambda_{T-1} + \mu_{T-1}}}{2} \leq \frac{D^2(1 + \sqrt{\lambda_{T-1}} + \sqrt{\mu_{T-1}})}{2}.$$

From Lemma F.3 and the fact that $\gamma_{t+1}^2 \leq 1/\sqrt{1 + \lambda_{t-1}}$, we next have

$$\frac{6L^2 + 5L}{2K^2}\sum_{t=1}^{T-1}\gamma_{t+1}^2\left\|\sum_{k=1}^{K}\hat{V}_{k,t+1/2}\right\|^2 \leq \frac{3L^2 + 5L}{2K^2}\sum_{t=1}^{T-1}\frac{\left\|\sum_{k=1}^{K}\hat{V}_{k,t+1/2}\right\|^2}{\sqrt{1 + \lambda_{t-1}}}$$

$$= \frac{6L^2 + 5L}{2}\sum_{t=1}^{T-1}\frac{\left\|\sum_{k=1}^{K}\hat{V}_{k,t+1/2}/K\right\|^2}{\sqrt{1 + \lambda_{t-1}}}$$

$$\leq (6L^2 + 5L)\left(\sqrt{\lambda_{T-1}} + J^2 + \sigma^2\right).$$

where the last step stems from Lemma F.3 with $s = 1, r = 1/2$. With a similar observation that $\eta_t \leq 1/\sqrt{1 + \lambda_{t-2}}$, we can similarly apply Lemma F.3 and obtain

$$
\begin{aligned}
\sum_{t=1}^{T} \frac{\eta_t}{2K^2} \left\| \sum_{k=1}^{K} \hat{V}_{k,t+1/2} \right\|^2 &\leq \frac{1}{2K^2} \sum_{t=1}^{T} \frac{\left\| \sum_{k=1}^{K} \hat{V}_{k,t+1/2} \right\|^2}{\sqrt{1 + \lambda_{t-2}}} \\
&= \sum_{t=1}^{T} \frac{\left\| \sum_{k=1}^{K} \hat{V}_{k,t+1/2}/K \right\|^2}{2\sqrt{1 + \lambda_{t-2}}} \\
&\leq \sqrt{\lambda_T} + 2(J^2 + \sigma^2).
\end{aligned}
$$

Combining the above three inequalities, we obtain

$$
\sup_{p \in \mathcal{X}} \mathbb{E} \left[ \frac{1}{K} \sum_{k=1}^{K} \left\langle V_{k,t+1/2}, \bar{X}_{t+1/2} - p \right\rangle \right]
$$

$$
\leq \mathbb{E} \left[ \left( 12aL^2 + 6L^2 + 5L + \frac{D^2}{2} \right) \sqrt{\lambda_{T-1}} + \sqrt{\lambda_T} + \frac{D^2}{2} \right.
$$

$$
\left. + (6L^2 + 5L)(J^2 + \sigma^2) + \frac{D^2 \sqrt{\mu_{T-1}}}{2} + 2(J^2 + \sigma^2) + 12L^2 b \right],
$$

implying

$$
\mathbb{E} \left[ \sup_{p \in \mathcal{X}} \left\langle A(p), \bar{X}_{t+1/2} - p \right\rangle \right]
$$

$$
\leq \frac{1}{T} \mathbb{E} \left[ \left( 6L^2 + 5L + \frac{D^2}{2} \right) \sqrt{\lambda_{T-1}} + \sqrt{\lambda_T} + \frac{D^2 \sqrt{\mu_{T-1}}}{2} \right.
$$

$$
\left. + (6L^2 + 5L)(J^2 + \sigma^2) + \frac{D^2}{2} + 2(J^2 + \sigma^2) + \frac{3L^2}{2} \sum_{t=1}^{T-1} \|X_{t+1} - X_t\|^2 \right].
$$

∎

We will now show the convergence of Algorithm 1 with Alt learning rates under absolute noise

**Theorem F.15** (Convergence under Absolute Noise with Alt learning rates). *Let $\mathcal{X} \subset \mathbb{R}^d$ denote a compact neighborhood of a solution for (VI). Let $D^2 := \sup_{p \in \mathcal{X}} \|X_1 - p\|^2$. Let the average square root expected code-length bound $\widehat{\varepsilon_Q} = \sum_{m=1}^{M} \sum_{j=1}^{J^m} T_{m,j} \sqrt{\varepsilon_{Q,m,j}}/T$. Suppose that the oracle and the problem (VI) satisfy Assumptions 2.1, 2.2, 2.3, and 2.4. Let $(X_t)_{t \in \mathbb{N}}$ and $(X_{t+1/2})_{t \in \mathbb{N}}$ be generated by (5) with non-increasing learning rates $\eta_t$ and $\gamma_t$ from Alt schedule, such that $\eta_t \leq \gamma_t$ for all $t \in \mathbb{N}$. It holds that*

$$
\mathbb{E} \left[ \text{Gap}_{\mathcal{X}} \left( \bar{X}_{t+1/2} \right) \right] = \mathcal{O} \left( \frac{\left( (LD + \|A(X_1)\|_2 + \sigma)\widehat{\varepsilon_Q} + \sigma \right) D^4}{\sqrt{T}} \right).
$$

*Proof.* First we consider no compression, i.e. $\varepsilon_Q = 0$. Note that from Lemma F.1, we have $\lambda_T$ and $\lambda_{T-1}$ are $\mathcal{O}(T)$, so $\sqrt{\lambda_T}$ and $\sqrt{\lambda_{T-1}}$ are $\mathcal{O}(\sqrt{T})$. Next by note that

$$
\begin{aligned}
\frac{D^2 \sqrt{\mu_{T-1}}}{2} + \frac{3L^2}{2} \sum_{t=1}^{T-1} \|X_{t+1} - X_t\|^2 &\leq \left( \frac{D^2}{2} + \frac{3L^2}{2} \right) \sum_{t=1}^{T-1} \|X_{t+1} - X_t\|^2 \\
&\leq \left( \frac{D^2}{2} + \frac{3L^2}{2} \right) \sum_{t=1}^{T-1} \frac{\|X_{t+1} - X_t\|^2}{8\eta_t} \\
&\leq \left( \frac{D^2}{2} + \frac{3L^2}{2} \right) \left( a\mathbb{E} \left[ \sqrt{\lambda_{T-1}} \right] + b \right) \\
&= \mathcal{O} \left( D^4 \sqrt{T} \right)
\end{aligned}
$$

where the second last step holds due to Lemma F.12 with the constants $a$ and $b$ defined in the same above lemma, and the last step holds from Lemma F.1. Combining these bounds with Lemma F.14, we obtain

$$\sup_{p \in \mathcal{X}} \mathbb{E}\left[ \frac{1}{K} \sum_{k=1}^{K} \left\langle V_{k,t+1/2}, \bar{X}_{t+1/2} - p \right\rangle \right] = \mathcal{O}\left( D^4 \sqrt{T} \right).$$

Then, without compression, we have

$$\mathbb{E}\left[ \frac{1}{K} \sum_{k=1}^{K} \sup_{p \in \mathcal{X}} \left\langle A_k(p), \bar{X}_{t+1/2} - p \right\rangle \right] \leq \frac{1}{T} \sup_{p \in \mathcal{X}} \mathbb{E}\left[ \sum_{t=1}^{T} \left\langle \frac{1}{K} \sum_{k=1}^{K} V_{k,t+1/2}, X_{t+1/2} - p \right\rangle \right]$$

$$= \mathcal{O}\left( \frac{D^4}{\sqrt{T}} \right).$$

Now, we consider applying layer-wise compression to this bound. Firstly, recall that the average square root expected code-length bound is denoted as

$$\widehat{\varepsilon_Q} = \sum_{m=1}^{M} \sum_{j=1}^{J^m} \frac{T_{m,j} \sqrt{\varepsilon_{Q,m,j}}}{T}.$$

With Lemma D.6, we can follow the ideas established by (Faghri et al., 2020, Theorem 4) and (Ramezani-Kebrya et al., 2023, Theorem 3) and obtain the final computation complexity with layer-wise compression

$$\mathbb{E}\left[ \mathrm{Gap}_{\mathcal{X}} \left( \bar{X}_{t+1/2} \right) \right] = \mathcal{O}\left( \frac{((LD + \|A(X_1)\|_2 + \sigma)\widehat{\varepsilon_Q} + \sigma))D^4}{\sqrt{T}} \right).$$

■

## F.5 GAP ANALYSIS UNDER RELATIVE NOISE

Next for the relative noise case, we first consider this known general bounds for any $N$ non-negative real-valued random variables.

**Lemma F.16.** *(Hsieh et al., 2022, Lemma 21) Let $p, r, s \in \mathbb{R}_+$ such that $p > r, s \in \mathbb{R}_+$, and $(a^1, \ldots, a^N)$ be a collection of any $N$ non-negative real-valued random variables. If, we have*

$$\sum_{i=1}^{N} \mathbb{E}[(a^i)^p] \leq s \sum_{i=1}^{N} \mathbb{E}[(a^i)^r],$$

*then we obtain*

$$\sum_{i=1}^{N} \mathbb{E}[(a^i)^p] \leq N s^{\frac{p}{p-r}}, \sum_{i=1}^{N} \mathbb{E}[(a^i)^r] \leq N s^{\frac{r}{p-r}}.$$

To obtain a better complexity, we now provide a set of improved bounds for the key quantities in the analysis.

**Lemma F.17.** *Assume that the assumption Assumption 2.5 is satisfied, and Alt learning rate update schedule is used. Then, for any $T \in \mathbb{N}$, we obtain*

$$\mathbb{E}\left[ (1 + \lambda_T)^{1/2+\hat{q}} \right] \leq ((1 + \sigma_R)(a + b) + 1)^{1+\frac{1}{2q}}$$

$$\mathbb{E}\left[ \sqrt{1 + \lambda_T} \right] \leq ((1 + \sigma_R)(a + b) + 1)^{\frac{1}{2q}}$$

$$\mathbb{E}\left[ \mu_T \right] \leq 8a((1 + \sigma_R)(a + b) + 1)^{\frac{1}{2q}} + 8b,$$

*where $a, b$ are defined constants in Lemma F.12*

*Proof.* To begin with, we have from Assumption 2.5 that

$$\mathbb{E}\left[\frac{1}{K^2}\left\|\sum_{k=1}^{K}\hat{V}_{k,t+1/2}\right\|^2\right] = \mathbb{E}\left[\frac{1}{K^2}\left\|\sum_{k=1}^{K}V_{k,t+1/2} + \sum_{k=1}^{K}U_{k,t+1/2}\right\|^2\right]$$

$$\leq \mathbb{E}\left[\left\|\frac{1}{K}\sum_{k=1}^{K}V_{k,t+1/2}\right\|^2 + \left\|\frac{1}{K}\sum_{k=1}^{K}U_{k,t+1/2}\right\|^2\right]$$

$$\leq \mathbb{E}\left[\left\|\frac{1}{K}\sum_{k=1}^{K}V_{k,t+1/2}\right\|^2 + \frac{1}{K}\sum_{k=1}^{K}\left\|U_{k,t+1/2}\right\|^2\right]$$

$$\leq \mathbb{E}\left[\left\|\frac{1}{K}\sum_{k=1}^{K}V_{k,t+1/2}\right\|^2 + \frac{\sigma_R}{K}\sum_{k=1}^{K}\left\|A_k(X_{t+1/2})\right\|^2\right]$$

$$\leq \mathbb{E}\left[\left\|\frac{1}{K}\sum_{k=1}^{K}V_{k,t+1/2}\right\|^2 + \sigma_R\left\|A(X_{t+1/2})\right\|^2\right]$$

$$\leq \mathbb{E}\left[\left\|\frac{1}{K}\sum_{k=1}^{K}V_{k,t+1/2}\right\|^2 + \sigma_R\left\|\frac{1}{K}\sum_{k=1}^{K}A_k(X_{t+1/2})\right\|^2\right]$$

$$= (1+\sigma_R)\mathbb{E}\left[\left\|\frac{1}{K}\sum_{k=1}^{K}V_{k,t+1/2}\right\|^2\right],$$

where the last few steps utilize the fact that $A_i = A_j = A$ for all $i, j \in [K]$. Since the learning rates $\gamma_t$ are non-increasing, we can write

$$\sum_{t=1}^{T}\mathbb{E}\left[\frac{\gamma_t}{K^2}\left\|\sum_{k=1}^{K}V_{k,t+1/2}\right\|^2\right] \geq \frac{1}{1+\sigma_R}\sum_{t=1}^{T}\mathbb{E}\left[\frac{\gamma_t}{K^2}\left\|\sum_{k=1}^{K}V_{k,t+1/2}\right\|^2\right]$$

$$\geq \frac{1}{1+\sigma_R}\sum_{t=1}^{T}\mathbb{E}\left[\frac{\gamma_{T+2}}{K^2}\left\|\sum_{k=1}^{K}V_{k,t+1/2}\right\|^2\right]$$

$$= \frac{1}{1+\sigma_R}\mathbb{E}\left[\frac{\sum_{t=1}^{T}\left\|\sum_{k=1}^{K}V_{k,t+1/2}\right\|^2/K^2}{(1+\lambda_T)^{1/2-\hat{q}}}\right]$$

$$= \frac{1}{1+\sigma_R}\mathbb{E}\left[\frac{\lambda_T + 1 - 1}{(1+\lambda_T)^{1/2-\hat{q}}}\right]$$

$$= \frac{1}{1+\sigma_R}\mathbb{E}\left[(1+\lambda_T)^{1/2+\hat{q}}\right] - \frac{1}{1+\sigma_R}\mathbb{E}\left[\frac{1}{(1+\lambda_T)^{1/2-\hat{q}}}\right]$$

$$\geq \frac{1}{1+\sigma_R}\mathbb{E}\left[(1+\lambda_T)^{1/2+\hat{q}}\right] - \frac{1}{1+\sigma_R},$$

implying

$$\mathbb{E}\left[(1+\lambda_T)^{1/2+\hat{q}}\right] \leq (1+\sigma_R)\sum_{t=1}^{T}\mathbb{E}\left[\frac{\gamma_t}{K^2}\left\|\sum_{k=1}^{K}V_{k,t+1/2}\right\|^2\right] + 1.$$

By Lemma F.12, we deduce

$$\mathbb{E}\left[(1+\lambda_T)^{1/2+\hat{q}}\right] \leq a(1+\sigma_R)\mathbb{E}\left[\sqrt{\lambda_{T-1}}\right] + b(1+\sigma_R) + 1$$

$$\leq ((1+\sigma_R)(a+b)+1)\mathbb{E}\left[\sqrt{1+\lambda_{T-1}}\right]$$

where $a, b$ are constants defined in Lemma F.12. Now we utilize Lemma F.16 for $N = 1, p = 1/2 + \hat{q}$, $r = 1/2, s = (1 + \sigma_R)(a + b) + 1$ and $a^1 = 1 + \lambda_T$. This implies

$$\mathbb{E}\left[(1 + \lambda_T)^{1/2+\hat{q}}\right] \leq ((1 + \sigma_R)(a + b) + 1)^{1+\frac{1}{2\hat{q}}}$$

$$\mathbb{E}\left[\sqrt{1 + \lambda_T}\right] \leq ((1 + \sigma_R)(a + b) + 1)^{\frac{1}{2\hat{q}}}.$$

Now combining the second inequality above and Lemma F.12, we finally get

$$\mathbb{E}\left[\mu_T\right] = \sum_{t=1}^{T} \|X_t - X_{t+1}\|^2 \leq \sum_{t=1}^{T} \frac{\|X_t - X_{t+1}\|^2}{8\eta_t} \leq 8a((1 + \sigma_R)(a + b) + 1)^{\frac{1}{2\hat{q}}} + 8b,$$

where $a, b$ are defined constants in Lemma F.12. ∎

**Theorem 5.2** (Algorithm 1 under Relative Noise **without Co-coercivity Assumption**). *Suppose the iterates $X_t$ of Algorithm 1 are updated with learning rate schedule in (Alt) for all $t = 1/2, 1, \ldots, T$. Let $\mathcal{X} \subset \mathbb{R}^d$ be a compact neighborhood of a solution for (VI), $\overline{\varepsilon_Q}$ as in Section 4.2 and $D^2 := \sup_{p \in \mathcal{X}} \|X_1 - p\|_2^2$. Under Assumptions 2.1, 2.2, 2.3, 2.5, and 5.1, for Algorithm 1 with learning rates (Alt), we have*

$$\mathbb{E}\left[\mathrm{Gap}_{\mathcal{X}}\left(\overline{X}_{t+1/2}\right)\right] = \mathcal{O}\left((\sigma_R\overline{\varepsilon_Q} + \overline{\varepsilon_Q} + \sigma_R)D^4/T\right).$$

*Proof.* By plugging Lemma F.17 into Lemma F.14, we have the complexity with no compression is $\mathcal{O}\left(D^4/T\right)$. With the bound from Lemma D.7, we can follow the ideas established by (Faghri et al., 2020, Theorem 4) and (Ramezani-Kebrya et al., 2023, Theorem 4) and obtain the final computation complexity with layer-wise compression

$$\mathbb{E}\left[\mathrm{Gap}_{\mathcal{X}}\left(\bar{X}_{t+1/2}\right)\right] = \mathcal{O}\left(\frac{(\sigma_R\overline{\varepsilon_Q} + \overline{\varepsilon_Q} + \sigma_R)D^4}{T}\right),$$

where $\overline{\varepsilon_Q}$ is the average variance upper bound as

$$\overline{\varepsilon_Q} = \sum_{m=1}^{M} \sum_{j=1}^{J^m} \frac{T_{m,j}\varepsilon_{Q,m,j}}{T}.$$

∎