# OpenReview forum: "Quantized Optimistic Dual Averaging with Adaptive Layer-wise Compression"
_ICLR.cc/2025/Conference — Submitted to ICLR 2025_

### Official Review · Reviewer_qDMJ · 2024-10-16

**Soundness:** 3
**Presentation:** 3
**Contribution:** 3
**Rating:** 6
**Confidence:** 3

**Summary:**

The paper develops a general layer-wise compression framework to solve variational inequalities in a large scale distributed setting. Convergence guarantee for the proposed QODA algorithm with adaptive learning rates is provided. Empirical evidences supporting the claims of the paper are also provided.

**Strengths:**

1. The paper provides theories and guarantees for layer-wise adaptive compression schemes, including tight-variance and code length bounds. The proposed algorithm combines optimism with adaptive layer-wise quantization, achieving state of the art performance.

2. Numerical evidences are provided in support of the effectiveness of the proposed algorithm, with improved in convergence performance.

3. The assumptions used in the paper are less restrictive compared to other existing work.

**Weaknesses:**

See questions.

**Questions:**

1. How do we select between protocal 1 and 2 in real scenario? Is there a clear guideline we can rely on?

2. Will the proposed mechanism impact on the stability of training especially in GANs, are there any experimental evidence?

3. The layer-wised scheme in the paper refers to using different hyper-parameters for compression happens at each layer. Given the importance determined by existing information of each layer, is it possible to develop a compressor, that selects certain layers at some probabilities while temporarily ignores the other layers in this iteration? Previous studies suggests that this leads to benefits in solving empirical risk minimization problem, see [1].

4. The experiments focus on a relatively narrow range of applications (WGAN training), it could benefits from further validation on other models if possible.

---

> ### Author Response · Authors · 2024-11-15
> **Author Rebuttal**
>
> We would like to thank you for your comments on our work. We will address the weaknesses and questions section in the following QnA format:
>
> **Q1**: How do we select between protocol 1 and 2 in real scenario? Is there a clear guideline we can rely on?
>
> **A1**: We kindly refer you to Remark 3.2 in which we have given some suggestions regarding the two protocols. When the network is stable and delays are deterministic, we propose to adopt Protocol 2. On the other hand, when the end-to-end delay for message passing in the underlying network is highly random such as jitters (Verma et al., 1991), we believe that Protocol 1 will be a better choice as a rule of thumb.
>
> **Q2**: Will the proposed mechanism impact on the stability of training especially in GANs, are there any experimental evidence?
>
> **A2**: In the our implementation (supplementary material), we use the same hyperparameters as of the baseline and our method converges, so our method does not adversely affect the stability of training.
>
>
> **Q3**: Given the importance determined by existing information of each layer, is it possible to develop a compressor, that selects certain layers at some probabilities while temporarily ignores the other layers in this iteration? Previous studies suggests that this leads to benefits in solving empirical risk minimization problem, see [1].
>
> **A3**: We hope you can clarify what the work "[1]" is. Each iteration, our framework optimizes the quantization sequences by minimizing the overall quantization variance. If we were to choose certain layers at some probabilities while temporarily ignores the other layers, the solution, i.e. the layer structure, might not be optimal.
>
> References:
>
> D.C. Verma, H. Zhang, and D. Ferrari. Delay jitter control for real-time communication in a packet switching network. In Proceedings of TRICOMM '91: IEEE Conference on Communications Software: Communications for Distributed Applications and Systems, pp. 35–43, 1991. doi:
> 10.1109/TRICOM.1991.152873.

---

> > ### Author Response · Authors · 2024-11-19
> > **Follow-up Discussion**
> >
> > Hi Reviewer qDMJ, we hope that we have answered all of your questions. In summary, we point to the guideline for protocol 1 and 2 in our work and clarify that the training is stable. We hope you can provide the details about the work "[1]" so that we can better answer your third question. We are also more than happy to provide any additional explanations in the remaining time for the discussion period.

---

> > > ### Comment · Reviewer_qDMJ · 2024-11-22
> > > **Response to the authors**
> > >
> > > Thank you very much for the response, I think most of the concerns of mine are resolved. I accidentally missed the reference [1] I am referring to. There is a section in [1] about layer-wised compression for neural networks, which might be relevant to the paper and worth mentioning.
> > >
> > > [1] Det-CGD: Compressed Gradient Descent with Matrix Stepsizes for Non-Convex Optimization. H. Li, A. Karagulyan, P. Richtarik.

---

> > > > ### Author Response · Authors · 2024-11-23
> > > > **Response to Reviewer qDMJ**
> > > >
> > > > Thank you very much for your response. We agree that the layer-wise discussion in [1] is relevant in our related works section. While [1] considers a different setting with sketches which are generally linear operators, our layer-wise framework is for general adaptive unbiased compression. However, the motivation is indeed similar to us, where [1] also try to leverage the layer-wise structure of neural networks to design matrix step-sizes and sketches that are layer-dependent. We will hence include [1] in our related works section (main paper) and a short discussion in the appendix. If our responses have fully addressed your concerns, we kindly hope you might consider revisiting the score. Please let us know if there is anything further we can clarify or improve.
> > > >
> > > > [1] Li, Hanmin, Avetik Karagulyan, and Peter Richtárik. "Det-CGD: Compressed Gradient Descent with Matrix Stepsizes for Non-Convex Optimization." The Twelfth International Conference on Learning Representations.

---

### Official Review · Reviewer_738Y · 2024-11-03

**Soundness:** 3
**Presentation:** 2
**Contribution:** 3
**Rating:** 6
**Confidence:** 2

**Summary:**

The authors develop a general layer-wise and adaptive compression framework with applications to solving variational inequality problems (VI) in a large-scale and distributed setting. This paper also propose a specific algorithm called Quantized and Generalized Optimistic Dual Averaging (QODA) with adaptive learning rates. The best convergence rate is achieved under the monotonicity assumption. Finally, the authors demenstrate the superiority of the QODA algorithm in training time through experiments.

**Strengths:**

1.This work is the first to incorporate optimism in solving distributed VI with adaptive learning rates and layer-wise quantization. The authors propose QODA algorithm and establish joint convergence and communication guarantees for it.

2.The authors provide a very detailed and solid theoretical proof.

3.The experimen show that QODA with layer-wise compression significantly improves the convergence and training time compared to both the SoTA Q-GenX and the full precision baseline.

**Weaknesses:**

I have tried my best to understand the authors' contributions in this paper, but I am not very familiar with the authors' research field. I can only give the authors some suggestions from the perspective of writing.

**Questions:**

I am not an expert in this field, I can only give some advice on writing papers：

1.Line 116, "our work is the first work", deleting the second work.

2.Line 119, missing comma at the end of sentence.

3.Line 166, "such that each node retains only local copy of the current parameter vector" -> "such that each node retains only a local copy of the current parameter vector".

4.Line 179, "We first outline the general formulation for the layer-wise and unbiased quantization." -> "We first outline the general formulation for layer-wise and unbiased quantization.".

5.Line 344, "The proof is in Appendix C." -> "The proof is provided in Appendix C.".

6.Line 365, "The proofs are in Appendix D.2 and D.3, respectively." -> "The proofs are provided in Appendix D.2 and D.3, respectively.".

7.Line 385, "...... are updated with learning rate schedule in (6) for all t=......" -> "...... are updated with the learning rate schedule given in (6) for all t=......".

8.Line 456~Line 460, the tense is different.

---

> ### Author Response · Authors · 2024-11-15
> **Author Rebuttal**
>
> Thank you very much for your time and the writing feedback. We have fixed every writting error you mentioned, and we are more than happy to address any further issue that you have.

---

> > ### Author Response · Authors · 2024-11-29
> > **Follow-up Discussion**
> >
> > Hi Reviewer 738Y,
> >
> > We hope that we have addressed all your concerns in the revised manuscript. We are also more than happy to provide any additional explanations in the remaining time for the discussion period.
> >
> > Best,
> >
> > Submission 6044 Authors

---

### Official Review · Reviewer_rc6N · 2024-11-03

**Soundness:** 3
**Presentation:** 2
**Contribution:** 2
**Rating:** 5
**Confidence:** 3

**Summary:**

This paper proposes a theoretical framework for layer-wise and adaptive unbiased quantization with novel fine-grained coding protocol analysis, which is the first to incorporate distributed VI with adaptive learning rate and layer-wise quantization. They obtain acceptable convergence results under standard noises without the almost sure boundness.

**Strengths:**

1. The theoretical analysis of compression bounds and convergence complexity propose a new framework for understanding distributed VI with layer-wise quantization, illustrating that distributed VI with layer-wise quantization can obtain standard $\Omega(d)$ variance and $\mathcal{O}(1/ \sqrt{T})$ or $\mathcal{O}(1/T)$ convergence rate.
2. The theoretical analysis achieve same rate with more relaxed assumptions compared with SOTA algorithm.

**Weaknesses:**

1. This paper needs more clarification on the importance on the problem they study. In other words, why $\textbf{layer-wise}$ quantization is important in distributed VI? Even though the author introduces the heterogeneity observed by other work, this paper is also lack of enough examples and evidences to explain why this setting makes sense. In large-scale models (more than 1b parameters), especially beyond Transformer-XL or ResNet mentioned in [1], the effect of layer-wise quantization seems not obvious.
2. Even though this paper proposes the compression bounds and convergence rates, it is lack of the theoretical analysis of total communication complexity, which is the per-iteration communication cost multiplies the communication rounds. Noticing that the convergence rate includes $\epsilon_Q$ which is $\Omega(d)$, I think the total communciation complexity is the same order of the algorithm without quiantization, which means no improvement in theory.
3. The writing of  this paper need improvement. This paper is lack of motivation and has too much notations and definitions (like encoding techniques) in main body, and the main theorem needs more remarks. Some typo needs to be modified like a superfluous "literature" in line 60.

[1]  Ilia Markov,Kaveh Alim,Elias Frantar,and Dan Alistarh.L-greco:Layerwise-adaptive gradient compression for efficient data-parallel deep learning.

**Questions:**

1. In this paper, the author considers the unbiased quatization in VI. However, in distributed optimization error feedback technique is widely used to reduce the information loss when encoding. I am interested in whether error feedback and corresponding biased quantization can be  applied to distributed VI and obtain faster rate?
2. The whole framework is still worthy, especially loosing the assumption. If author could explain my concerns in weakness and questions, I would like to improve my scores.

---

> ### Author Response · Authors · 2024-11-15
> **Author Rebuttal**
>
> We first would like to thank you for your time and comments. We will address the weaknesses and questions section in the following QnA format:
>
> **Q1**: This paper needs more clarification on the importance on the problem they study. Why is layer-wise quantization important in distributed VI? In large-scale models (more than 1b parameters), in (Markov et al., 2024), the effect of layer-wise quantization seems not obvious.
>
> **A1**: We would like to clarify that we have stressed the importance of layerwise quantization (in the introduction and in line 173-177). This further motivates us to provide the theory for general layer-wise compression methods. We also hope to clarify that (Markov et al., 2024) indeed shows the superior performance of layer-wise quantization across all the tasks. For instance, in (Markov et al., 2024, Figure 4), we can observe that layer-wise quantization (L-GreCo) leads to gains up to 25% end-to-end speedup compared to uniform global quantization for training Transformer-XL (TXL) on WikiText-103. For the theoretical advatanges over global quantization, we hope to refer you to Remark 3.1, in which we show that layer-wise quantization is always better than global quantization.
>
> In the specific case of distributed VIs, we have shown that our layer-wise quantization framework can practically outperform the global quantization counterpart (Q-GenX) in Figure 1,2. Moreover, we obtain more significant $150\%$ speed up with respect to the unquantized baseline. Theoretically, our layer-wise communication bounds are the generalization of those in Q-GenX, and we provide similar convergence bound under weaker assumptions.
>
> **Q2**: Lack of the theoretical analysis of total communication complexity, which is the per-iteration communication cost multiplies the communication rounds. Noticing that the convergence rate includes $\epsilon_Q$ which is $\Omega(d)$, I think the total communciation complexity is the same order of the algorithm without quiantization, which means no improvement in theory.
>
> **A2**: Regarding the theoretical analysis of total communication complexity, we actually have the results for both the number of communication rounds and per-iteration communication cost. Under the absolute noise case, from Theorem 4.4, we can show that to achieve an $\epsilon$ error, we need $O(1/(K \epsilon^2))$ iterations or communication rounds. Under both protocols, we can get the number of bits per iteration in Theorem 4.2 in the order $O(d)$. Therefore, the total communication complexity is $O(d/(K \epsilon^2))$. This result indeed shows that having nodes, i.e. increase $K$, helps to improve the complexity. Thank you for your suggestion and we will include this discussion in the revised manuscript.
>
> Regarding the improvement with respect to methods without quantization, we first hope to clarify that the key point of quantization is to reduce the cost of communication during training by reducing the amount of information communicated. It is inherent that any quantization scheme will somehow impact the worst-case convergence rate, i.e., number of iterations required to reach a certain accuracy. However, we emphasize as we have shown that, both our theoretically convergence rates and variance bounds are tight. Regarding $\Omega(d)$ in the communication complexity, please note that this is not avoidable even for convex optimization problems with finite-sum structures (Tsitsiklis & Luo, 1987; Korhonen & Alistarh, 2021).
>
> As you mentioned, the rates of our methods indeed have the extra term $\varepsilon_Q$, which accounts for the quantization. This bound is similar to that in Q-GenX. However, we achieve significantly up to 150% speed up with respect to the baseline (no quantization). Hence, the **theoretical improvement** here lies in designing and proving the guarantees of our new methods (QODA + layer-wise quantization) with weaker assumptions while maintaining the optimal rates as Q-GenX. The **practical improvement** is that in our Table 1 and 2, where we achieve significant speedup compared to no quantization method.

---

> > ### Author Response · Authors · 2024-11-15
> > **Author Rebuttal (Continued)**
> >
> > **Q3**: This paper is lack of motivation and has too much notations and definitions (like encoding techniques) in main body, and the main theorem needs more remarks.
> >
> > **A3**: Thank you for your feedback, we have fixed all the typos mentioned by you and the other reviewers. Regarding the heavy notations and definitions, we believe that it is due the complexity of the layer-wise architecture. For each main theorem, we have included comments comparing the bounds to previous bounds in the literature (such as line 344-349, line 366-370, line 407-412). With your feedback, we can write some of them as "remarks" for better clarity.
> >
> > **Q4**: Whether error feedback and corresponding biased quantization can be applied to distributed VI and obtain faster rate?
> >
> > **A4**: We believe that EF is used for extremely high compression scenarios (low-rank or sparsity), but is not necessary for recovering accuracy in quantization, as shown by e.g. NUQSGD (Ramezani-Kebrya et al., 2021, Figure 8) or L-GreCo (Markov et al., 2024). Moreover, EF has additional memory cost (since the error buffer has to be maintained on each node). The fact that we are able to avoid it with good practical results is a strength of the method, and our analysis supports this use as well.
> >
> > References:
> >
> > Ali Ramezani-Kebrya, Fartash Faghri, Ilya Markov, Vitalii Aksenov, Dan Alistarh, and Daniel M.
> > Roy. NUQSGD: Provably communication-efficient data-parallel SGD via nonuniform quantization.
> > Journal of Machine Learning Research (JMLR), 22(114):1–43, 2021.
> >
> > Ilia Markov, Kaveh Alim, Elias Frantar, and Dan Alistarh. L-greco: Layerwise-adaptive gradient
> > compression for efficient data-parallel deep learning. Proceedings of Machine Learning and
> > Systems, 6:312–324, 2024.

---

> ### Author Response · Authors · 2024-11-19
> **Follow-up Discussion**
>
> Hi Reviewer rc6N, we hope that we have answered all of your questions and concerns. In brief, we talk about the importance of layer-wise compression, theoretical analysis of total communication complexity, and error feedback. As you mentioned, resolving the concerns raised in your comments could help further illuminate the strengths of our framework. We look forward to continuing this discussion and are more than happy to provide any additional explanations or details needed.

---

> ### Author Response · Authors · 2024-11-29
> **Author Response to Reviewer rc6N's Follow-up Questions**
>
> We would hope to first thank you for your comment. We will now address the follow-up questions:
> -  Thank you for your insightful comment on error feedback (EF). We agree that EF is popular in practice with impressive practical results in training large language models (LLMs) [1,2,3]. However, from the references you provide, we would argue that in theoretical analysis, EF does not shows better results. Particularly, in both [1, Corollary 1] and [3, Theorem 1], the authors discussed how that 1-bit Adam and 0/1 Adam essentially admits the same convergence rate as distributed SGD, at rate $O(1 / \sqrt{Tn})$. This implies that EF has no **theoretical** advantages compared to general unbiased quantization. Hence, we hope whether you know any theoretical analysis that shows EF has better rate than unbiased global quantization.
>
> - Regarding the communication cost comment, we hope to argue how our rate is not similar to [4,5]. With respect to the rate in [4], we want to highlight that our dependence on the number of workers or nodes is much better $1/\sqrt{K}$ with respect to $1/K^{1/4}$ [4, Table 1] in the regime of more workers. In fact as suggested in line our results **match the known lower bound for convex and smooth optimization $\Omega(1/\sqrt{TK})$ [9, Theorem 1]** under similar setting of absolute noise. Regarding [5], they consider either strongly convex or nonconvex of the finite sum problem which is different from us, so we hope the reviewer can clarify which similar guarantees you are refering to. Regarding the discussion on $d$, we hope to highlight that **our bound matches the lower bound in [8]** in terms of $d$ in the distributed optimisation setting. Both works [6,7] you suggested **are not in the distributed setting**, so the results do not follow the lower bound $\Omega(d)$. Hence, we argue that following the lower bound in [8], we believe that our dependence on $d$ is optimal without further assumptions.
>
> We hope this further clarifies our theoretical contribution, and you can positively re-evaluate our theoretical contributions.
>
> References:
>
> [1] H. Tang, S. Gan, A. A. Awan, S. Rajbhandari, C. Li, X. Lian et al., “1-bit adam: Communication efficient large-scale training with adam’s convergence speed,” in International Conference on Machine Learning. PMLR, 2021, pp. 10118–10129.
>
> [2] C. Li, A. A. Awan, H. Tang, S. Rajbhandari, and Y. He, “1-bit LAMB: communication efficient large-scale large-batch training with lamb’s convergence speed,” in IEEE 29th International Conference on High Performance Computing, Data, and Analytics, 2022, pp. 272–281.
>
> [3] Y. Lu, C. Li, M. Zhang, C. De Sa, and Y. He, “Maximizing communication efficiency for large-scale training via 0/1 adam,” arXiv preprint arXiv:2202.06009, 2022.
>
> [4] Zhize Li, Dmitry Kovalev, Xun Qian, and Peter Richtárik. Acceleration for compressed gradient descent in distributed and federated optimization. arXiv preprint arXiv:2002.11364, 2020.
>
> [5] Konstantin Mishchenko, Eduard Gorbunov, Martin Takáč, and Peter Richtárik. Distributed learning with compressed gradient differences. arXiv preprint arXiv:1901.09269, 2019.
>
> [6] Zeyuan Allen-Zhu, Zheng Qu, Peter Richtárik, and Yang Yuan. Even faster accelerated coordinate descent using non-uniform sampling. In International Conference on Machine Learning, pages 1110–1119. PMLR, 2016.
>
> [7] Filip Hanzely, Konstantin Mishchenko, Peter Richtárik, and A. Sega: Variance reduction via gradient sketching. Advances in Neural Information Processing Systems, 31, 2018.
>
> [8] Korhonen, Janne H., and Dan Alistarh. "Towards tight communication lower bounds for distributed optimisation." Advances in Neural Information Processing Systems 34 (2021): 7254-7266.
>
> [9] Woodworth, Blake E., et al. "The min-max complexity of distributed stochastic convex optimization with intermittent communication." Conference on Learning Theory. PMLR, 2021.

---

> > ### Author Response · Authors · 2024-12-02
> > **Regarding Reviewer rc6N's Follow-up Questions**
> >
> > Dear Reviewer rc6N,
> >
> > As there are fewer than 24 hours left in the discussion period, we hope that our previous reponse to your follow-up questions has clarified our theoretical contributions. If so, we kindly ask you to take a moment to consider increasing the score in support of our work. We are more than happy to clarify any further concerns in the remaining time of the discussion.
> >
> > Best regards,
> >
> > Submission 6044 Authors

---

### Official Review · Reviewer_696T · 2024-11-05

**Soundness:** 3
**Presentation:** 2
**Contribution:** 2
**Rating:** 5
**Confidence:** 3

**Summary:**

## Update

After some discussion with the authors, I am increasing my score as they clarified the novelty of their quantization scheme compared to the prior literature. I remain convinced that the studied setting is still somewhat niche and the empirical results are very limited.

---


This paper develops a theoretical framework for adaptive layer-wise compression in distributed variational inequality (VI) problems, along with a novel algorithm called Quantized Optimistic Dual Averaging (QODA). The work makes several key contributions:
1. It establishes a rigorous theoretical framework for layer-wise and adaptive quantization schemes that accounts for heterogeneity across layers, providing variance and code-length bounds that generalize previous bounds for global quantization.
2. It introduces QODA, which uses optimistic updated with adaptive learning rates. The algorithm achieves optimal convergence rates of $O(1/\sqrt{T})$ and $O(1/T)$ under absolute and relative noise models respectively, without requiring the common assumption of almost sure boundedness of stochastic dual vectors. The provided theory shows how convergence rates depend on the gradient compression constant.
3. The authors also test QODA in training Wasserstein GAN on distributed systems with 4/8/12/16 GPUs.

The work builds upon the combination of Markov et al. (2022, 2024) and Ramezani-Kebrya et al., (2023). The former presented compression schemes and their empirical evaluations, while the latter had theory for distributed VI problems.

**Strengths:**

1. Distributed training remains a challenge and good methods that compress gradients are desired to make it faster.
2. Relaxation of common boundedness assumptions while maintaining optimal convergence rates.

**Weaknesses:**

This paper presents two contributions. The first one is to study the compression schemes previously proposed and studied empirically.
1. It appears to me that the theoretical study is very similar to the well-known bounds of Alistarh et al. (2017), and while the notation is heavy, the underlying math is quite simple. I'm also not confident how important the studied compressions techniques are as they have not been around for a long time and do not seem to be popular in practice, for instance, the github repo of L-Greco has 12 stars.
2. The algorithmic contribution of QODA is marginal. It is a method based on the combination of two widely studied techniques: gradient/operator compression and optimism. Since the compression operator is unbiased, it seems very straightforward to derive the convergence guarantees. The adaptive learning rates have been used in the prior literature on VI problems too.
3. The experimental evaluation is a bit limited and serves more as a proof of concept. However, the work appears to be mostly theoretical, so I wouldn't put too much emphasis on this.

## Minor
line 48 "communication though local updates" -> "communication through local updates"
line 60 "literature communication-efficient liteature" -> "communication-efficient literature"
line 66, "dearth of generalization" -> "death of generalization"?

**Questions:**

1. The authors refer to Q-GenX as "SoTA". Can you clarify in what sense it is state of the art? What exactly makes it worthy improving upon? Is there evidence the method is widely used in practice?
2. The authors study a general family of compressors using arbitrary $\ell_q$ norms. Should we expect a specific choice to always perform better in practice?

---

> ### Author Response · Authors · 2024-11-15
> **Author Rebuttal**
>
> Thank you very much for your time and your comments on our work. We will address the weaknesses and questions section in the following QnA format:
>
> **Q1**: It appears to me that the theoretical study is very similar to the well-known bounds of Alistarh et al. (2017).
>
> **A1**: We kindly refer the reviewer to **line 346-348 and 367-269**, in which we have commented how our bounds generalize or are tigheter than those in (Alistarh et al., 2017). In line 346-348, our quantization variance bound  holds for general $L^q$ normalization and arbitrary sequence of quantization levels in comparison to (Alistarh et al., 2017, Theorem 3.2) which only holds for $L^2$ with uniform quantization levels. In line 367-269, our bound for Protocol 1 is arbitrarily smaller than (Alistarh et al., 2017, Theorem 3.4) in their own setting with global uniform quantization and $L^2$ normalization.
>
> More importantly, (Alistarh et al., 2017) has nothing with VI problems and provides only bounds for general minimization problems, while we also provide joint convergence guarantees for both VI and quantization (refer to Theorem 4.4 and 4.6) aside from the above improvements for the communication bound. Unforutnatley, this is not a fair feedback as we have clearly stated improvements over (Alistarh et al., 2017). We hope the reviewer can re-evaluate our theoretical contributions with the above clarifications, and we are more than happy to clarify any further concerns.
>
> **Q2**: Compressions techniques have not been around for a long time and do not seem to be popular in practice, for instance, the github repo of L-Greco has 12 stars.
>
> **A2**: We do not believe L-GreCo which was recently publised in top ML and systems venue (MLSys) in May 2024 should be devalued based on github stars. We also believe that our contribution to compression literature can be discredited since "compressions techniques have not been around for a long time and do not seem to be popular in practice". In fact, we believe that by providing a comprehensive framework and strong theory for these new layer-wise methods, we can give practioners some ground to start experimenting and implementing these newly published methods for applications such as training large LLMs.
>
> **Q3**: The algorithmic contribution of QODA is marginal. It is a method based on the combination of two widely studied techniques: gradient/operator compression and optimism. The adaptive learning rates have been used in the prior literature on VI problems too.
>
> **A3**: We have developed the first theoretical framework for layer-wise compression, so we kindly request the reviewer to provide any work that theoretically study a similar framework if the reviewer is aware of them. We have also clearly stated in Remark 2.6 that we have eliminate the (almost sure) boundedness of the operator assumption. We believe that it is not "trivial" to remove such assumption since various work as mentioned in Remark 2.6 have utilized it to prove their bounds. Optimism is studied widely in the context of nondistributed VIs, but to the best of our knowledge, it is not yet studied in the context of distributed VIs. As explained in line 300-304, the one less gradient update of optimism reduces the communication burden compared to the non-optimistic counterpart Q-GenX. We hope you can provide further specific feeback about our methods other than suggesting that it is just combination of very general frameworks.

---

> > ### Author Response · Authors · 2024-11-15
> > **Author Rebuttal (Continued)**
> >
> > **Q4**: The authors refer to Q-GenX as "SoTA". Can you clarify in what sense it is state of the art? What exactly makes it worthy improving upon? Is there evidence the method is widely used in practice?
> >
> > **A4**: Q-GenX was published in ICLR last year, and we have improved both theoretically (with weaker assumption) and practically in Figure 1 and 2. SotA here referred to as the latest publication at a top ML venue focusing on unbiased quantization for VI solvers. To address your concern, we will refer to Q-GenX as the global quantization baseline rather than SoTA in the revised manuscript.
> >
> > **Q5**: The authors study a general family of compressors using arbitrary $L^q$ norms. Should we expect a specific choice to always perform better in practice?
> >
> > **A5**: Our framework is general, so we prove our guarantees in arbitrary $L^q$ norms. In practice, the practitioners can utilize whatever norm they prefer. A common choice are $L^2$ and $L^{\infty}$, in QSGD (Alistarh et al., 2017), NUQSGD (Ramezani-Kebrya et al., 2021), ... and our theory also cover all the guarantees in this setting (refer to A1 above).
> >
> > **Minor typos**: Regarding the minor typo errors, we will fix the first two errors in the revised manuscript. For the third error you pointed out, we believe that there is no problem since we use the word "dearth" with the meaning of "scarcity".
> >
> > References:
> >
> > Ali Ramezani-Kebrya, Fartash Faghri, Ilya Markov, Vitalii Aksenov, Dan Alistarh, and Daniel M.
> > Roy. NUQSGD: Provably communication-efficient data-parallel SGD via nonuniform quantization.
> > Journal of Machine Learning Research (JMLR), 22(114):1–43, 2021.
> >
> > Dan Alistarh, Demjan Grubic, Jerry Li, Ryota Tomioka, and Milan Vojnovic. QSGD: Communication-
> > efficient SGD via gradient quantization and encoding. In Advances in Neural Information Processing Systems (NeurIPS), 2017.

---

> > ### Comment · Reviewer_696T · 2024-11-18
> >
> > > our quantization variance bound holds for general $L^q$ normalization
> >
> > Analysis of $L^q$ quantization is not new either, see for example Wang et al. "Atomo: Communication-efficient Learning via Atomic Sparsification" and multiple papers that built upon it. And while Alistarth et al. (2017) did not consider variational inequalities, multiple works of Beznosikov et al. cited in this paper already have the tooling for VI extensions.
> >
> > > we can give practioners some ground to start experimenting and implementing these newly published methods for applications such as training large LLMs
> >
> > As far as I know, LLMs are not trained using VI framework and, to the best of my knowledge, optimistic dual averaging is not used for that either.
> >
> > > we kindly request the reviewer to provide any work that theoretically study a similar framework if the reviewer is aware of them
> >
> > There is plenty of theoretical work on variational inequality, including in distributed setting, e.g. the works of Beznosikov et al. I think the main contribution of this paper is to improve the analysis and remove the bounded gradient assumption. While valuable, the analysis does not seem to be vastly different from what exists in the literature.

---

> ### Author Response · Authors · 2024-11-18
> **Response to Reviewer 696T**
>
> Thank you very much for your response. We hope to clarify that
>
> 1. We refer the reviewer to line 99-104, where we have discussed  Atomo. We have stated that all these quantization schemes are global w.r.t. layers and do not take into account heterogeneities in terms of representation power and impact on the final learning outcome across various layers of neural networks and across training for each layer. We do not claim that our analysis for $L^q$ quantization is novel since it is known in global quantization. Rather, we are the first to provide it for **layer-wise compression**, and we clarify the bounds improve from the global quantization methods such as Alistarth et al. (2017) since the reviewer asked.
>
> 2. When we say "give practioners some ground to start experimenting and implementing these newly published methods for applications such as training large LLMs," we refer the previous words"...**these new layer-wise methods**, we can give practioners ..." Hence, we refer to the use of **layer-wise methods for training LLMs**. The examples are in L-Greco (Markov et al., 2024, Figure 6) where the authors show that layer-wise compression methods outperform that of global compression method (uniform) on training Transformer-XL (TXL) on WikiText-103. Futher examples can be found therein.
>
> 3. We also refer the reviewer to the previous phrases in our answer "**We have developed the first theoretical framework for layer-wise compression**, so we kindly request the reviewer to provide any work that theoretically study a similar framework if the reviewer is aware of them." We hope that the reviewer can share the works that developed the theoretical framework for **layer-wise compression**. Throughout our paper we do not claim the first work for distributed VIs like your response suggested; rather we refer to the **first** theoretical framework for layer-wise compression. We hope this clarify the key contribution.
>
> We hope these clarify our key contributions, and hope that the reviewer can refer to the key details we have provided in the whole answer rather than the later part of the sentence.

---

> > ### Comment · Reviewer_696T · 2024-11-18
> >
> > >  we kindly request the reviewer to provide any work that theoretically study a similar framework if the reviewer is aware of them
> >
> > Layer-wise quantization was studied in (Mishchenko et al., "Distributed Learning with Compressed Gradient Differences") in the context of minimization. However, as mentioned by (Horvath et al., "Stochastic Distributed Learning with Gradient Quantization and Variance Reduction") block quantization is just an instance of unbiased quantization and all studies that allow for general unbiased compression also imply convergence for block- or layer-wise compression.

---

> ### Author Response · Authors · 2024-11-18
> **Response to Reviewer 696T**
>
> Thank you for pointing out these two works. (Horvath et al., 2023) studies the generalization of p-quantization (Mishchenko et al., 2024) without focusing on block quantization, so we will focus on the comparison with block quantization in (Mishchenko et al., 2024). We will add the following comparisons to (Mishchenko et al., 2024) in the revised manuscript:
>
> **Block (p-)quantization** is fundementally different from **layer-wise quantization in our paper**: As (Mishchenko et al., 2024, Definition B.1) suggests, the various blocks here follow the **same** scheme that is p-quantization ($\text{Quant}_p$) which is explained in (Mishchenko et al., 2024, Definition 3.2). There are two fundemental differences compared to our layer-wise quantization:
> - Each of our layer or block in this context has **different adaptive sequences of levels** (refer to our Section 3.1). This is why our method is named **"layer-wise."** (Mishchenko et al., 2024) on the other hand applies the **same** p-quantization scheem $\text{Quant}_p$ to blocks with different sizes, implying that the nature and analysis of two methods are very different. Hence we believe block quantization is not "layer-wise," and its analysis does not apply to **the convergence of our methods**.
> - The way the quantization is calculated for each block or layer are different. In (Mishchenko et al., 2024), the author study and provide guarantees for the following type of p-quantization (for all blocks): $\widetilde{\Delta}=\|\Delta\|_p \operatorname{sign}(\Delta) \circ \xi,$ where the $\xi$ are stacks of random Bernoulli variables. In our work, the sequence of levels for **each layer** is adaptively chosen according to the statistical heterogeneity over the course of training (refer to our Section 3.1, MQV).
>
> Furthermore, (if we are not missing any theorem) the guarantee in (Mishchenko et al., 2024, Theorem 3.3) only cover p-quantization rather block p-quantization. In our Theorem 4.1, we provide the quantization variance bound for any **arbitrary** sequence of levels for each layer in constrast with only levels only based on p-quantization $\widetilde{\Delta}=\|\Delta\|_p \operatorname{sign}(\Delta) \circ \xi$.
>
> In brief, the block quantization is similar to bucketing in unbiased **global** quantizaiton (QSGD (Alistarh et al., 2017), NUQSGD (Ramezani-Kebrya et al., 2021)), which takes into account only the size of different blocks (sub-vectors), while for **layer-wise** quantization we take into account the statistical heterogentiy and impact of different layers on the final accuracy. Due to fundamental differences, our variance and code-length bounds require substantially more involved and different analyses that are not possible by simple extensions of block quantization in those works.
>
> We hope this clarifies your doubts about our novelty with respect to the framework layer-wise compression and its analysis. We will include the above comparisons to (Mishchenko et al., 2024) in revised manuscript. If we have clarified all your concerns, we hope you can positively reconsider your evaluation of our works. We are more than happy to clarify any other concerns.
>
> References:
>
> Mishchenko, Konstantin, Eduard Gorbunov, Martin Takáč, and Peter Richtárik. "Distributed learning with compressed gradient differences." Optimization Methods and Software (2024): 1-16.
>
> Horváth, Samuel, Dmitry Kovalev, Konstantin Mishchenko, Peter Richtárik, and Sebastian Stich. "Stochastic distributed learning with gradient quantization and double-variance reduction." Optimization Methods and Software 38, no. 1 (2023): 91-106.
>
> Ali Ramezani-Kebrya, Fartash Faghri, Ilya Markov, Vitalii Aksenov, Dan Alistarh, and Daniel M. Roy. NUQSGD: Provably communication-efficient data-parallel SGD via nonuniform quantization. Journal of Machine Learning Research (JMLR), 22(114):1–43, 2021.
>
> Dan Alistarh, Demjan Grubic, Jerry Li, Ryota Tomioka, and Milan Vojnovic. QSGD: Communication- efficient SGD via gradient quantization and encoding. In Advances in Neural Information Processing Systems (NeurIPS), 2017.

---

> > ### Comment · Reviewer_696T · 2024-11-18
> >
> > Thanks, I see that there is some novelty to the quantization scheme. I'll raise my score accordingly.

---

> ### Author Response · Authors · 2024-11-19
> **Author Official Comment**
>
> We would first like to thank you for taking the time to discuss with us. We hope to emphasize that our result theoretically (with weaker assumption) and practically surpass that of the global quantization baseline QGen-X in ICLR 2024. Additionally, we introduce further novelty through our layer-wise quantization approach, as outlined in the discussion above. Please let us know if there is anything we can clarify further or improve to enhance the rating.
>
> Regarding "very limited experiments," we have provided realistic experiments, as opposed to simulation in QGen-X, on up to 16 GPUs in a multi-GPU system for a theory-oriented paper. As acknowledged by our section on weaknesses, we agree additional applications beyond training GANs can be provided, but it will also make the paper overly convoluted. Several applications of layer-wise quantization, such as training Transformer-XL on WikiText-103, have been explored in Markov et al. (2022; 2024), which is not our main focus or novelty.
>
> Regarding the "niche" area comment, "global" adaptive methods have been widely studied since QSGD in various setting such as emperical risk minimization (Makarenko et al., 2022; Faghri et al., 2020; Wang et al., 2018; Mishchenko et al. 2024; Guo et al., 2020; Agarwal et al., 2021). Several recent works such as (Xin et al., 2023) suggest that gradient compression methodologies need account for the diverse attributes and that L-Greco adatively adjust layer-specific compression ratios, reducing overall compressed size. Therefore, similar to global quantization methods, we believe that while these layer-wise methods are new and "niche", they have the potential to be adopted more widely in practice in the near future.
>
> In the remaining time of the discussion period, we are more than happy to clarify any other concerns.
>
> References:
>
> 1. Fartash Faghri, Iman Tabrizian, Ilia Markov, Dan Alistarh, Daniel M. Roy, and Ali Ramezani-Kebrya. Adaptive gradient quantization for data-parallel SGD. In Advances in Neural Information Processing Systems (NeurIPS), 2020.
>
> 2. Maksim Makarenko, Elnur Gasanov, Rustem Islamov, Abdurakhmon Sadiev, and Peter Richtárik. Adaptive compression for communication-efficient distributed training. arXiv preprint arXiv:2211.00188, 2022.
>
> 3. Jinrong Guo, Wantao Liu, Wang Wang, Jizhong Han, Ruixuan Li, Yijun Lu, and Songlin Hu. Accelerating distributed deep learning by adaptive gradient quantization. In IEEE International Conference on Acoustics, Speech and Signal Processing, 2020.
>
> 4. Saurabh Agarwal, Hongyi Wang, Kangwook Lee, Shivaram Venkataraman, and Dimitris Papailiopoulos. Adaptive gradient communication via critical learning regime identification. In Conference on Machine Learning and Systems (MLSys), 2021.
>
> 5. Hongyi Wang, Scott Sievert, Shengchao Liu, Zachary Charles, Dimitris Papailiopoulos, and Stephen Wright. Atomo: Communication-efficient learning via atomic sparsification. Advances in Neural Information Processing Systems (NeurIPS), 31, 2018.
>
> 6. Mishchenko, Konstantin, Eduard Gorbunov, Martin Takáč, and Peter Richtárik. "Distributed learning with compressed gradient differences." Optimization Methods and Software (2024): 1-16.
>
> 7. Jihao Xin, Ivan Ilin, Shunkang Zhang, Marco Canini, and Peter Richtárik. Kimad: Adaptive gradient compression with bandwidth awareness. In Proceedings of the 4th International Workshop on Distributed Machine Learning, DistributedML ’23, pp. 35–48, New York, NY, USA, 2023. Association for Computing Machinery. ISBN 9798400704475. doi: 10.1145/3630048.3630184. URL https://doi.org/10.1145/3630048.3630184.
>
> 8. Ilia Markov, Hamidreza Ramezanikebrya, and Dan Alistarh. Cgx: adaptive system support for communication-efficient deep learning. In Proceedings of the 23rd ACM/IFIP International Middleware Conference, pp. 241–254, 2022.
>
> 9. Ilia Markov, Kaveh Alim, Elias Frantar, and Dan Alistarh. L-greco: Layerwise-adaptive gradient compression for efficient data-parallel deep learning. Proceedings of Machine Learning and Systems, 6:312–324, 2024.

---

> > ### Author Response · Authors · 2024-11-29
> >
> > Dear Reviewer 696T,
> >
> > We have revised the manuscript according to several of your recommendations. While you have acknowledged the novelty of our frameworks, you also have concerns regarding the "very limited experiments" and "niche area." We have addressed those points in the previous response, so we hope you can have a look and clarify any other concerns you have in the remaining time of the discussion period.
> >
> > Best,
> >
> > Submission 6044 Authors

---

### Author Response · Authors · 2024-11-26
**Summary of Revisions**

Dear Reviewers,

Incorporating valuable comments and recommendations from reviewers, we hope summarize the main changes in the revised manuscript as follows:
1. Regarding Reviewer 696T's comment on block quantization: We have included two references (Mishchenko et al., 2024, and Horvath et al., 2023) in Related Works Section 1.2 along with a follow-up discussion (Appendix A.2) that highlights how our layer-wise compression fundamentally differs from block quantization. In brief, each of our layer (or block in this context) has different adaptive sequences of levels (refer to our Section 3.1), while block quantization (Mishchenko et al., 2024) applies the same p-quantization scheme to blocks with various sizes. We are grateful that Reviewer 696T has recognized the novelty of our framework after the discussion on block quantization.

2. Regarding Reviewer 696T's comment on using "SoTA" for Q-GenX (Ramezani-Kebrya et al., 2023): We have changed Q-GenX description to the global quantization baseline. Published in ICLR 2023, Q-GenX is a suitable distributed VI-solver with global quantization for us to compare against. Under similar theoretical settings with one less assumption of almost sure boundedness (of the operator), we achieve similar rates as Q-GenX. Our code-length bounds are also generalizations of Q-GenX (Remark 4.2, 4.5). Finally, we practically outperform this global quantization baseline in Figure 1.

3. Regarding the typos raised by reviewers: We have fixed all of the mentioned typos.

4. Regarding Reviewer rc6N's suggestion that the theorems need more remarks for clarity: We have formatted the text after key theorems as remarks for the sake of clarity. Hence there are three new remarks: Remarks 4.2, 4.5, and 4.9. We hope to clarify that these discussions with previous bounds in the literature are present in the past manuscript, so we only now formatted them as remarks for better clarity.

5. Regarding Reviewer qDMJ's comment on a relevant work for layer-wise compression for neural networks (Li et al., 2024): We have include this relevant work in our introduction, and the Section 1.2 of related works that leverage the layer-wise structure of DNNs for more efficient training techniques.

6. Regarding the comment of limited experiments: We have provided realistic experiments, as opposed to simulation in the global quantization baseline QGen-X, on up to 16 GPUs in a multi-GPU system for this theory-oriented paper. As acknowledged by our section on weaknesses, we agree additional applications beyond training GANs can be provided, but it will also make the paper overly convoluted. Several applications of layer-wise quantization, such as training Transformer-XL on WikiText-103, have been explored in (Markov et al. 2022; 2024), which is not our main focus or novelty. We hope our theoretical novelty of the layer-wise compression framework and optimal theoretical guarantees under fewer assumptions outweight this limitation.

We hope these changes strengthen our work and look forward to further discussion if you identify additional areas for improvement. Thank you very much!

Best regards,

Submission 6044 Authors

References:
- Mishchenko, Konstantin, Eduard Gorbunov, Martin Takáč, and Peter Richtárik. "Distributed learning with compressed gradient differences." Optimization Methods and Software (2024): 1-16.

- Horváth, Samuel, Dmitry Kovalev, Konstantin Mishchenko, Peter Richtárik, and Sebastian Stich. "Stochastic distributed learning with gradient quantization and double-variance reduction." Optimization Methods and Software 38, no. 1 (2023): 91-106.

- Dan Alistarh, Demjan Grubic, Jerry Li, Ryota Tomioka, and Milan Vojnovic. QSGD: Communication- efficient SGD via gradient quantization and encoding. In Advances in Neural Information Processing Systems (NeurIPS), 2017.

- Ali Ramezani-Kebrya, Kimon Antonakopoulos, Igor Krawczuk, Justin Deschenaux, and Volkan Cevher. Distributed extra-gradient with optimal complexity and communication guarantees. In International Conference on Learning Representations (ICLR), 2023.

- Li Hanmin, Avetik Karagulyan, and Peter Richtárik. Det-CGD: Compressed Gradient Descent with Matrix Stepsizes for Non-Convex Optimization. In International Conference on Learning Representations (ICLR), 2024

---

### Meta-Review · Area_Chair_aeaM · 2024-12-19

**Metareview:**

Dear Authors,

Thank you for your valuable contribution to ICLR and the ML community. Your submitted paper has undergone a rigorous review process, and I have carefully read and considered the feedback provided by the reviewers.

This work presents a framework for adaptive layer-wise compression in distributed variational inequality problems. The approach uses adaptive learning rates and layer-wise quantization.

The paper received borderline final review scores (5,5,6,6). Reviewers pointed out certain critical issues including (i) issues with clarity and heavy notation, (ii) limited experimental results, (iii) limited novelty of the proposed methods. The authors provided a detailed rebuttal refuting some claims of the reviewers, in particular, regarding the novelty of the layer-wise quantization scheme, and relations to Mishchenko et al., 2024. I disagree the reviewer comment regarding the relevance of the popularity or number of Github stars of the L-Greco repository. Nevertheless, the reviewers did not find the rebuttal convincing enough to improve their scores further.

Given the current form of the paper and the reviewer discussion, I regret to inform you that I am unable to recommend the acceptance of the paper for publication at ICLR. I want to emphasize that this decision should not be viewed as a discouragement. In fact, the reviewers and I believe that your work has valuable insights and, with further development and refinement, can make a meaningful impact on the field.

I encourage you to carefully address the feedback provided by the reviewers and consider resubmitting the paper. Please use the comments and suggestions in the reviews to improve and refine your work.

Best,
AC

**Additional Comments On Reviewer Discussion:**

Reviewers 696T and rc6N pointed out (i) issues with clarity and heavy notation, (ii) limited experimental results, (iii) limited novelty of the proposed methods. The authors provided a detailed rebuttal refuting some claims of the reviewers, in particular, regarding the novelty of the layerwise quantization scheme, and relations to Mishchenko et al., 2024. However, the reviewers did not find the rebuttal convincing enough to improve their scores further.

---

### Decision · Program_Chairs · 2025-01-22

Reject